# LoRA: The Past, Present, and Future

## Abstract

Full fine-tuning of large pre-trained models is constrained by computational and memory overhead, motivating parameter-efficient fine-tuning approaches, such as low-rank adaptation (LoRA) and its variants. Yet, questions remain about their convergence behavior, comparative generalization, and practical limits compared to full fine-tuning. We present a historical framing (the past: full fine-tuning and original LoRA; the present: different variants of LoRA) and introduce simpler, cheaper, parameter-efficient extensions: Cheap LoRA (cLA)—training a single low-rank factor with the other fixed (deterministically or, in its randomized variant, stochastically)—and the chained circulant variant, $c^3$LA. While analyzing these LoRA variants, we realized that nonconvex convergence analysis is only feasible for the variants where one low-rank factor is kept frozen; for LoRA, Lipschitz smoothness of the loss function does not hold. However, we derived information-theoretic generalization error bounds for all variants, which, to our knowledge, is one of the first endeavors in this area. We conduct an extensive empirical study that spans 7 LoRA-based methods and full fine-tuning across 9 pre-trained models on diverse tasks and datasets, and dissect their performance using a multitude of analytical tools, including the loss landscape of the resulting fine-tuned models, their spectral properties, and generalizability. Despite the theoretical results, our experimental study shows that fine-tuning performance, in practice, may or may not be better, depending on the actual trained model, the datasets used, and multiple other factors. In summary, the performance of LoRA-based PEFT methods suggests that using their cheaper variants would be advantageous for effective cost reduction and improved generalizability of pre-trained models.

## 1 Introduction

Full fine-tuning (FFT) (60) modifies a pre-trained neural network's parameters on new datasets that might be relatively expensive to curate, and adapts the network to new downstream tasks. Due to the growth of model sizes and datasets, full fine-tuning is often computationally infeasible or prohibitively costly. Additionally, the growth of these complex models and the hardware's compute capacity are incoherent (12; 59). Large multimodal models (LMMs) such as OpenAI's GPT series (5), Meta's LLaMA (2), Google's Gemini (52), image-text model CLIP (43), video-text model DeepMind's Flamingo (3), etc., are pre-trained on massive high-quality data corpora and fine-tuned to adapt to different tasks or domains. The smallest variant of the recent large language model, Llama-3 (2), has 8B parameters; it requires 32GB GPU memory to load and 64GB GPU memory to train with state-of-the-art training protocols. Compared to this, the half-precision of the H100 GPU accelerator released in 2022 is barely $2.4\times$ more than its 2020-released A100 predecessor, while their memory capacity remains the same (1; 45).

An alternative to FFT, parameter-efficient fine-tuning (PEFT) (24; 60), saves space and time, circumvents overfitting, and is widely used. In that direction, low-rank adaptation (LoRA) (24) achieves albeit similar performance to fully fine-tuned models, but with an extreme reduction in trainable parameters. However, questions remain regarding the generalizability of LoRA-adapted models compared to their fully fine-tuned counterparts (49). To mitigate LoRA's flaws, researchers have proposed different variants, such as the chain of LoRA (CoLA) (57), asymmetric LoRA (65), randomized asymmetric chain of LoRA (38), LoRA+ (19), adaptive LoRA (62), and a few others. At the same time, efforts were made to analyze and compare these PEFT methods with full fine-tuning (60; 49; 38). However, none of these benchmarks is conclusive; see Figure 1. We have a limited theoretical understanding of how these methods work, and ignore many real deployment artifacts.

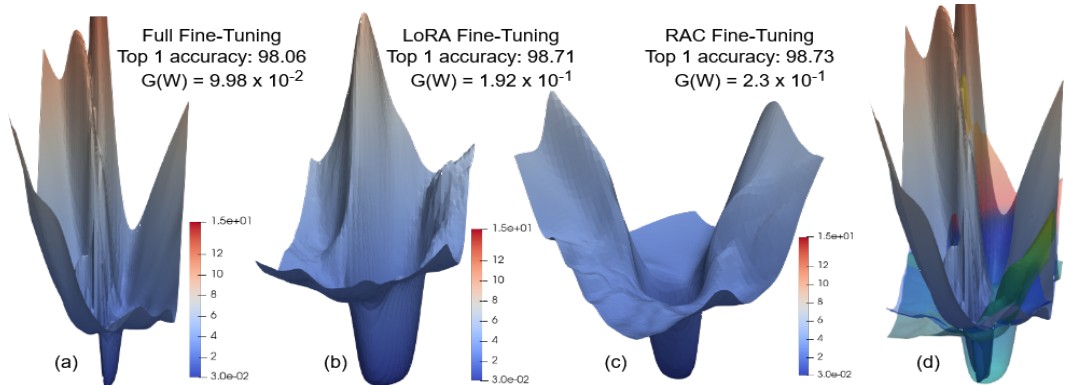

Figure 1: 3D loss landscapes of ViT-Base (11) pretrained on ImageNet-1K (7) and fine-tuned on CIFAR-10 (27) by different fine-tuning strategies, including FFT. FFT has the narrowest volume local minima among the other PEFT methods, and the fine-tuned model renders the worst test accuracy. However, it has the least generalization error, $\mathcal{G}(\mathbf{W})$, among all the methods; see Definition 1 and Table 15. In (d), when we superimpose the loss landscapes, FFT shows the spikiest landscape; RAC (38) has the smoothest landscape with the highest $\mathcal{G}(\mathbf{W})$. According to (29), this is counterintuitive; a model with a spiky landscape and small-volume local minima does not generalize well.

In the era of resource-constrained IoTs and edge deployments (13; 23), pushing parameter efficiency to the point that storage, memory bandwidth, and hardware interface constraints are satisfied during adaptation, and that inference latency benefits from optimized sparse or structured libraries (46; 14) have become a practical imperative. Can sparse training be the new trend? GPT-4.5, the new OpenAI LLM, has an order of magnitude larger parameters compared to GPT4's 1.3T, but only obtained a marginal performance improvement, and could be indicative of the idea that effective parameter reduction might be beneficial for these models (15). At this end, *we propose four simpler, cheaper, and parameter-efficient extensions of LoRA*: Cheap LoRA (cLA), which trains only one low-rank factor and sets the other low-rank factor deterministically, its randomized variant, random-cLA, its chain circulant variant, $c^3$LA, and its randomized chain variant, random-$c^3$LA. But how does the theoretical behavior of these methods practically compare on different fine-tuning tasks? In practice, are there significant differences and trade-offs in terms of convergence behavior and performance of our reduced-parameter LoRA variants? And if there are, how do these differences vary across PEFT methods, hyperparameter configurations, and DNN models? To answer these questions, we make the following contributions:

**Theoretical insights through generalization and nonconvex convergence (§3).** To gain more insights into the PEFT methods discussed in this paper, we use an *information-theoretic approach* to measure their generalization error bounds. See summary of results in Table 1. We also adapt the optimization framework of (38), and present the convergence analysis of the PEFT methods for smooth, nonconvex loss functions, under our *layerwise setup*, where each layer's adapters are updated using gradient descent (GD) and show $O(T^{-1})$ convergence rate for these methods.

**Evaluation and benchmarking (§4).** We empirically evaluate 9 LoRA-based fine-tuning methods (LoRA (24), CoLA (57), Asymmetric LoRA (65), RAC LoRA (38), LoRA Plus (19), cLA, r-cLA, $c^3$LA, and r-$c^3$LA) and full fine-tuning, encompassing 9 different pretrained models: (*i*) GPT2-small (44), (*ii*) DeBERTa v3 Base (20), (*iii*) DeBERTa v2 XXL (21), (*iv*) RoBERTa Base (35), (*v*) RoBERTa Large (35), (*vi*) Deepseek-Coder-1.3B-base (16), (*vii*) TinyLlama-1.1B (61), and (*viii*) vision Transformers, (ViTs), tiny and base (11), on 4 different fine-tuning tasks, natural language processing on PAWS (64), TREC-50 (31), and various GLUE benchmarks (56), image recognition on OfficeHome (55) and CIFAR-10 (27), coding generation on DJANGO (40), and logical reasoning tasks on OpenBookQA (39), FOLIO (17), LogiQA (33), and CLUTRR (50) datasets.

## 2 DNN FINE-TUNING: THE PAST, PRESENT, AND FUTURE

Historically, full fine-tuning updates all parameters of deep networks, an approach that becomes increasingly impractical as model size and deployment multiplicity grow. This leads to the advent of LoRA and its variants. Based on their evolutionary timeline, we divide this section into three temporal phases. The *past* contains full-fine tuning, and we introduce LoRA, while different LoRA variants dominate the *present*. Finally, extreme compute efficiency characterizes the *future* where we propose our new variants.

## 2.1 THE PAST: FULL FINE-TUNING (FFT) AND LoRA

**Pre-training.** Without loss of generality, consider a $L$-layer, fully-connected, neural network whose layers are, $\{W^i\}_{i=1}^L$, where $W^i \in \mathbb{R}^{n_i \times m_i}$ are trainable weights. Let $x \in \mathbb{R}^{m_1}$ be the input and $\mathbf{W} = (W^1, ..., W^L)$. The network $f_{\mathbf{W}}(\cdot) : \mathbb{R}^{d_{\text{in}}} \to \mathbb{R}^{d_{\text{out}}}$ is of the form:

$$f_{\mathbf{W}}(x) = \sigma^L(W^L \cdots \sigma^3(W^3 \sigma^2(W^2 \sigma^1(W^1(x))...))), \tag{1}$$

where $\sigma_i(\cdot) : \mathbb{R}^{n_i} \to \mathbb{R}^{n_i}$ is a nonlinear activation function for the $i^{\text{th}}$ layer. Given a pre-training set, $(x_i, y_i) \in N_{\text{pre}} \subset \mathbb{R}^{m_1} \times \mathbb{R}^{d_{\text{out}}}$, and the loss function, $\ell_{\text{pre}}(\cdot) : \mathbb{R}^{d_{\text{out}}} \times \mathbb{R}^{d_{\text{out}}} \to \mathbb{R}$, we train the network by solving:

$$\mathbf{W}_0 \approx \text{argmin}_{\mathbf{W}} \left[ \mathcal{L}_{\text{pre}}(\mathbf{W}) \overset{\text{def}}{=} \frac{1}{|N_{\text{pre}}|} \sum_{i=1}^{|N_{\text{pre}}|} \ell_{\text{pre}}(f_{\mathbf{W}}(x_i), y_i) \right], \tag{2}$$

obtaining the trained weights $\mathbf{W}_0 = [W_0^1, \cdots, W_0^L]$. Sophisticated DNNs, such as CNNs, RNNs, Transformers, etc., can be adapted with some modification to (1).

**Full fine-tuning (FFT) (9; 22; 60; 24).** Given the pre-trained weights, $\mathbf{W}_0$, FFT updates each DNN layer with corresponding $\Delta W^i$ to adapt the model to a downstream task defined by the domain-specific training data, $(x_i', y_i') \in N$. Denote $\Delta \mathbf{W}$ as the update, and define $\mathbf{W}_0 + \Delta \mathbf{W} := [W_0^1 + \Delta W^1, \cdots, W_0^L + \Delta W^L]$. Given a loss function, $\ell(\cdot) : \mathbb{R}^{d_{\text{out}} \times d_{\text{out}}} \to \mathbb{R}$, FFT updates the model weights by solving:

$$\Delta \hat{\mathbf{W}} \approx \text{argmin}_{\Delta \mathbf{W}} \left[ \mathcal{L}(\mathbf{W}_0 + \Delta \mathbf{W}) \overset{\text{def}}{=} \frac{1}{|N|} \sum_{i=1}^{|N|} \ell(f_{\mathbf{W}_0 + \Delta \mathbf{W}}(x_i'), y_i') \right], \tag{3}$$

and obtains the fine-tuned model, $f_{\mathbf{W}_0 + \Delta \hat{\mathbf{W}}}$, adapted to the downstream task. The computational overhead for FFT can be prohibitively expensive. E.g., LLMs for task-specific fine-tuning. In contrast, parameter-efficient fine-tuning (PEFT) trains orders of magnitude fewer parameters while often attaining performance comparable to FFT (22; 60).

**LoRA (24)** is a popular PEFT method that replaces the layer-wise updates $\Delta W^i$ with a low-rank representation $B^i A^i$, such that $B^i \in \mathbb{R}^{n_i \times r}$, $A^i \in \mathbb{R}^{r \times m_i}$, $r \ll \min(m_i, n_i)$ for all $i \in [L]$. Denote $\mathbf{W}_0 + \frac{\alpha}{r} \mathbf{BA} := [W_0^1 + \frac{\alpha}{r} B^1 A^1, \cdots, W_0^L + \frac{\alpha}{r} B^L A^L]$, where $\alpha > 0$ is a scaling factor. LoRA initializes each $B^i = 0$, $A^i \sim \mathcal{N}(0, 0.02^2)$, and solves:

$$(\hat{\mathbf{B}}, \hat{\mathbf{A}}) \approx \text{argmin}_{\mathbf{B}, \mathbf{A}} \left[ \mathcal{L}(\mathbf{W}_0 + \frac{\alpha}{r} \mathbf{BA}) := \frac{1}{|N|} \sum_{i=1}^{|N|} \ell(f_{\mathbf{W}_0 + \frac{\alpha}{r} \mathbf{BA}}(x_i'), y_i') \right], \tag{4}$$

to obtain $B^i, A^i$ for each layer that results in the fine-tuned model. LoRA may not need to be applied to all layers; some layers can remain frozen (24). LoRA substantially reduces trainable parameters, saves training time, and the update $\mathbf{BA}$ can be merged into the base weights to avoid additional inference latency. For adapting the same pre-trained model to multiple downstream $K$ tasks, each update, $\{\hat{\mathbf{B}}_j \hat{\mathbf{A}}_j\}_{j=1}^K$, is stored separately. Then each task can be switched to by taking the current model $f_{\mathbf{W}_0 + \frac{\alpha}{r} \hat{\mathbf{B}}_j \hat{\mathbf{A}}_j}$, for $j \in [K]$, subtracting the current update $\hat{\mathbf{B}}_j \hat{\mathbf{A}}_j$, and adding the update corresponding to the new task. LoRA is computationally and storage-efficient, but renders worse generalization compared to FFT (49); LoRA may also fail (25).

## 2.2 THE PRESENT: EVOLUTION OF LoRA

Many variants of LoRA exist to enhance efficiency while addressing weaknesses. They excel in certain tasks but are less optimal in others. Including full fine-tuning, empirical evidence suggests that no single fine-tuning method is the best fit for all cases, and that different variations are successful in varying circumstances (60). Thus, there exists compelling reasoning as to why new variants of LoRA continue to emerge. For limited space, we move the discussion in §A.

## 2.3 THE FUTURE: CAN WE PUSH FOR MORE COMPUTE EFFICIENCY?

With rapidly increasing model dimensionality amplifying memory and adaptation costs, we characterize this phase as one of the next evolutionary steps for LoRA: maximizing efficiency while maintaining parity with the current variants of LoRA. Training $B$ generally performs better (65), together with insights from structured chaining methods (57; 38), leads us to two simple, easy-to-

analyze and implement variants, where we postulate that the update of the pre-trained parameter can be restricted to $r$ columns of $B$.

(*i*) **Cheap LoRA (cLA)** is a simplified instance of Asymmetric LoRA (65), where only the low-rank factor $B$ is optimized, while $A$ is kept fixed to the Identity matrix of rank $r$ concatenated with zeros. We consider two instantiations of the fixed factor, deterministic (*cLA*) and random (*random-cLA*). Empirical results show that the deterministic choice suffices (the randomized variant does not yield better performance), even though the random version is more convenient for convergence analysis. In cLA, the fixed matrix, $A^i$, for each layer $i$, is set to an $r \times r$ identity matrix, concatenated with $\mathbf{0}_{r \times m_i - r}$, and is of the form $A^i := \left[ I_r | \mathbf{0}_{r \times (m_i - r)} \right] \in \mathbb{R}^{r \times m_i}$. For each layer, with $W^i \in \mathbb{R}^{n_i \times m_i}$, and $B^i \in \mathbb{R}^{n_i \times r}$, we have $\Delta W^i = B^i \left[ I_r | \mathbf{0}_{r \times (m_i - r)} \right] = \left[ B^i | \mathbf{0}_{n_i \times (m_i - n_i)} \right]$. Denote $\mathbf{B}^c$ as the layer-wise update with $B^i, A^i$ chosen above, and $\mathbf{W}_0 + \frac{\alpha}{r} \mathbf{B}^c := \left[ W_0^1 + \frac{\alpha}{r} B^1 \left[ I_r | \mathbf{0}_{r \times (m_1 - r)} \right], \cdots, W_0^L + \frac{\alpha}{r} B^L \left[ I_r | \mathbf{0}_{r \times (m_L - r)} \right] \right]$. Then cLA solves:

$$\hat{\mathbf{B}}^c \approx \text{argmin}_{\mathbf{B}^c} \left[ \mathcal{L}(\mathbf{W}_0 + \frac{\alpha}{r} \mathbf{B}^c) = \frac{1}{|N|} \sum_{i=1}^{|N|} \ell(f_{\mathbf{W}_0 + \frac{\alpha}{r} \mathbf{B}^c}(x_i'), y_i') \right]. \tag{5}$$

(*ii*) **Circulant Chain of Cheap LoRA ($c^3$LA).** As noted in CoLA (57) and RAC-LoRA (38), chaining LoRA modules leverages repeated initializations to avoid poor minima. We extend this principle to cLA with a structured chaining, $c^3$LA. This method shifts the identity $I_r$ in each matrix $\left[ I_r | \mathbf{0}_{r \times (m_i - r)} \right]$ by $r$ columns to the left. That is, starting with $\left[ I_r | \mathbf{0}_{r \times (m_i - r)} \right]$, the next chain is $\left[ \mathbf{0}_{r \times r} \mid I_r \mid \mathbf{0}_{r \times (m_i - 2r)} \right]$, and so on. Let $\mathbf{B}^{c^3}$ denote $c^3$LA's update and denote $\mathbf{W}^{(k, B^{c^3})} := \mathbf{W}_0 + \sum_{j=1}^k \frac{\alpha}{r} \hat{\mathbf{B}}^{c^3, j}$, and $\mathbf{W}^{(0, B^{c^3})} = \mathbf{W}_0$, then $c^3$LA of chain length $k$ solves:

$$\text{For } j \in [k], \quad \hat{\mathbf{B}}^{c^3, j} \approx \text{argmin}_{\mathbf{B}^{c^3, j}} \left[ \mathcal{L} \left( \mathbf{W}_0^{(j-1, B^{c^3})} + \frac{\alpha}{r} \hat{\mathbf{B}}^{c^3, j} \right) \right], \tag{6}$$

to obtain the fine-tuned model $f_{\mathbf{W}^{(k, B^{c^3})}}$ for a chain of length $k$. Given sufficient epochs and chain length, this ensures we can update all elements in each $W_0$ layer-wise. We formalize this in the following proposition.

**Proposition 1.** *Let $k \in \mathbb{N}$ be such that $d_{in} = kr$. Let $E$ be the total number of epochs used in $c^3 LA$ fine-tuning. Then by creating a new chain in every $\lfloor \frac{E}{k} \rfloor$ epochs, $c^3 LA$ updates each element in $W_0$.*

The intuition behind $c^3$LA goes beyond merely chaining cheap LoRA modules; its structured shifts expand the representational capacity of the learned $B$ matrices. We provide pseudocode of our proposed variants in §B.

## 3 THEORETICAL INSIGHTS

In this Section, we follow two different angles: (*i*) we use an information-theoretic approach to measure the generalization error bounds; (*ii*) we adapt the optimization framework of (38), and present the convergence analysis of the PEFT methods for smooth, nonconvex loss functions, under our layerwise setup, where each layer's adapters are updated using gradient descent (GD).

### 3.1 ON THE GENERALIZATION OF DIFFERENT VARIANTS OF LORA

In this section, we provide the generalization error upper bounds of the PEFT methods discussed in this paper under an information-theoretic framework (48; 58).

Generalizability measures how well a model's loss on its training dataset represents the model's loss on its entire feature space, and demonstrates the model's capacity to avoid overfitting. Let $\mathcal{X} \times \mathcal{Y}$ be an input space and label space with $\nu$ distribution of pairs $(x, y) \in \mathcal{X} \times \mathcal{Y}$. Let $N = \{(x_i, y_i)\}_{i=1}^{|N|}$ represent the training dataset, where each $(x_i, y_i)$ is i.i.d. from $\nu$ distribution of $\mathcal{X} \times \mathcal{Y}$. Given a hypothesis, $f_{\mathbf{W}}(\cdot) : \mathcal{X} \to \mathcal{Y}$, and a nonnegative loss function, $\ell(\cdot) : \mathcal{Y} \times \mathcal{Y} \to \mathbb{R}$, the empirical risk of a hypothesis on the dataset is defined as, $\mathcal{L}(\mathbf{W}) := \frac{1}{|N|} \sum_{i=1}^{|N|} \ell(f_{\mathbf{W}}(x_i), y_i)$. The true risk of the hypothesis $f_{\mathbf{W}}(\cdot)$ is defined as, $\hat{\mathcal{L}}_{\text{global}}(\mathbf{W}) := \mathbb{E}_{\mathcal{X}, \mathcal{Y} \sim \nu}[\ell(f_{\mathbf{W}}(X), Y)]$. With the above setup, next we define *generalization error*, which tells us how well the hypothesis, $f_{\mathbf{W}}$, generalizes from the training sample to the underlying population distribution.

**Definition 1.** (Generalization Error (58)) The generalization error, $\mathcal{G}(\mathbf{W})$, is the difference between a hypothesis's true risk and its empirical risk on the training dataset, i.e., $\mathcal{G}(\mathbf{W}) := \hat{\mathcal{L}}_{\text{global}}(\mathbf{W}) - \mathcal{L}(\mathbf{W})$.

| Variant | Reference | Chain Construction? | Non-convex convergence applies? | Upper bound on $\mathcal{G}(\mathbf{W}_0 + \Delta\mathbf{W})$ |
|---------|-----------|---------------------|--------------------------------|-----------------------------------|
| LoRA | (24) | $\times$ | $\times$ | $\Phi_{\mathbf{W}_0} + \sqrt{\frac{2rq\sigma^2 \ln 2 \sum_{i=1}^{L}(m_i+n_i)}{|N|}}$ |
| LoRA+ | (19) | $\times$ | $\times$ | $\Phi_{\mathbf{W}_0} + \sqrt{\frac{2rq\sigma^2 \ln 2 \sum_{i=1}^{L}(m_i+n_i)}{|N|}}$ |
| Asym-LoRA | (65) | $\times$ | $\checkmark$ | $\Phi_{\mathbf{W}_0} + \sqrt{\frac{2rq\sigma^2 \ln 2 \sum_{i=1}^{L} n_i}{|N|}}$ |
| CoLA | (57) | $\checkmark$ | $\times$ | $\Phi_{\mathbf{W}_0} + \sqrt{\frac{2rq\sigma^2 k \ln 2 \sum_{i=1}^{L}(m_i+n_i)}{|N|}}$ |
| RAC | (38) | $\checkmark$ | $\checkmark$ | $\Phi_{\mathbf{W}_0} + \sqrt{\frac{2rq\sigma^2 k \ln 2 \sum_{i=1}^{L} n_i}{|N|}}$ |
| *random-cLA* | This paper | $\times$ | $\checkmark$ | $\Phi_{\mathbf{W}_0} + \sqrt{\frac{2rq\sigma^2 \ln 2 \sum_{i=1}^{L} n_i}{|N|}}$ |
| $c^3$LA | This paper | $\checkmark$ | $\checkmark$ | $\Phi_{\mathbf{W}_0} + \sqrt{\frac{2rq\sigma^2 k \ln 2 \sum_{i=1}^{L} n_i}{|N|}}$ |

Table 1: **Generalization error upper bounds** of LoRA variants. The expression, $\Phi_{\mathbf{W}_0}$ is in Theorem 1. Note that, $r$ is the adapter rank, $k$ is the chain length, $|N|$ is the size of fine-tuned dataset, $q$ is the quantization bitwidth, $(m_i, n_i)$ are the input and output dimensions of the $i^{\text{th}}$ layer, and the loss, $\mathcal{L}$ is $\sigma$-sub-Gaussian (Assumption 6).

**Assumption 1.** *(Boundedness of input vectors) The input vectors are bounded, i.e., there exists a constant $C \geq 0$ such that $\|x\| \leq C$, for all $x \in \mathcal{X}$.*

**Assumption 2.** *(Lipschitz continuity of the loss) The loss function, $\ell(\cdot) : \mathbb{R}^d \to \mathbb{R}$ is $L_{\mathcal{L}}$-Lipschitz continuous, i.e., $|\ell(f_{\mathbf{W}}(x), y) - \ell(f_{\mathbf{W}'}(x), y)| \leq L_{\mathcal{L}} \|f_{\mathbf{W}}(x) - f_{\mathbf{W}'}(x)\|$ for all $\mathbf{W}, \mathbf{W}' \in \mathbb{R}^d$ and $(x, y) \in \mathcal{X} \times \mathcal{Y}$.*

**Assumption 3.** *(Lipschitz continuity of activation) The vector-valued activation function, $\sigma_i(\cdot) : \mathbb{R}^{n_i} \to \mathbb{R}^{n_i}$, for each layer, $i$, is $L_{\sigma_i}$-Lipschitz continuous, i.e., $\|\sigma_i(u) - \sigma_i(v)\| \leq L_{\sigma_i} \|u - v\|$, for all $u, v \in \mathbb{R}^{n_i}$.*

The following theorem upper bounds the generalization error of a fine-tuned, $L$-layer fully connected DNN, parameterized by $\mathbf{W}_0 + \Delta\mathbf{W}$, by the better of two alternatives: the generalization error of $\mathbf{W}_0$ and a correction term, or the generalization error of $\Delta\mathbf{W}$ and a different correction term.

**Theorem 1.** *(Generalization bounds) Let $f_{\mathbf{W}_0+\Delta\mathbf{W}}(x) = \sigma_L([W_0^L + \Delta W^L](\cdots \sigma_2([(W_0^2 + \Delta W^2]\sigma_1([W_0^1 + \Delta W^1]x))\cdots))$ be a $L$-layers fine-tuned DNN, where $\mathbf{W}_0 + \Delta\mathbf{W}$ is a fine-tuned update. Let the loss function, $\mathcal{L}$ for fine-tuning, follow Assumption 2 and Assumptions 1–3 hold. Then $\mathcal{G}(\mathbf{W}_0 + \Delta\mathbf{W}) \leq \min\left(\mathcal{G}(\mathbf{W}_0) + \Phi_{\Delta\mathbf{W}}, \mathcal{G}(\Delta\mathbf{W}) + \Phi_{\mathbf{W}_0}\right)$, where*

$$\Phi_{\Delta\mathbf{W}} := 2L_{\mathcal{L}}\left[C \prod_{i=1}^{L} L_{\sigma_i} \sum_{i=1}^{2^L-1} \prod_{j=1}^{L} P(i,j) + \sum_{i \neq 2^a-1 : a \in [L]}^{2^L-1} F(i)\right] \text{ and}$$

$$\Phi_{\mathbf{W}_0} := 2L_{\mathcal{L}}\left[C \prod_{i=1}^{L} L_{\sigma_i} \sum_{i=2}^{2^L} \prod_{j=1}^{L} P(i,j) + \sum_{i \neq 2^a : a \in [L]}^{2^L-1} F(i)\right],$$

*are the correction terms, $F(i) := \|\sigma_{L-\psi(i)}(0)\| \prod_{j=1}^{\psi(i)}[L_{\sigma_{L-j+1}} H(i,j)], \psi(i) := \lfloor \log_2(i) \rfloor$, and*

$$P(i,j) := \begin{cases} \|W_0^{(L-j+1)}\| & \text{if } \lfloor \frac{i-1}{2^{L-1}} \rfloor \text{ is odd,} \\ \|\Delta W^{(L-j+1)}\| & \text{if } \lfloor \frac{i-1}{2^{L-1}} \rfloor \text{ is even} \end{cases} \qquad H(i,j) := \begin{cases} \|\Delta W^{(L-j+1)}\| & \text{if } \lfloor \frac{i}{2^{\psi(i)-j}} \rfloor \text{ is odd,} \\ \|W_0^{(L-j+1)}\| & \text{if } \lfloor \frac{i}{2^{\psi(i)-j}} \rfloor \text{ is even.} \end{cases}$$

In Theorem 1, the expression, $\sum_{i=1}^{2^L} \prod_{j=1}^{L} P(i,j) = (\|W_0^{(L)}\| + \|\Delta W^{(L)}\|)(\|W_0^{(L-1)}\| + \|\Delta W^{(L-1)}\|)\cdots(\|W_0^{(1)}\| + \|\Delta W^{(1)}\|)$ is the sum of the product of all possible combinations of $\{\|W_0^{(i)}\|, \|\Delta W^{(i)}\|\}_{i \in [L]}$. We note that $\prod_{j=1}^{L} P(2^L, j) := \|W_0^{(L)}\|\|W_0^{(L-1)}\|\cdots\|W_0^{(1)}\|$, and $\prod_{j=1}^{L} P(1,j) := \|\Delta W^{(L)}\|\|\Delta W^{(L-1)}\|\cdots\|\Delta W^{(1)}\|$, as they are the terms not included in $\Phi_{\mathbf{W}_0}$ and $\Phi_{\Delta\mathbf{W}}$, respectively. The term, $F(i)$, represents the sum of all offset terms $\|\sigma_{i'}(0)\|$ based on the recursive collapse of the difference of $\|f_{\mathbf{W}_0+\Delta\mathbf{W}} - f_{\mathbf{W}_0}\|$; see Figure 4 for an illustration. In §C.1.4, we show that the bounds provided in Theorem 1 are tight, and we present special cases in §C.1.3.

**Theorem 1 under special conditions.** The generalization upper bound $\mathcal{G}(\mathbf{W}_0 + \Delta\mathbf{W})$ in Theorem 1 contains two terms: (*i*) $\mathcal{G}(\mathbf{W}_0) + \Phi_{\Delta\mathbf{W}}$ and (*ii*) $\mathcal{G}(\Delta\mathbf{W}) + \Phi_{\mathbf{W}_0}$. We can adapt some additional assumptions on loss, quantization bit-width, size of fine-tuning datasets, and layer dimensions; see §C.1.5 and bound $\mathcal{G}(\mathbf{W}_0)$ and $\mathcal{G}(\Delta\mathbf{W})$.

(*i*) **Bounding $\mathcal{G}(\mathbf{W}_0)$.** We use the PAC-Bayes generalization bound for fine-tuning using Theorem 4.1

in (28); see Theorem 3 in §C.1.5. The loss function, $\mathcal{L}$, is bounded by $C_2$. Since $\|W_0^{(i)} - W_0^{(i)}\| = 0$, for all $i \in [L]$, in Theorem 3, we obtain $Q_i := 0$. Hence, $\mathcal{G}(\mathbf{W}_0) \le \epsilon + C_2\sqrt{|N|^{-1}(3\ln|N|\delta^{-1} + 8)}$, holds with probability at least $1 - 2\delta$, where $\epsilon, \delta > 0$ are arbitrary small numbers. Together with Theorem 1, we arrive at $\mathcal{G}(\mathbf{W}_0 + \Delta\mathbf{W}) \le \epsilon + C_2\sqrt{|N|^{-1}(3\ln|N|\delta^{-1} + 8)} + \Phi_{\Delta\mathbf{W}}$; we quote this result formally in Theorem 4 in §C.1.5.

(*ii*) **Bounding $\mathcal{G}(\Delta\mathbf{W})$.** For a DNN, let $q$ be the quantization bitwidth. We assume $\mathcal{L}$ is $\sigma$-sub-gaussian for all $\mathbf{W}$ and use the generalization upper bound of $\mathcal{G}(\Delta\mathbf{W})$ as in Lemma 4.5 of (65), for each PEFT method. Together with Theorem 1, we arrive at $\mathcal{G}(\Delta\mathbf{W}_0 + \Delta\mathbf{W}) \le \Phi_{\mathbf{W}_0} + \mathcal{G}(\mathbf{BA})$, where $\mathcal{G}(\mathbf{BA})$ represents the generalization error of different LoRA variants; see Table 1 and § C.1.5.

Although Table 1 demonstrates the generalization behavior of different PEFT methods, we emphasize that these results are upper bounds, and in practice, different DNN models may deviate from them. E.g., Although $c^3$LA has similar upper bounds to RAC-LoRA in theory, in practice, we notice stronger generalization trends for $c^3$LA in comparison to all other variants; see Tables 3 and 15.

## 3.2 ON THE NONCONVEX CONVERGENCE OF DIFFERENT VARIANTS OF LoRA

We adapt the convergence results of (38) for the layer-wise case, where each layer is updated using GD, and the rest are kept frozen. For our analysis, we make some general assumptions in §C.2.

**The update step.** Our structure considers layer-wise gradient calculation, which is a natural artifact in deep learning toolkits. The update step follows directly from (38): For each layer $i \in [L]$, the update step of the $i^{\text{th}}$ layer with RAC-LoRA structure is
$$W^{i,t+1} = W^{i,t} - \gamma\nabla_i f(\mathbf{W}^t)H^i,$$
where $H^i = (A_0^i)^\top (A_0^i (A_0^i)^\top)^\dagger A_0^i$ is a projection matrix.

The smallest eigenvalue of the expected projection matrix, $H^i$, plays a critical role in the optimization process (38). For each layer $i \in [L]$, let $A_0^i \sim \mathcal{D}^i$, where $\mathcal{D}^i$ is the set of all possible $A_0^i$s when training the model. We denote $\lambda_{\min}^{H,i} := \lambda_{\min}[\mathbb{E}_{\mathcal{D}^i}[H^i]]$ and let $\lambda_{\min}^g := \min\{\lambda_{\min}^{H,i}\}_{i=1}^L$ be the smallest. Now we are set to state our convergence result.

**Theorem 2.** *(Nonconvex convergence)* Let Assumption 7 and 8 hold. Let $\lambda_{\min}^g > 0$ and the stepsize satisfy $0 < \gamma < \frac{1}{L_G}$. Let $\mathbf{W}^{(t,\cdot)}$ represent RAC-LoRA (9), or random-cLA (5), or $c^3$LA (6) update trained using gradient descent. Then $\{\mathbf{W}^{(0,\cdot)}, ..., \mathbf{W}^{(T,\cdot)}\}$ satisfies $\mathbb{E}[\|\nabla\mathcal{L}(\tilde{\mathbf{W}}^{(T)})\|^2] \le \frac{2(\mathcal{L}(\mathbf{W}_0) - \mathcal{L}^*)}{\lambda_{\min}^g \gamma T}$, where $\tilde{\mathbf{W}}^{(T)}$ is sampled uniformly at random from $\{\mathbf{W}^{(0,\cdot)}, ..., \mathbf{W}^{(T,\cdot)}\}$.

Next, we adapt Theorem 2, to show the convergence of Asymmetric-LoRA (8), RAC-LoRA (9), *random-cLA* (5), and $c^3$LA (6) by explicitly determining $\lambda_{\min}^g$ for each PEFT method's from their commonly chosen $D^i$ in practice; see Table 1. For Asymmetric-LoRA and RAC-LoRA, $D^i = \mathcal{N}(0, \sigma^2)$ (65; 38), and $\lambda_{\min}^{H,i} = \frac{r}{n_i}$. This implies, $\lambda_{\min}^g = \frac{r}{n_{\max}}$, where $n_{\max} := \max_{i \in [L]} n_i$. For $c^3$LA and *random-cLA*, $\lambda_{\min}^{H,i} = \frac{r}{n_i}$, for $i \in [L]$; see Proposition 4.

In the LoRA adaptation, the Lipschitz smoothness is lost even if the loss, $\mathcal{L}$, is Lipschitz-smooth (38). One can recover $\mathcal{L}$-smoothness when freezing one of the matrices $B$ or $A$; see Theorem 6 in §C.2. Therefore, Theorem 2 cannot be used to describe LoRA, LoRA+, and CoLA's convergence behavior. This is indeed a shortcoming of the result. Moreover, the layerwise analysis does not bring any new insight into the efficacy of the PEFT methods, as the upper bound on $\mathbb{E}[\|\nabla\mathcal{L}(\tilde{\mathbf{W}}^{(T)})\|^2]$ remains the same for all the methods that we could analyze.

## 4 BENCHMARKING AND EVALUATION

Despite theoretical studies, our experimental study of 8 fine-tuning methods shows that fine-tuning, in practice, may or may not be better, depending on the actual pre-trained model, datasets used, and a multitude of other factors. The unpredictable performance of LoRA-based PEFT methods suggests that it would be advantageous to use their cheaper variants for effective cost reduction and a better generalizability of pre-trained models.

**Implementation details and models used.** We provide implementation details of each fine-tuning method in §D.1. Our empirical evaluation encompasses 9 pretrained models: (*i*) DeBERTa v3 Base (20), (*ii*) DeBERTa v2 XXL (21), (*iii*) GPT2-small (44), (*iv*) RoBERTa Base (35), (*v*) RoBERTa

Table 2: Performance of fine-tuned models with adapter rank $r = 16$. We use green, red, and blue to indicate the best, second best, and third best result. For our variants, ↓ indicates the accuracy drop percentage compared to the best.

| Model | Dataset | The Past | | The Present | | | | The Future | | | |
|---|---|---|---|---|---|---|---|---|---|---|---|
| | | FFT | LoRA | CoLA | Asym | RAC | LoRA+ | cLA | $c^3$LA | r-cLA | r-$c^3$LA |
| ViT-Tiny (11) | OfficeHome | 79.68 | 80.13 | 79.54 | 78.02 | 78.55 | 77.87 | 78.01 (↓2.65%) | 78.69 (↓1.80%) | 78.01 (↓2.65%) | 79.32 (↓1.01%) |
| | CIFAR10 | 96.59 | 96.17 | 95.85 | 94.80 | 95.36 | 95.29 | 94.94 (↓1.71%) | 95.23 (↓1.41%) | 95.12 (↓1.52%) | 95.22 (↓1.42%) |
| ViT-Base (11) | OfficeHome | 86.42 | 88.96 | 89.01 | 89.00 | 89.33 | 87.87 | 89.21 | 89.18 | 88.83 | 89.17 |
| | CIFAR10 | 98.06 | 98.71 | 98.48 | 98.68 | 98.73 | 98.36 | 98.63 | 98.54 | 98.78 | 98.72 |
| DeBERTa v2 XXL (21) | MRPC | 87.49 | 88.28 | 87.47 | 87.03 | 86.97 | 87.53 | 86.13 (↓2.44%) | 85.11 (↓3.59%) | 85.55 (↓3.09%) | 85.15 (↓3.55%) |
| | TREC-50 | 91.99 | 91.47 | 85.65 | 92.26 | 92.02 | 84.92 | 91.73 (↓0.57%) | 90.87 (↓1.51%) | 91.67 (↓0.64%) | 91.07 (↓1.29%) |
| | PAWS | 94.69 | 94.97 | 95.22 | 94.95 | 94.66 | 95.20 | 94.77 (↓0.47%) | 94.90 (↓0.34%) | 94.38 (↓0.88%) | 94.71 (↓0.54%) |
| DeBERTa v3 Base (20) | MRPC | 85.80 | 88.33 | 87.91 | 86.40 | 86.34 | 84.51 | 84.43 (↓4.42%) | 80.22 (↓9.18%) | 85.42 (↓3.29%) | 84.17 (↓4.71%) |
| | RTE | 82.47 | 86.34 | 83.80 | 78.94 | 79.40 | 84.72 | 76.00 (↓11.98%) | 75.08 (↓13.04%) | 79.40 (↓8.04%) | 79.40 (↓8.04%) |
| | STSB | 89.52 | 89.09 | 89.34 | 89.04 | 88.71 | 89.15 | 87.56 (↓2.19%) | 87.90 (↓1.81%) | 88.05 (↓1.64%) | 87.90 (↓1.81%) |
| | TREC-50 | 90.15 | 89.29 | 89.88 | 90.67 | 89.22 | 85.52 | 86.04 (↓5.11%) | 87.96 (↓2.99%) | 86.04 (↓5.11%) | 87.70 (↓3.28%) |
| | PAWS | 94.76 | 94.62 | 94.40 | 94.48 | 94.45 | 94.44 | 94.23 | 94.60 | 94.36 | 94.42 |
| RoBERTa-Base (35) | MRPC | 87.40 | 86.34 | 86.76 | 86.40 | 86.67 | 84.29 | 84.83 (↓2.94%) | 84.39 (↓3.44%) | 85.08 (↓2.65%) | 85.33 (↓2.37%) |
| | CoLA | 56.08 | 57.33 | 58.39 | 52.35 | 53.76 | 50.40 | 51.86 (↓11.18%) | 53.29 (↓8.73%) | 52.56 (↓9.98%) | 53.10 (↓9.06%) |
| RoBERTa-Large (35) | MRPC | 87.57 | 88.46 | 88.43 | 87.56 | 87.69 | 72.91 | 87.81 | 86.36 | 86.24 | 86.59 |
| | CoLA | 64.58 | 62.42 | 60.03 | 63.42 | 59.84 | 28.80 | 59.47 (↓7.71%) | 59.60 (↓7.71%) | 58.60 (↓9.26%) | 60.24 (↓6.72%) |
| TinyLlama (61) | OpenBookQA | 55.47 | 52.41 | 52.47 | 45.96 | 47.59 | 53.26 | 44.92 (↓19.02%) | 45.12 (↓18.66%) | 47.07 (↓15.14%) | 27.34 (↓50.71%) |
| | FOLIO | 60.71 | 57.59 | 59.40 | 58.33 | 55.45 | 54.17 | 58.97 | 58.01 | 54.81 | 59.82 |
| | LogiQA | 47.54 | 41.54 | 43.70 | 41.50 | 40.86 | 45.83 | 39.09 (↓17.77%) | 39.30 (↓17.33%) | 39.09 (↓17.77%) | 39.31 (↓17.31%) |
| | CLUTRR | 42.01 | 37.44 | 39.38 | 37.98 | 37.98 | 38.10 | 39.12 | 37.79 | 36.23 | 37.03 |
| DeepseekCoder (16) | DJANGO | 22.73 | 23.60 | 19.79 | 35.12 | 30.27 | 27.27 | 7.83 (↓77.71%) | 19.48 (↓44.53%) | 19.36 (↓44.87%) | 15.34 (↓56.32%) |
| GPT2-Small (44) | E2E | 2.98 | 3.18 | 3.29 | 3.36 | 3.34 | 3.23 | 3.34 (↑12.08%) | 3.29 (↑10.4%) | 3.30 (↑10.7%) | 3.29 (↑10.4%) |

Large (35), (*vi*) DeepseekCoder-1.3B-base (16), (*vii*) TinyLlama-1.1B (61), (*viii*) ViT Base (11), and (*ix*) ViT-Tiny (11). See Table 4 in §D.1 for a detailed summary of the models and Table 5 for reproducibility. We report the epoch when a model has the lowest validation loss.

**Fine-tuning tasks and datasets.** We perform 4 different fine-tuning tasks:(*i*) **Natural Language Processing (NLP).** We use the datasets, PAWS (64), TREC-50 (31), and various GLUE benchmarks (56), including MRPC, CoLA, STS-B, and RTE for NLP tasks. (*ii*) **Image Classification.** We fine-tuned LoRA and its variants on OfficeHome (55) and CIFAR-10 (27). (*iii*) **Coding Generation.** Code generation presents unique challenges; minor deviations can lead to runtime errors or semantic mismatches. There is relatively limited LoRA-focused literature on programming tasks; we evaluate how different LoRA variants adapt to these tasks on DJANGO (40), and report results using Exact Match (EM). (*iv*) **Logical Reasoning.** We use OpenBookQA (39) for elementary science multiple-choice reasoning, FOLIO (17) for natural language reasoning with first-order logic, LogiQA (33) for logical comprehension, and CLUTRR (50) for compositional relational reasoning from text.

## 4.1 QUALITY OF THE FINE-TUNED MODELS

In Table 2, we present fine-tuning performance of various models with full fine-tuning and LoRA-based fine-tuning. For the CoLA dataset, we report the Matthews Correlation Coefficient (the higher the better) (6), for reporting GPT2-small's results, we use perplexity (the lower the better), and for other models and datasets, we report test accuracies (the higher the better). Each model and dataset is trained over three seeds, and we average the results. We find that no one method substantially outperforms the others for adapting the model to their downstream tasks, including FFT, which confirms the previous findings in (60). In many cases, FFT performs rather poorly (e.g., ViT-Base on OfficeHome, DeBERTa v3 on RTE, DeepseekCoder on DJANGO). Importantly, our sparse LoRA variants outperform FFT and LoRA in some tasks by a larger margin (e.g., ViT-Base on OfficeHome, DeBERTA v3 on MRPC); in many cases, their performance drop is modest. We note that our variants cannot always produce the best accuracy in low-epoch fine-tuning but they still generalize well; see Table 15. This suggests that, when fine-tuning a model for a downstream task, it may be optimal to select a fine-tuning method based on its other characteristics and user-specific needs, rather than just the generated accuracy. To highlight this point, in §4.2, we analyze the performance of each method based on its training time, generalizability, and robustness for adapting to further downstream tasks. Additionally, we note that our sparse variants do not reduce the number of trainable parameters, but reduce the number of FLOPs, even with naïve, non-optimized, sparse implementation; see §D.3.

## 4.2 PERFORMANCE ANALYSIS

We dissect the performance of different LoRA variants using the following tools:

(*i*) **Loss Landscape (29)** is a 3D surface that visualizes how the empirical loss of a model differs under small parameter perturbations; see details in §D.4.1. The sharpness of a model's loss landscape correlates with better generalization, and smoother landscapes indicate the PEFT method is more

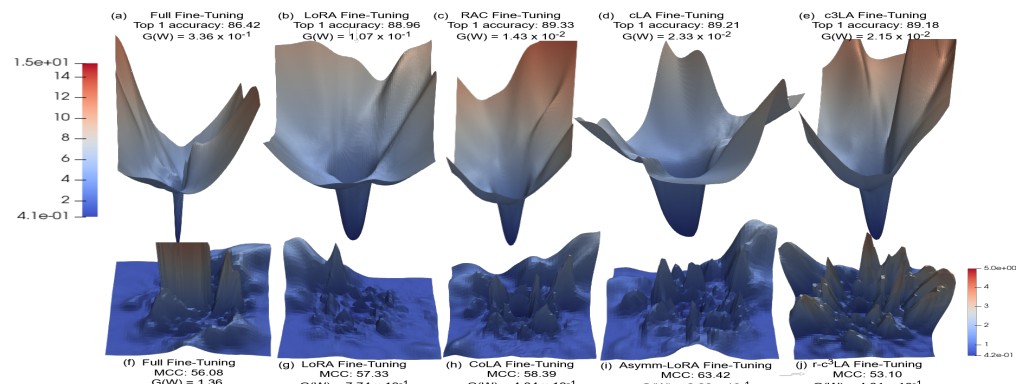

Figure 2: Loss landscapes of ViT-Base fined tuned on OfficeHome (top row) with PCA directions, and RoBERTa-Base fine-tuned on CoLA (bottom row) with random directions. In both cases, we observe the worst generalization error, $\mathcal{G}(\mathbf{W})$, in (a) and (f), respectively, which are the spikiest landscapes in their class of models. Additionally, chain methods consistently produce spikier landscapes.

robust to initialization (29). Fine-tuning pretrained models tends to produce smoother landscapes than training from scratch (18; 37). In Figure 2, top row shows the loss-landscapes of ViT-Base, pretrained on Imagenet-21K, and fine-tuned on OfficeHome by cLA, $c^3$LA, LoRA, FFT, and RAC, while the bottom row shows the loss-landscapes of RoBERTa-Base, pretrained on a large corpus of English data and fine-tuned on CoLA by r-$c^3$LA, LoRA, CoLA, Asymmetric LoRA, and FFT. For ViT-Base, we used PCA directions, whereas for RoBERTa-Base, we used random directions; see §D.4.1, for comparison of these two implementations. We present a direct comparison of non-chain LoRA methods (LoRA, Asymmetric LoRA, cLA) with their chain counterparts (CoLA, RAC-LoRA, $c^3$LA) in Figure 6. In §D.4.1, we plotted the 2D contour plots to show the optimizer path.

Based on the characteristics of the loss landscapes as in (29), FFT would generalize worse, as it has the spikiest losses, and our results in Figure 2 confirm that. Based on the sharpness of the landscapes, chain methods sharpen the minima, and this perspective indicates they should generalize worse. However, in practice, this is not the case. E.g., For ViT-Base, in Figure 2, RAC has the least $\mathcal{G}(\mathbf{W})$. Therefore, the loss landscape rhetoric, as we also witnessed in Figure 1, does not always match with the generalization error. Motivated by this discrepancy, we present an alternative analysis that is more aligned with our empirical observation on the generalization error.

*(ii)* **Intruder Dimensions (49).** Given the pretrained and fine-tuned models, $\mathbf{W}_0$ and $\mathbf{W}_0 + \Delta\mathbf{W}$, the number of intruder dimensions correlates with their performance in a downstream task; see §D.4.2. Higher intruder dimensions correlate to a worse performance (49). We analyze the number of intruder dimensions present in FFT and various LoRA-based PEFT methods for the RoBERTa-Base (35) and ViT-Base (11). We divide the total fine-tuning epochs for each method into 4 equal points, and report the number of intruder dimensions present at $25^{\text{th}}$, $50^{\text{th}}$, $75^{\text{th}}$, and $100^{\text{th}}$ percentile of the training epoch by using the $\varepsilon$-thresholds set at a lower and higher value, 0.4 and 0.8, respectively. Tables 13 and 14 present the average number of intruder dimensions per layer for RoBERTa-Base and ViT-Base, respectively. Figures 3a and 3b present the number of intruder dimensions of a fine-tuned model, obtained from each method by varying the range of threshold, $\varepsilon \in (0, 1]$. As shown in the Figures 3a and 3b, the chain variant of each LoRA-based PEFT method produces more intruders than its non-chain counterpart. This effect is least pronounced in LoRA and CoLA, which produced almost the same number of intruders for RoBERTa. This is consistent with our empirical results in Figure 2—If CoLA produced more intruders than LoRA, it would never have a better $\mathcal{G}(\mathbf{W})$ than LoRA. But, for ViT-Base, this observation does not hold. Also, we note that RAC has the best $\mathcal{G}(\mathbf{W})$ on ViT-Base, while producing substantially more intruders than LoRA and LoRA+. Additionally, from Figure 3c, we find that FFT has the highest intruder dimensions but the least $\mathcal{G}(\mathbf{W})$ (Table 15), LoRA+ has more than average intruder dimensions but the second best $\mathcal{G}(\mathbf{W})$ (Table 15); only LoRA's intruder dimensions and $\mathcal{G}(\mathbf{W})$ follow the correct trend.

*(iii)* **Generalizability.** The generalization error, $\mathcal{G}(\mathbf{W})$ (Definition 1), is hard to realize in practice, as the true distribution of a feature space and label space, $\mathcal{X} \times \mathcal{Y}$, cannot be obtained. Therefore, we cannot use the theoretical bounds on $\mathcal{G}(\mathbf{W})$ in Table 1 without modification. Since test samples are i.i.d. from $(\mathcal{X} \times \mathcal{Y})$, as an alternative, the difference between the loss of a model on a collection of unseen test samples and the loss on its training set approximates how well the model generalizes to the true distribution of the instance space it was trained on. Therefore, we approximate $\mathcal{G}(\mathbf{W}) \approx \mathbb{E}(\mathcal{L}_{\text{test}}) - \mathcal{L}_{\text{train}}$. As the size of the test set increases, the difference approaches the actual $\mathcal{G}(\mathbf{W})$

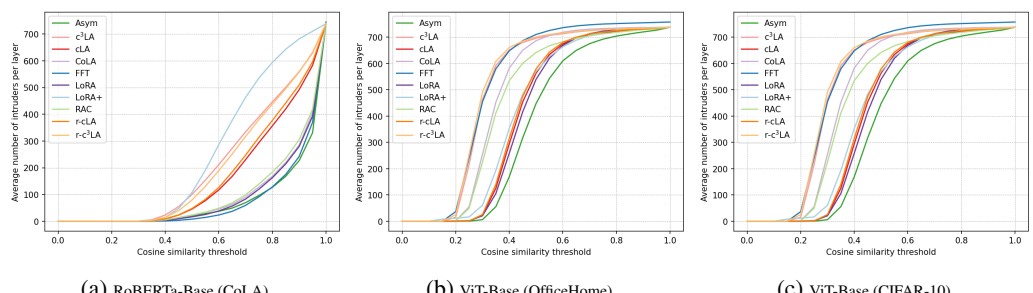

| (a) RoBERTa-Base (CoLA) | (b) ViT-Base (OfficeHome) | (c) ViT-Base (CIFAR-10) |

Figure 3: The average number of intruder dimensions present in different fine-tuned models.

Table 3: Generalization error, $\mathcal{G}(\mathbf{W})$, of the fine-tuning methods over various models and datasets.

| Model | Dataset | The Past | | The Present | | | | The Future | | | |
|---|---|---|---|---|---|---|---|---|---|---|---|
| | | FFT | LoRA | CoLA | Asym | RAC | LoRA+ | cLA | $c^3$LA | r-cLA | r-$c^3$LA |
| ViT-Tiny (11) | OfficeHome | $4.85e^{-1}$ | $6.96e^{-2}$ | $9.55e^{-3}$ | $7.22e^{-2}$ | $6.17e^{-2}$ | $7.39e^{-2}$ | $1.98e^{-2}$ | $3.40e^{-2}$ | $2.16e^{-2}$ | $3.51e^{-2}$ |
| DeBERTa v2 XXL (21) | PAWS | $6.07e^{-2}$ | $1.99e^{-2}$ | $3.63e^{-2}$ | $3.26e^{-2}$ | $3.95e^{-2}$ | $5.41e^{-2}$ | $6.68e^{-2}$ | $5.11e^{-2}$ | $1.98e^{-2}$ | $6.99\ e^{-2}$ |
| DeBERTa v3 Base (20) | MRPC | $1.06e^{-1}$ | $8.90e^{-2}$ | $2.59e^{-2}$ | $7.28e^{-2}$ | $9.86e^{-2}$ | $1.52e^{-2}$ | $2.58e^{-2}$ | $8.52e^{-3}$ | $1.16e^{-1}$ | $2.57e^{-2}$ |
| | TREC50 | $4.56e^{-1}$ | $2.73e^{-1}$ | $3.99e^{-1}$ | $2.16e^{-1}$ | $2.67e^{-1}$ | $2.61e^{-2}$ | $2.25e^{-1}$ | $3.70e^{-1}$ | $3.36e^{-1}$ | $2.63e^{-2}$ |
| | PAWS | $2.62e^{-2}$ | $6.43e^{-2}$ | $2.40e^{-2}$ | $6.27e^{-2}$ | $8.17e^{-2}$ | $5.55e^{-2}$ | $7.39e^{-2}$ | $5.77e^{-2}$ | $1.01e^{-1}$ | $5.82e^{-2}$ |
| RoBERTa-Base (35) | CoLA | 1.39 | $7.74e^{-1}$ | $4.04e^{-1}$ | $2.22e^{-1}$ | $1.96e^{-1}$ | $8.10e^{-1}$ | $4.70e^{-1}$ | $4.43e^{-1}$ | $4.38e^{-1}$ | $4.01e^{-1}$ |
| TinyLlama (61) | OpenBookQA | $1.78e^{-1}$ | $2.82e^{-1}$ | $3.41e^{-1}$ | $2.15e^{-1}$ | $1.86e^{-1}$ | $2.07e^{-1}$ | $1.51e^{-1}$ | $2.20e^{-1}$ | $3.16e^{-1}$ | $7.59e^{-2}$ |
| | FOLIO | $1.82e^{-1}$ | $2.37e^{-1}$ | $2.17e^{-1}$ | $1.75e^{-1}$ | $1.93e^{-1}$ | $5.11e^{-2}$ | $2.35e^{-1}$ | $1.91e^{-1}$ | $1.05e^{-1}$ | $2.49e^{-1}$ |
| | LogiQA | $3.61e^{-1}$ | $6.12e^{-3}$ | $1.45e^{-1}$ | $1.16e^{-2}$ | $1.75e^{-1}$ | $2.37e^{-1}$ | $8.60e^{-2}$ | $1.1e^{-1}$ | $6.64e^{-2}$ | $6.25e^{-2}$ |
| | CLUTRR | 4.29 | 2.25 | 1.55 | 2.34 | 2.27 | 5.48 | 2.16 | 2.19 | 2.59 | 4.23 |
| DeepseekCoder (16) | DJANGO | $3.48e^{-2}$ | $4.65e^{-2}$ | $3.4e^{-2}$ | $5.16e^{-2}$ | $4.64e^{-2}$ | $3.87e^{-2}$ | $4.19e^{-2}$ | $3.89e^{-2}$ | $3.64e^{-2}$ | $3.62e^{-2}$ |
| GPT2-Small (44) | E2E | $1.65e^{-1}$ | $1.93e^{-1}$ | $1.85e^{-1}$ | $1.83e^{-1}$ | $1.85e^{-1}$ | $1.87e^{-1}$ | $1.77e^{-1}$ | $1.82e^{-1}$ | $1.88e^{-1}$ | $1.82e^{-1}$ |

of the model; see discussion in §D.5. We report the approximate generalizability of all fine-tuned models in Tables 3; also, see 15 in §D.5.

Drawing a connection from our theoretical upper bounds in Table 1, we find PEFT methods with the same upper bounds perform similarly in practice. More precisely, cLA has a smaller upper bound on $\mathcal{G}(\mathbf{W})$ than r-$c^3$LA in practice, indicating the validity of theoretical upper bounds. This observation also holds for cLA and RAC, and $c^3$LA and Asymmetric LoRA pairs. On the other hand, cLA and r-cLA have the same upper bound on $\mathcal{G}(\mathbf{W})$, and they also perform almost similarly in practice. Nevertheless, there are some discrepancies, and we attribute them to the fact that Table 1 gives us an upper bound on $\mathcal{G}(\mathbf{W})$. E.g., although the upper bound on $\mathcal{G}(\mathbf{W})$ of Asymmetric LoRA is smaller than RAC by a factor of $\sqrt{k}$, they behave similarly in practice. Similarly, r-cLA performs marginally worse than RAC, although RAC has a higher theoretical bound on $\mathcal{G}(\mathbf{W})$. In an extreme case, r-$c^3$LA empirically outperforms r-cLA while having a higher theoretical bound on $\mathcal{G}(\mathbf{W})$.

## 5 CONCLUSION

Through extensive benchmarking spanning four different fine-tuning tasks, nine models, and fourteen datasets, we show that no fine-tuning method, including full fine-tuning, is a clear choice for fine-tuning an arbitrary task. This observation confirms the finding in some previous works that dissect LoRA's efficacy. As the future of computing and hardware interfaces moves towards memory- and compute-efficiency, we propose simple LoRA variants with inherent sparsity, cLA and $c^3$LA and their randomized variants, and observe their surprisingly good performance. Therefore, we postulate that it is advantageous to choose a fine-tuning method based on its characteristics and user-specific needs rather than on generated accuracy. To support this, we analyzed our methods and various common LoRA PEFT variants through the lens of generalizability. To our knowledge, we are among the first to obtain generalization error bounds for a wide range of PEFT methods. We show that, in theory, our methods have the same generalization error upper bounds as their non-sparse counterparts. While comparing the theoretical results of generalization error bounds with experimentally observed generalization error, we find that our generalization error upper bounds closely follow the generalizability of the models in practice, among all other experimental perspectives, such as loss-landscape and intruder dimensions. In the advent of artificial general intelligence, when we want a model to behave human-like across many tasks, it is better to choose PEFT methods that generalize well and are computationally efficient.

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

CONTENTS

**E  Limitations and Discussion**                                                                                    **41**

**F  Table of notations**                                                                                            **41**

**Organization of Appendix.** We organize the Appendix with the following structure:In §A, we discuss the popular contemporary LoRA variants; this is a continuation of §2.2 of the main paper. In §B, we give the pseudocode of our proposed LoRA variants, cLA, random-cLA, and $c^3$LA. §C contains the proofs to the theorems in §3. Particularly, it contains the proofs for Theorem 2, Theorem 1, Theorem 4 and Theorem 6. In §D, we discuss the implementation details and extend our empirical study by including various ablation studies and developing discussion topics. This section acts as an addendum to §4 of the main paper. For notations used in this paper, we refer to Table 16 in §F.

## A  THE PRESENT: EVOLUTION OF LORA—CONTINUED

Below, we sketch a few popular LoRA variants.

**Chain of LoRA (CoLA) (57)** increases LoRA's performance without substantially increasing compute or memory costs. After fine-tuning $\mathbf{B}^1\mathbf{A}^1$ for the downstream task to obtain $\hat{\mathbf{B}}^1\hat{\mathbf{A}}^1$, CoLA merges $\hat{\mathbf{B}}^1\hat{\mathbf{A}}^1$ into the base weights and continues training with a new $\mathbf{B}^2\mathbf{A}^2$ on the same task, treating $\mathbf{W}_0 + \frac{\alpha}{r}\hat{\mathbf{B}}^1\hat{\mathbf{A}}^1$ as the base weights. Denote $\mathbf{W}^{(k,BA)} := \mathbf{W}_0 + \sum_{j=1}^{k}\frac{\alpha}{r}\hat{\mathbf{B}}^j\hat{\mathbf{A}}^j$ and $\mathbf{W}^{(0,BA)} = \mathbf{W}_0$ for convenience. CoLA of chain length $k$ solves:

$$\text{For } j \in [k], \quad \hat{\mathbf{B}}^j\hat{\mathbf{A}}^j \approx \text{argmin}_{\mathbf{B}^j\mathbf{A}^j}\left[\mathcal{L}(\mathbf{W}_0^{(j-1,BA)} + \frac{\alpha}{r}\hat{\mathbf{B}}^j\hat{\mathbf{A}}^j)\right] \tag{7}$$

to obtain the fine-tuned model, $f_{\mathbf{W}^{(k,BA)}}$. CoLA simulates a higher-rank approximation of a single LoRA update (32) and claims to reduce LoRA's failure (25).

**Asymmetric LoRA (65)** modifies LoRA adaptation for each layer by freezing one of the low-rank matrices, conventionally, $A$ to $A_0$, initializing the frozen matrix via a Normal distribution, and setting the trainable matrix to 0, and solves:

$$\hat{\mathbf{B}} \approx \text{argmin}_{\mathbf{B}}[\mathcal{L}(\mathbf{W}_0 + \frac{\alpha}{r}\mathbf{B}\mathbf{A}_0) = \frac{1}{|N|}\sum_{i=1}^{|N|}\ell(f_{\mathbf{W}_0 + \frac{\alpha}{r}\mathbf{B}\mathbf{A}_0}(x_i), y_i)], \tag{8}$$

to obtain the fine-tuned model $f_{\mathbf{W}_0 + \hat{\mathbf{B}}\mathbf{A}_0}$. Under trainable-parameter constraints, Asymmetric LoRA competes with LoRA (65) and retains the Lipschitz smoothness of the loss function, which LoRA does not (51).

**Randomized Asymmetric Chain of LoRA (RAC-LoRA) (38)** combines Asymmetric LoRA and CoLA. RAC-LoRA fixes one of the low-rank matrices (conventionally $A$), initializing via some fixed distribution of matrices $\mathcal{D}$, and sets the trainable one to 0. Like CoLA, the trained $\hat{\mathbf{B}}^1\mathbf{A}_0^1$ is then merged into the base weights, and a new $\mathbf{B}\mathbf{A}_0$ is trained on the same task. Denote $\mathbf{W}^{(k,B)} := \mathbf{W}_0 + \sum_{j=1}^{k}\frac{\alpha}{r}\hat{\mathbf{B}}^j\mathbf{A}_0^j$ and $\mathbf{W}^{(0,B)} = \mathbf{W}_0$. RAC-LoRA of chain length $k$ solves:

$$\text{For } j \in [k], \quad \hat{\mathbf{B}}^j \approx \text{argmin}_{\mathbf{B}^j}\left[\mathcal{L}(\mathbf{W}_0^{(j-1,B)} + \frac{\alpha}{r}\hat{\mathbf{B}}^j\mathbf{A}_0^j)\right] \tag{9}$$

to obtain the fine-tuned model $f_{\mathbf{W}^{(k,B)}}$.

**LoRA+ (19)** applies separate learning rates $\{\gamma_B^i, \gamma_A^i\}$ to the adapter matrices, $\{B^i, A^i\}$ of each layer, respectively, and maintains the identical structure to LoRA. LoRA+ prioritizes a substantially higher learning rate $(2 - 16\times)$ for $B$.

**Other variants.** There are other popular LoRA variants, such as HydraLoRA (53), designed for fine-tuning on datasets with high heterogeneity. LoRA-SB (42) simulates the FFT process within low-rank subspaces by adding a trainable $r \times r$ matrix $R$, initializing $BRA$ based on the $SVD$ of the first step of FFT, and freezing $B, A$. QLoRA (8) fine-tunes quantized LLMs. AdaLoRA (62) uses varying rank by layer and uses an SVD initialization. SoRA (10) introduces sparsity in the low-rank updates. DoRA (34) separates fine-tuning the direction and magnitude components of the model.

AutoLoRA (63) trains each LoRA update as a sum of rank-one matrices and learns which to discard during training. DyLoRA (54) concentrates the more important features in the first columns and rows of $B$ and $A$, respectively.

## B  PSEUDO CODE OF OUR PROPOSED LoRA VARIANTS

In this Section, we present the pseudocode of our proposed LoRA variants, cLA (Algorithm 1), random-cLA (Algorithm 2), $c^3$LA (Algorithm 3) and r-$c^3$LA (Algorithm 4).

---

**Algorithm 1 Cheap LoRA (cLA)**

---

1: **Parameters:** Loss function $\mathcal{L}$ and model $f_{\mathbf{W}}(\cdot)$. Pretrained weights $\mathbf{W}_0 = (W_0^1, ..., W_0^L)$, where $W_0^i \in \mathbb{R}^{n_i \times m_i}$. rank $r \ll \min\{m_i, n_i\}_{i \in [L]}$, learning rate $\gamma > 0$, scaling factor $\alpha > 0$, total training iterations $T$.

2: **Initialize** $A_0^j = [I_r \mid \mathbf{0}_{r \times (m_j - r)}]$; $B^{0,j} = \mathbf{0}$ for $j \in [L]$

3: **for** $t = 1, ..., T$ **do**

4:     forward pass with LoRA modules

5:     backward pass then update $\mathbf{B}^t$

6:     **for** $j = 1, ..., L$ **do**

7:         $B^{t,j} = B^{t-1,j} - \gamma \frac{\alpha}{r} \nabla_j \mathcal{L}(\mathbf{W}_0 + \frac{\alpha}{r} \mathbf{B}^{t-1} \mathbf{A}_0) \operatorname{Diag}(\overbrace{1, ..., 1}^{1 \text{ to } r}, 0, ..., 0)$

8:     **end for**

9: **end for**

10: $\hat{j} = \operatorname{argmin}_{j \in [T]} \mathcal{L}(\mathbf{W}_0 + \frac{\alpha}{r} \mathbf{B}^j \mathbf{A}_0)$ or task-based metric.

11: **return** Fine-tuned weights $\mathbf{W}_0 + \frac{\alpha}{r} \mathbf{B}^{\hat{j}} \mathbf{A}_0$

---

---

**Algorithm 2 random Cheap LoRA (r-cLA)**

---

1: **Parameters:** Loss function $\mathcal{L}$ and model $f_{\mathbf{W}}(\cdot)$. Pretrained weights $\mathbf{W}_0 = (W_0^1, ..., W_0^L)$, where $W_0^i \in \mathbb{R}^{m_i \times n_i}$. rank $r \ll \min\{m_i, n_i\}_{i \in [L]}$, learning rate $\gamma > 0$, scaling factor $\alpha > 0$, total training iterations $T$.

2: **Initialize**

3:     $\xi_j = \operatorname{randint}(0, \lfloor \frac{n_j}{r} \rfloor - 1)$ for $j \in [L]$

4:     $A_0^j = \left[ \mathbf{0}_{r \times \xi_j} \mid I_r \mid \mathbf{0}_{r \times (n_j - \xi_j - r)} \right]$; $B^{0,j} = \mathbf{0}$ for $j \in [L]$

5: **for** $t = 1, ..., T$ **do**

6:     forward pass with LoRA modules

7:     backward pass then update $\mathbf{B}^t$

8:     **for** $j = 1, ..., L$ **do**

9:         $B^{t,j} = B^{t-1,j} - \gamma \frac{\alpha}{r} \nabla_j \mathcal{L}(\mathbf{W}_0 + \frac{\alpha}{r} \mathbf{B}^{t-1} \mathbf{A}_0) \operatorname{Diag}(0, ..., 0, \overbrace{1, ..., 1}^{\xi_j + 1 \text{ to } \xi_j + r}, 0, ..., 0)$

10:     **end for**

11: **end for**

12: $\hat{j} = \operatorname{argmin}_{j \in [T]} \mathcal{L}(\mathbf{W}_0 + \frac{\alpha}{r} \mathbf{B}^j \mathbf{A}_0)$ or task-based metric.

13: **return** Fine-tuned weights $\mathbf{W}_0 + \frac{\alpha}{r} \mathbf{B}^{\hat{j}} \mathbf{A}_0$

---

## C  THEORETICAL RESULTS

This section complements Section 3 in the main paper.

### C.1  GENERALIZATION

In this section, we give a detailed proof of the generalization error bound. We start by listing the inequalities used in this section.

---

**Algorithm 3** Circulant Chain of Cheap LoRA ($c^3$LA)

1: **Parameters:** Loss function $\mathcal{L}$ and model $f_{\mathbf{W}}(\cdot)$. Pretrained weights $\mathbf{W}_0^{(0)} = (W^1, ..., W^L)$, where $W^i \in \mathbb{R}^{m_i \times n_i}$. rank $r \ll \min\{m_i, n_i\}_{i \in [L]}$, learning rate $\gamma > 0$, scaling factor $\alpha > 0$, total training iterations $T$, chain-length $k <= T$.

2: **Initialize** $A_0^j = [I_r \mid \mathbf{0}_{r \times (n_j - r)}]$; $B^{0,j} = \mathbf{0}$ for $j \in [L]$, current chain $c = 0$.

3: **for** $t = 1, ..., T$ **do**

4:      **if** $t \equiv 0 \pmod{\lfloor \frac{T}{k} \rfloor}$ **then**

5:          $c = c + 1$

6:          Merge LoRA to backbone weights $\mathbf{W}_0^{(c)} = \mathbf{W}_0^{(c-1)} + \frac{\alpha}{r} \mathbf{B}^{t-1} \mathbf{A}_0$

7:          Re-initialize with $\mathbf{A}_0$ shifted by $r$:

8:             $A_0^j = [\mathbf{0}_{r \times cr} \mid I_r \mid \mathbf{0}_{r \times n_i - r - cr}]$; $B^{t-1,j} = \mathbf{0}$ for $j \in [L]$

9:      **end if**

10:     forward pass with LoRA modules

11:     backward pass then update $\mathbf{B}^t$

12:     **for** $j = 1, ..., L$ **do**

13:          $B^{t,j} = B^{t-1,j} - \gamma \frac{\alpha}{r} \nabla_j \mathcal{L}(\mathbf{W}_0^{(c)} + \frac{\alpha}{r} \mathbf{B}^{t-1} \mathbf{A}_0) \operatorname{Diag}(0, ..., 0, \overbrace{1, ..., 1}^{cr \text{ to } (c+1)_r}, 0, ..., 0)$

14:     **end for**

15: **end for**

16: $\hat{c}, \hat{j} = \operatorname{argmin}_{j \in [\lfloor \frac{T}{k} \rfloor], c \in [k]} \mathcal{L}(\mathbf{W}_0^{(c)} + \frac{\alpha}{r} \mathbf{B}^{cj} \mathbf{A}_0)$ or task-based metric.

17: **return** Fine-tuned weights $\mathbf{W}_0^{\hat{c}} + \frac{\alpha}{r} \mathbf{B}^{\hat{c}\hat{j}} \mathbf{A}_0$

---

---

**Algorithm 4** Random Circulant Chain of Cheap LoRA (r-$c^3$LA)

1: **Parameters:** Loss function $\mathcal{L}$ and model $f_{\mathbf{W}}(\cdot)$. Pretrained weights $\mathbf{W}_0^{(0)} = (W^1, ..., W^L)$, where $W^i \in \mathbb{R}^{m_i \times n_i}$. rank $r \ll \min\{m_i, n_i\}_{i \in [L]}$, learning rate $\gamma > 0$, scaling factor $\alpha > 0$, total training iterations $T$, chain-length $k <= T$.

2: **Initialize**

3: $\xi_j = \operatorname{randint}(0, \lfloor \frac{n_j}{r} \rfloor - 1)$ for $j \in [L]$.

4: $A_0^j = \left[ \mathbf{0}_{r \times \xi_j} \mid I_r \mid \mathbf{0}_{r \times (n_j - \xi_j - r)} \right]$; $B^{0,j} = \mathbf{0}$ for $j \in [L]$, current chain $c = 0$.

5: **for** $t = 1, ..., T$ **do**

6:      **if** $t \equiv 0 \pmod{\lfloor \frac{T}{k} \rfloor}$ **then**

7:          $c = c + 1$

8:          Merge LoRA to backbone weights $\mathbf{W}_0^{(c)} = \mathbf{W}_0^{(c-1)} + \frac{\alpha}{r} \mathbf{B}^{t-1} \mathbf{A}_0$

9:          Re-initialize with $\mathbf{A}_0$ shifted by a new random variable $\xi_j'$:

10:         $A_0^j = \left[ \mathbf{0}_{r \times \xi_j'} \mid I_r \mid \mathbf{0}_{r \times (n_j - \xi_j' - r)} \right]$; $B^{t-1,j} = \mathbf{0}$ for $j \in [L]$

11:      **end if**

12:     forward pass with LoRA modules

13:     backward pass then update $\mathbf{B}^t$

14:     **for** $j = 1, ..., L$ **do**

15:          $B^{t,j} = B^{t-1,j} - \gamma \frac{\alpha}{r} \nabla_j \mathcal{L}(\mathbf{W}_0 + \frac{\alpha}{r} \mathbf{B}^{t-1} \mathbf{A}_0) \operatorname{Diag}(0, ..., 0, \overbrace{1, ..., 1}^{\xi_j + 1 \text{ to } \xi_j + r}, 0, ..., 0)$

16:     **end for**

17: **end for**

18: $\hat{c}, \hat{j} = \operatorname{argmin}_{j \in [\lfloor \frac{T}{k} \rfloor], c \in [k]} \mathcal{L}(\mathbf{W}_0^{(c)} + \frac{\alpha}{r} \mathbf{B}^{cj} \mathbf{A}_0)$ or task-based metric.

19: **return** Fine-tuned weights $\mathbf{W}_0^{\hat{c}} + \frac{\alpha}{r} \mathbf{B}^{\hat{c}\hat{j}} \mathbf{A}_0$

---

### C.1.1 INEQUALITIES USED

1. If $A, B \in \mathbb{R}^{m \times n}$ and $x \in \mathbb{R}^n$, then the Triangle-Inequality gives:
$$\|(A + B)x\| \le \|Ax\| + \|Bx\|. \tag{10}$$

2. For $A \in \mathbb{R}^{m \times n}$ and $x \in \mathbb{R}^n$, we have:
$$\|Ax\| \le \|A\|_2 \|x\|. \tag{11}$$

3. If $\sigma(0) \ne 0$, then by the triangle inequality and using Assumption 3 we have:
$$\|\sigma(Ax)\| \le \|\sigma(Ax) - \sigma(0)\| + \|\sigma(0)\| \le L_\sigma \|Ax\| + \|\sigma(0)\|. \tag{12}$$

4. For a finite collection of matrices, $\{A_1, \cdots, A_k\}$; $A_i \in \mathbb{R}^{m \times n}$, we have:
$$\text{rank}(\sum_{i=1}^{k} A_i) \le \sum_{i=1}^{k} \text{rank}(A_i). \tag{13}$$

5. Let $\mathbf{I}(X; Y)$ denote the mutual information between random variables $X$ and $Y$. It measures how much the knowledge of one random variable reveals about measuring the other, i.e.,
$$\mathbf{I}(X; Y) = D(P_{XY} \| P_X \otimes P_Y) = \sup_F \{ \int F dP_{XY} - \log \int e^F d(P_X \otimes P_Y) \},$$
where $F$ is a bounded, measurable function (58). Let $T$ be a deterministic map for $A \in \mathbb{R}^{m \times n}$. Then the **Data Processing Inequality (DPI)** gives us $\mathbf{I}(T(A); N) \le \mathbf{I}(A; N)$. If $T$ is a bijective mapping then **(DPI)** gives us (65):
$$\mathbf{I}(A; N) = \mathbf{I}(T(A); N). \tag{14}$$

### C.1.2 PROOF OF THEOREM 1.

**Theorem 1.** *(Generalization bounds) Let $f_{\mathbf{W}_0 + \Delta \mathbf{w}}(x) = \sigma_L([W_0{}^L + \Delta W^L](\cdots \sigma_2([(W_0^2 + \Delta W^2]\sigma_1([W_0^1 + \Delta W^1]x))\cdots))$ be a L-layers fine-tuned DNN, where $\mathbf{W}_0 + \Delta \mathbf{W}$ is a fine-tuned update. Let the loss function, $\mathcal{L}$ for fine-tuning, follow Assumption 2 and Assumptions 1–3 hold. Then $\mathcal{G}(\mathbf{W}_0 + \Delta \mathbf{W}) \le \min(\mathcal{G}(\mathbf{W}_0) + \Phi_{\Delta \mathbf{W}}, \mathcal{G}(\Delta \mathbf{W}) + \Phi_{\mathbf{W}_0})$, where*

$$\Phi_{\Delta \mathbf{w}} := 2L_{\mathcal{L}} \left[ C \prod_{i=1}^{L} L_{\sigma_i} \sum_{i=1}^{2^L - 1} \prod_{j=1}^{L} P(i, j) + \sum_{i \ne 2^a - 1 : a \in [L]}^{2^L - 2} F(i) \right] \text{ and}$$

$$\Phi_{\mathbf{W}_0} := 2L_{\mathcal{L}} \left[ C \prod_{i=1}^{L} L_{\sigma_i} \sum_{i=2}^{2^L} \prod_{j=1}^{L} P(i, j) + \sum_{i \ne 2^a : a \in [L]}^{2^L - 1} F(i) \right],$$

*are the correction terms, $F(i) := \|\sigma_{L - \psi(i)}(0)\| \prod_{j=1}^{\psi(i)} [L_{\sigma_{L-j+1}} H(i, j)]$, $\psi(i) := \lfloor \log_2(i) \rfloor$, and*

$$P(i, j) := \begin{cases} \|W_0^{(L-j+1)}\| \text{ if } \lfloor \frac{i-1}{2^{L-1}} \rfloor \text{ is odd,} \\ \|\Delta W^{(L-j+1)}\| \text{ if } \lfloor \frac{i-1}{2^{L-1}} \rfloor \text{ is even} \end{cases} \quad H(i, j) := \begin{cases} \|\Delta W^{(L-j+1)}\| \text{ if } \lfloor \frac{i}{2^{\psi(i)-j}} \rfloor \text{ is odd,} \\ \|W_0^{(L-j+1)}\| \text{ if } \lfloor \frac{i}{2^{\psi(i)-j}} \rfloor \text{ is even.} \end{cases}$$

*Proof.* Let
$$f_{\mathbf{W}_0 + \Delta \mathbf{w}} := \sigma_{(L)}([W_0^{(L)} + \Delta W^{(L)}] \sigma_{(L-1)}(\dots \sigma_{(1)}([W_0^{(1)} + \Delta W^{(1)}]x)\dots))$$
represent our fine-tuned model and
$$f_{\mathbf{W}_0} := \sigma_{(L)}(W_0^{(L)} \sigma_{(L-1)}(\dots \sigma_{(1)}(W_0^{(1)} x)\dots))$$

represent our pretrained model. First, we upper bound the quantity $\|f_{\mathbf{W}_0+\Delta\mathbf{W}} - f_{\mathbf{W}_0}\|$. We have

$$\|f_{\mathbf{W}_0+\Delta\mathbf{W}} - f_{\mathbf{W}_0}\|$$

$$= \left\|\sigma_{(L)}([W_0^{(L)} + \Delta W^{(L)}]\,\sigma_{(L-1)}(\cdots\sigma_{(1)}([W_0^{(1)} + \Delta W^{(1)}]x)\cdots))\right.$$

$$\left. - \sigma_{(L)}(W_0^{(L)}\,\sigma_{(L-1)}(\cdots\sigma_{(1)}(W_0^{(1)}x)\cdots))\right\|$$

$$\overset{\text{Assumption 3}}{\leq}\; L_{\sigma_L}\left\|[W_0^{(L)} + \Delta W^{(L)}]\sigma_{(L-1)}(\cdots\sigma_{(1)}([W_0^{(1)} + \Delta W^{(1)}]x)\cdots))\right.$$

$$\left. - [W_0^{(L)}]\sigma_{(L-1)}(\cdots\sigma_{(1)}(W_0^{(1)}x)\cdots))\right\|$$

$$= L_{\sigma_L}\left\|\Delta W^{(L)}\sigma_{(L-1)}(\cdots\sigma_{(1)}([W_0^{(1)} + \Delta W^{(1)}]x)\cdots))\right.$$

$$\left. - W_0^{(L)}[(\sigma_{(L-1)}(\cdots\sigma_{(1)}([W_0^{(1)} + \Delta W^{(1)}]x)\cdots)) - \sigma_{(L-1)}(\cdots\sigma_{(1)}(W_0^{(1)}x)\cdots)))]\right\|$$

$$\overset{\text{Triangle Inequality and Inequality (11)}}{\leq}\; L_{\sigma_L}[\|\Delta W^{(L)}\|_2\|\sigma_{(L-1)}(\cdots\sigma_{(1)}([W_0^{(1)} + \Delta W^{(1)}]x)\cdots))\|$$

$$+ \|W_0^{(L)}\|_2\|(\sigma_{(L-1)}(\cdots\sigma_{(1)}([W_0^{(1)} + \Delta W^{(1)}]x)\cdots) - \sigma_{(L-1)}(\cdots\sigma_{(1)}(W_0^{(1)}x)\cdots))\|].$$

Note that our inequality is now composed of two components:

(A): $L_{\sigma_L}\|\Delta W^{(L)}\|_2\|\sigma_{(L-1)}(\cdots\sigma_{(1)}([W_0^{(1)} + \Delta W^{(1)}]x)\cdots)\|$

(B): $L_{\sigma_L}\|W_0^{(L)}\|_2\|(\sigma_{(L-1)}(\cdots\sigma_{(1)}([W_0^{(1)} + \Delta W^{(1)}]x)\cdots) - \sigma_{(L-1)}(\cdots\sigma_{(1)}(W_0^{(1)}x)\cdots))\|.$

We will show for any $k > 1$ that the (B) component can expand out to two sub-components that mimic (A) and (B).

$$(\text{B})_k := L_{\sigma_k}\|W_0^{(k)}\|_2\left\|\sigma_{(k-1)}(\cdots\sigma_{(1)}([W_0^{(1)} + \Delta W^{(1)}]x)\cdots) - \sigma_{(k-1)}(\cdots\sigma_{(1)}(W_0^{(1)}x)\cdots)\right\|$$

$$\overset{\text{Assumption 3}}{\leq}\; L_{\sigma_k}\|W_0^{(k)}\|_2\,L_{\sigma_{k-1}}\|[W_0^{(k-1)} + \Delta W^{(k-1)}]\,\sigma_{(k-2)}(\cdots\sigma_{(1)}([W_0^{(1)} + \Delta W^{(1)}]x)\cdots)$$

$$- W_0^{(k-1)}\,\sigma_{(k-2)}(\cdots\sigma_{(1)}(W_0^{(1)}x)\cdots)\|$$

$$= L_{\sigma_k}L_{\sigma_{k-1}}\|W_0^{(k)}\|_2\left\|\Delta W^{(k-1)}\,\sigma_{(k-2)}(\cdots\sigma_{(1)}([W_0^{(1)} + \Delta W^{(1)}]x)\cdots)\right.$$

$$\left. + W_0^{(k-1)}\Big(\sigma_{(k-2)}(\cdots\sigma_{(1)}([W_0^{(1)} + \Delta W^{(1)}]x)\cdots) - \sigma_{(k-2)}(\cdots\sigma_{(1)}(W_0^{(1)}x)\cdots)\Big)\right\|$$

$$\overset{\text{Inequality (11)}}{\leq}\; L_{\sigma_k}L_{\sigma_{k-1}}\|W_0^{(k)}\|_2\left(\|\Delta W^{(k-1)}\|_2\,\|\sigma_{(k-2)}(\cdots\sigma_{(1)}([W_0^{(1)} + \Delta W^{(1)}]x)\cdots)\|\right.$$

$$\left. + \|W_0^{(k-1)}\|_2\,\|\sigma_{(k-2)}(\cdots\sigma_{(1)}([W_0^{(1)} + \Delta W^{(1)}]x)\cdots) - \sigma_{(k-2)}(\cdots\sigma_{(1)}(W_0^{(1)}x)\cdots)\|\right).$$

$$(\text{B})_k \leq L_{\sigma_k}\|W_0^{(k)}\|_2\,\underbrace{L_{\sigma_{k-1}}\|\Delta W^{(k-1)}\|_2\,\|\sigma_{(k-2)}(\cdots\sigma_{(1)}([W_0^{(1)} + \Delta W^{(1)}]x)\cdots)\|}_{(\text{A})_{k-1}}$$

$$+ L_{\sigma_k}\|W_0^{(k)}\|_2\,\underbrace{L_{\sigma_{k-1}}\|W_0^{(k-1)}\|_2\,\|\sigma_{(k-2)}(\cdots\sigma_{(1)}([W_0^{(1)} + \Delta W^{(1)}]x)\cdots) - \sigma_{(k-2)}(\cdots\sigma_{(1)}(W_0^{(1)}x)\cdots)\|}_{(\text{B})_{k-1}}.$$

When $k = 1$, we see that (B) no longer splits into (A) and (B) subcomponents, but rather:

$$(B)_1 := \Big( \prod_{j=2}^{L} L_{\sigma_j} \Big) \Big( \prod_{i=2}^{L} \|W_0^{(i)}\|_2 \Big) \Big\| \sigma_{(1)}([W_0^{(1)} + \Delta W^{(1)}]x) - \sigma_{(1)}(W_0^{(1)}x) \Big\|$$

$$\overset{\text{Assumption 3}}{\leq} \Big( \prod_{j=1}^{L} L_{\sigma_j} \Big) \Big( \prod_{i=2}^{L} \|W_0^{(i)}\|_2 \Big) \|[W_0^{(1)} + \Delta W^{(1)}]x - W_0^{(1)}x\|$$

$$= \Big( \prod_{j=1}^{L} L_{\sigma_j} \Big) \Big( \prod_{i=2}^{L} \|W_0^{(i)}\|_2 \Big) \|\Delta W^{(1)}x\|$$

$$\overset{\text{Inequality (11)}}{\leq} \Big( \prod_{j=1}^{L} L_{\sigma_j} \Big) \Big( \prod_{i=2}^{L} \|W_0^{(i)}\|_2 \Big) \|\Delta W^{(1)}\|_2 \, \|x\|$$

$$\overset{\text{Assumption 1}}{\leq} C \Big( \prod_{j=1}^{L} L_{\sigma_j} \Big) \Big( \prod_{i=2}^{L} \|W_0^{(i)}\|_2 \Big) \|\Delta W^{(1)}\|_2.$$

We now present the recursive step for the (A) terms:

$$(A)_k := L_{\sigma_k} \|\Delta W^{(k)}\|_2 \Big\| \sigma_{(k-1)} \Big( [W_0^{(k-1)} + \Delta W^{(k-1)}] \, \sigma_{(k-2)} \big( \cdots \sigma_{(1)}([W_0^{(1)} + \Delta W^{(1)}]x) \big) \Big) \Big\|$$

$$\overset{\text{Inequality (12)}}{\leq} L_{\sigma_k} \|\Delta W^{(k)}\|_2 \Big( L_{\sigma_{k-1}} \big\| [W_0^{(k-1)} + \Delta W^{(k-1)}] \, \sigma_{(1)}([W_0^{(1)} + \Delta W^{(1)}]x) \big\|$$

$$+ \|\sigma_{(k-1)}(0)\| \Big)$$

$$\overset{\text{Inequality (11)}}{\leq} L_{\sigma_k} \|\Delta W^{(k)}\|_2 \Big( L_{\sigma_{k-1}} \|W_0^{(k-1)}\|_2 \big\| \sigma_{(1)}([W_0^{(1)} + \Delta W^{(1)}]x) \big\|$$

$$+ L_{\sigma_{k-1}} \|\Delta W^{(k-1)}\|_2 \big\| \sigma_{(1)}([W_0^{(1)} + \Delta W^{(1)}]x) \big\| + \|\sigma_{(k-1)}(0)\| \Big).$$

**Recursive collapse of (A).** Applying Inequality (12) to the outer activation $\sigma_{(k-1)}$ splits $(A)_k$ into a $W_0$-branch, a $\Delta W$-branch, and an offset term $\sigma_{(k-1)}(0)$. The $W_0$ and $\Delta W$ branches recurse inward, each step multiplying by $L_{\sigma_i}$ and either $\|W_0^{(i)}\|_2$ or $\|\Delta W^{(i)}\|_2$, until $\sigma_{(1)}$ returns $L_{\sigma_1}\|W_0^{(1)}x\|$, $L_{\sigma_1}\|\Delta W^{(1)}x\|$, and $\|\sigma_{(1)}(0)\|$, which reduce via property (i). Hence $(A)_k$ collapses to a sum of terms based on permutations of matrix spectral norms, plus offset contributions from the nonlinear activation of zero vectors.

If $f_{\mathbf{W}_0}$ and $f_{\mathbf{W}_0 + \Delta \mathbf{W}}$ are both 1-layer, we can expand out their difference by:

$$\|f_{\mathbf{W}_0 + \Delta \mathbf{W}} - f_{\mathbf{W}_0}\| \leq C L_{\sigma_1} \|\Delta W^{(1)}\|_2$$

If $f_{\mathbf{W}_0}$ and $f_{\mathbf{W}_0 + \Delta \mathbf{W}}$ are both 2-layer, we can expand out their difference by:

$$\|f_{\mathbf{W}_0 + \Delta \mathbf{W}} - f_{\mathbf{W}_0}\| \leq C \, L_{\sigma_2} L_{\sigma_1} \|W_0^{(2)}\|_2 \|\Delta W^{(1)}\|_2 + \, C L_{\sigma_2} L_{\sigma_1} \|\Delta W^{(2)}\|_2 \|W_0^{(1)}\|_2$$

$$+ C L_{\sigma_2} L_{\sigma_2} \|\Delta W^{(2)}\|_2 \|\Delta W^{(1)}\|_2 + L_{\sigma_2} \|\Delta W^{(2)}\|_2 \|\sigma_1(0)\|$$

If $f_{\mathbf{W}_0}$ and $f_{\mathbf{W}_0 + \Delta \mathbf{W}}$ are both 3-layer, we can expand out their difference by:

$$\|f_{\mathbf{W}_0 + \Delta \mathbf{W}} - f_{\mathbf{W}_0}\| \leq C \, L_{\sigma_1} L_{\sigma_2} L_{\sigma_3} \Big( \|W_0^{(3)}\|_2 \|W_0^{(2)}\|_2 \|\Delta W^{(1)}\|_2$$

$$+ \|W_0^{(3)}\|_2 \|\Delta W^{(2)}\|_2 \|W_0^{(1)}\|_2 + \|W_0^{(3)}\|_2 \|\Delta W^{(2)}\|_2 \|\Delta W^{(1)}\|_2$$

$$+ \|\Delta W^{(3)}\|_2 \|W_0^{(2)}\|_2 \|W_0^{(1)}\|_2 + \|\Delta W^{(3)}\|_2 \|W_0^{(2)}\|_2 \|\Delta W^{(1)}\|_2$$

$$+ \|\Delta W^{(3)}\|_2 \|\Delta W^{(2)}\|_2 \|W_0^{(1)}\|_2 + \|\Delta W^{(3)}\|_2 \|\Delta W^{(2)}\|_2 \|\Delta W^{(1)}\|_2 \Big)$$

$$+ \, L_{\sigma_3} L_{\sigma_2} \|\sigma_{(1)}(0)\| \, \Big( \|W_0^{(3)}\|_2 \|\Delta W^{(2)}\|_2 + \, \|\Delta W^{(3)}\|_2 \|W_0^{(2)}\|_2 \Big)$$

$$+ \, L_{\sigma_3} L_{\sigma_2} \|\Delta W^{(3)}\|_2 \|\Delta W^{(2)}\|_2 \|\sigma_{(1)}(0)\| + \, L_{\sigma_3} \|\Delta W^{(3)}\|_2 \|\sigma_{(2)}(0)\|.$$

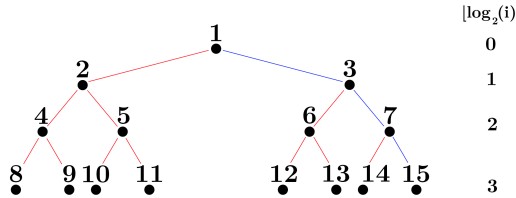

Figure 4: $\|f_{\mathbf{W}_0+\Delta\mathbf{w}} - f_{\mathbf{W}_0}\|$ Visual representation of the recursive collapse of $(\mathbf{A}, \mathbf{B})$.

Thus a proof by induction indicates the difference between $f_{\mathbf{W}_0}$ and $f_{\mathbf{W}_0+\Delta\mathbf{w}}$ L-layered hypothesis can be upper bounded by:

$$\|f_{\mathbf{W}_0+\Delta\mathbf{w}} - f_{\mathbf{W}}\| \leq C \prod_{i=1}^{L} L_{\sigma_i} \Big[ \sum_{i=1}^{2^L-1} \prod_{j=1}^{L} P_L(i,j) \Big] + \sum_{i=2; i \neq 2^a-1, a \in [L]}^{2^L-2} F(i)$$

If we treat $\Delta W^{(i)}$ and $W_0^{(i)}$ as binary classes, we can give each identity 0 and 1 respectively; thus $W_0^{(3)} W_0^{(2)} W_0^{(1)}$ corresponds to $111_2$ or 7 and $\Delta W^{(3)} W_0^{(2)} \Delta W^{(1)}$ corresponds to $010_2$ or 2. Thus, using this pattern, we can expand our summation using the following expression:

$$P_L(i,j) = \begin{cases} \|W_0^{(L-j+1)}\|_2, & \text{if } \left\lfloor \dfrac{i-1}{2^{L-j}} \right\rfloor \bmod 2 = 1, \\ \|\Delta W^{(L-j+1)}\|_2, & \text{if } \left\lfloor \dfrac{i-1}{2^{L-j}} \right\rfloor \bmod 2 = 0. \end{cases},$$

$$F(i) = \|\sigma_{(L-\lfloor \log_2(i) \rfloor)}(0)\| \prod_{j=1}^{\lfloor \log_2(i) \rfloor} [L_{\sigma_{(L-j+1)}} H(i,j)]$$

and

$$H(i,j) = \begin{cases} \|\Delta W^{(L-j+1)}\|_2 & \text{if } \lfloor \frac{i}{2^{\lfloor \log_2(i) \rfloor - j}} \rfloor \bmod 2 = 1, \\ \|W_0^{(L-j+1)}\|_2 & \text{if } \lfloor \frac{i}{2^{\lfloor \log_2(i) \rfloor - j}} \rfloor \bmod 2 = 0. \end{cases},$$

where $F(i)$ and $H(i,j)$ are index functions that can be visualized in Figure 4. For representational purposes, every vertex that has three red edges adds the $\ell_2$ norm of the layer below its activation function on the zero vector. When a vertex has two different colored edges strictly below it, it collapses into an $A$ and $B$ sub-component. When this occurs, no additional offset term is added to our summation. A total of $2^L - (L+1)$ of these offset terms will be added. Both $P(i,j)$ and $H(i,j)$ can also take cases by even and odd inputs as their indexing requires modulus arithmetic over binary classifications ($\|W_0^{(i)}\|$ and $\|\Delta W^{(i)}\|$).

Now that we have an upper bound for the difference of our hypotheses, we write the difference in terms of true loss and empirical loss:

$$\mathcal{L}_{\text{global}}(\mathbf{W}_0 + \Delta\mathbf{W}) - \mathcal{L}_{\text{global}}(\mathbf{W}_0) = \mathbb{E}_{\mathcal{X}, \mathcal{Y} \sim \nu}[\ell(f_{\mathbf{W}_0+\Delta\mathbf{w}}(X), Y)] - \mathbb{E}_{\mathcal{X}, \mathcal{Y} \sim \nu}[\ell(f_{\mathbf{W}_0}(X), Y)]$$

$$= \mathbb{E}_{\mathcal{X}, \mathcal{Y} \sim \nu}[\ell(f_{\mathbf{W}_0+\Delta\mathbf{w}}(X), Y) - \ell(f_{\mathbf{W}_0}(X), Y)].$$

$$\leq \mathbb{E}_{\mathcal{X}, \mathcal{Y} \sim \nu}[L_{\mathcal{L}} \|f_{\mathbf{W}_0+\Delta\mathbf{w}}(X) - f_{\mathbf{W}_0}(X)\|]$$

$$\leq \mathbb{E}_{\mathcal{X}, \mathcal{Y} \sim \nu} \left[ L_{\mathcal{L}} \left( C \prod_{k=1}^{L} L_{\sigma_i} \left[ \sum_{i=1}^{2^L-1} \prod_{j=1}^{L} P_L(i,j) \right] + \sum_{i \neq 2^a-1}^{2^L-2} F(i) \right) \right]$$

$$= L_{\mathcal{L}} \left[ C \prod_{k=1}^{L} L_{\sigma_i} \left[ \sum_{i=1}^{2^L-1} \prod_{j=1}^{L} P_L(i,j) \right] + \sum_{i \neq 2^a-1}^{2^L-2} F(i) \right].$$

Similarly,

$$\mathcal{L}(\mathbf{W}_0 + \Delta\mathbf{W}) - \mathcal{L}(\mathbf{W}_0) = \frac{1}{L}\sum_{i'=1}^{L}\ell\big(f_{\mathbf{w}_0+\Delta\mathbf{w}}(x_i'), y_i'\big) - \frac{1}{L}\sum_{i'=1}^{L}\ell\big(f_{\mathbf{w}_0}(x_i'), y_i'\big)$$

$$= \frac{1}{L}\sum_{i'=1}^{L}\Big[\ell\big(f_{\mathbf{w}_0+\Delta\mathbf{w}}(x_i'), y_i'\big) - \ell\big(f_{\mathbf{w}_0}(x_i'), y_i'\big)\Big]$$

$$\leq \frac{1}{L}\sum_{i'=1}^{L} L_{\mathcal{L}}\,\big\|f_{\mathbf{w}_0+\Delta\mathbf{w}}(x_i') - f_{\mathbf{w}_0}(x_i')\big\|$$

$$\leq \frac{1}{L}\sum_{i'=1}^{L} L_{\mathcal{L}}\Big( C\prod_{k=1}^{L}L_{\sigma_k}\Big[\sum_{i=1}^{2^L-1}\prod_{j=1}^{L}P_L(i,j)\Big] + \sum_{i\neq 2^a-1}^{2^L-2}F(i)\Big)$$

$$= L_{\mathcal{L}}\Big( C\prod_{k=1}^{L}L_{\sigma_k}\Big[\sum_{i=1}^{2^L-1}\prod_{j=1}^{L}P_L(i,j)\Big] + \sum_{i\neq 2^a-1}^{2^L-2}F(i)\Big).$$

Using the triangle inequality, we reach:

$$\big|\mathcal{G}(\mathbf{W}_0+\Delta\mathbf{W}) - \mathcal{G}(\mathbf{W}_0)\big| = \big|\mathcal{L}_{\text{global}}(\mathbf{W}_0+\Delta\mathbf{W}) - \mathcal{L}(\mathbf{W}_0+\Delta\mathbf{W}) - \mathcal{L}_{\text{global}}(\mathbf{W}_0) + \mathcal{L}(\mathbf{W}_0)\big|$$

$$\leq \big|\mathcal{L}_{\text{global}}(\mathbf{W}_0+\Delta\mathbf{W}) - \mathcal{L}_{\text{global}}(\mathbf{W}_0)\big| + \big|\mathcal{L}(\mathbf{W}_0+\Delta\mathbf{W}) - \mathcal{L}(\mathbf{W}_0)\big|$$

$$\leq 2\,L_{\mathcal{L}}\Big( C\prod_{k=1}^{L}L_{\sigma_k}\Big[\sum_{i=1}^{2^L-1}\prod_{j=1}^{L}P_L(i,j)\Big] + \sum_{i\neq 2^a-1}^{2^L-2}F(i)\Big).$$

Finally, we obtain the inequality:

$$\mathcal{G}(\mathbf{W}_0+\Delta\mathbf{W}) \leq \mathcal{G}(\mathbf{W}_0) + 2\,L_{\mathcal{L}}\left( C\prod_{k=1}^{L}L_{\sigma_k}\Big[\sum_{i=1}^{2^L-1}\prod_{j=1}^{L}P_L(i,j)\Big] + \sum_{i=1;i\neq 2^a-1;a\in[L]}^{2^L-2}F(i)\right).$$

**Bound around $f_{\Delta\mathbf{W}}$.** We can also perturb around $\mathcal{G}(\Delta\mathbf{W})$ by swapping the roles or conditions of $W_0^{(i)}$ and $\Delta W^{(i)}$ in the zero–activation bookkeeping function $H(i,j)$. This requires us to ignore the indices $2^a, a\in[L]$ as opposed to $2^a-1, a\in[L]$ as viewable in Figure 4. Similarly, the function $P_L(\cdot,\cdot)$ can be kept unchanged by shifting the summation index range from $1:2^L-1$ to $2:2^L$. Thus

$$\mathcal{G}(\mathbf{W}_0+\Delta\mathbf{W}) \leq \mathcal{G}(\Delta\mathbf{W}) + 2\,L_{\mathcal{L}}\left( C\prod_{k=1}^{L}L_{\sigma_k}\Big[\sum_{i=2}^{2^L}\prod_{j=1}^{L}P_L(i,j)\Big] + \sum_{i=3;i\neq 2^a;a\in[L]}^{2^L-1}F(i)\right).$$

Consequently, we can conclude with:

$$\mathcal{G}(\mathbf{W}_0+\Delta\mathbf{W}) \leq \min\big(\mathcal{G}(\mathbf{W}_0)+\Phi_{\Delta\mathbf{W}}, \mathcal{G}(\Delta\mathbf{W})+\Phi_{\mathbf{W}_0}\big)$$

$$\Phi_{\mathbf{W}} = \begin{cases} 2\,L_{\mathcal{L}}\left( C\prod_{k=1}^{L}L_{\sigma_k}\Big[\sum_{i=1}^{2^L-1}\prod_{j=1}^{L}P_L(i,j)\Big] + \sum_{i\neq 2^a-1}^{2^L-2}F(i)\right), & \text{for } \mathbf{W}=\Delta\mathbf{W}, \\[2mm] 2\,L_{\mathcal{L}}\left( C\prod_{k=1}^{L}L_{\sigma_k}\Big[\sum_{i=2}^{2^L}\prod_{j=1}^{L}P_L(i,j)\Big] + \sum_{i\neq 2^a}^{2^L-1}F(i)\right), & \text{for } \mathbf{W}=\mathbf{W}_0. \end{cases}$$

$\square$

### C.1.3 Neural Network with No activation Function—Special case of Theorem 1

We can upper bound the generalization error of a neural network with no nonlinear activation functions, i.e., $\sigma_i = I_{n_i}$ for all $i\in[L]$. We additionally include the simplest case of a one-layer linear network.

**Corollary 1.** *Let Assumption 1 hold and $\mathcal{L}$ follow Assumption 2. Let $\sigma_i = I_{n_i}$, for all $i\in[L]$, and* $f_{\mathbf{W}_0+BA}(x) = (W_0^L + B^L A^L(\cdots(W_0^2 + B^2 A^2(W_0^1 + B^1 A^1)x)\cdots))$. *Then we have:*

$$\mathcal{G}(\mathbf{W}_0+\Delta\boldsymbol{W}) \leq \min\big(\mathcal{G}(\mathbf{W}_0) + 2CL_{\mathcal{L}}\sum_{i=1}^{2^L-1}\prod_{j=1}^{L}P_L(i,j), \mathcal{G}(\Delta\boldsymbol{W}) + 2CL_{\mathcal{L}}\sum_{i=2}^{2^L}\prod_{j=1}^{L}P_L(i,j)\big).$$

**Remark 1.** Let Assumption 1 hold and $\mathcal{L}$ follow Assumption 2. If $L = 1$, i.e., the model consists of only 1 layer, then we have:
$$\mathcal{G}(\mathbf{W}_0 + \Delta\mathbf{W}) \leq \min(\mathcal{G}(\mathbf{W}_0) + 2CL_{\mathcal{L}}\|\Delta W\|, \mathcal{G}(\Delta\mathbf{W}) + 2CL_{\mathcal{L}}\|W_0\|).$$

### C.1.4 Tightness of the bounds in Theorem 1

We demonstrate Theorem 1 as an appropriate upper bound on the generalization error. We show the trivial case where $f_{\mathbf{W}_0 + \Delta\mathbf{W}} = f_{\mathbf{W}_0}$ and guarantee that $\mathcal{G}(\mathbf{W}_0 + \Delta\mathbf{W}) = \mathcal{G}(\mathbf{W}_0)$.

Assume $\Delta\mathbf{W}$ was never trained, i.e., $\|\Delta W^{(i)}\| = 0$, for all $i \in [L]$. Denote $\hat{F} := \sum_{i=1|i \neq 2^a - 1; i \in [L]}^{2^L - 2} F(i)$ Then we have:

$$|\mathcal{G}(\mathbf{W}_0 + \Delta\mathbf{W}) - \mathcal{G}(\mathbf{W}_0)| \overset{\text{Theorem 1}}{\leq} 2L_{\mathcal{L}}(C\prod_{i=1}^{L} L_{\sigma_i} \sum_{i=1}^{2^L - 1} \prod_{j=1}^{L} P_L(i,j) + \sum_{i \neq 2^a - 1 : a \in [L]}^{2^L - 2} F(i))$$

$$= 2L_{\mathcal{L}}\big(C\prod_{i=1}^{L} L_{\sigma_i}(\|W_0^{(i)}\|_2 + \|\Delta W^{(i)}\|_2) - C\prod_{i=1}^{L} L_{\sigma_i}\|W_0^{(i)}\|_2 + \hat{F}\big)$$

$$\overset{\|\Delta\mathbf{W}^{(i)}\|_2 = 0;}{=} 2L_{\mathcal{L}}\big(C\prod_{i=1}^{L} L_{\sigma_i}\|W_0^{(i)}\|_2 - C\prod_{i=1}^{L} L_{\sigma_i}\|W_0^{(i)}\|_2 + \hat{F}\big)$$

$$= 2L_{\mathcal{L}}\hat{F}.$$

Since each $F(i)$ does not take entries from $2^a - 1$, where $a \in [L]$, at least one $H(i,j)$ returns the spectral norm of one of the $\Delta\mathbf{W}$ layers, returning 0 by construction. Hence, each $F(i)$ returns 0 and we obtain the result: $|\mathcal{G}(\mathbf{W}_0 + \Delta\mathbf{W}) - \mathcal{G}(\mathbf{W}_0)| \leq 0$ confirming that $\mathcal{G}(\mathbf{W}_0 + \Delta\mathbf{W}) = \mathcal{G}(\mathbf{W}_0)$, if $\Delta\mathbf{W}$ was never trained. This way, we make sure the generalization measure would be unchanged and does not risk including unnecessary terms.

### C.1.5 Adapting Theorem 1 under special cases

To adapt Theorem 1 under special cases, we need the following general assumptions.

**Assumption 4.** *The loss function, $\ell(\cdot) : \mathbb{R}^d \to \mathbb{R}$, is 1-Lipschitz, i.e, $|\ell(f_{\mathbf{W}}(x), y) - \ell(f_{\mathbf{W'}}(x), y)| \leq \|f_{\mathbf{W}}(x) - f_{\mathbf{W'}}(x)\|$ for all $\mathbf{W}, \mathbf{W'} \in \mathbb{R}^d$ and $(x, y) \in \mathcal{X} \times \mathcal{Y}$.*

**Assumption 5.** *The loss function, $\ell(\cdot) : \mathbb{R}^d \to \mathbb{R}$, is bounded, i.e., there exists a constant $C_2 \geq 0$ such that $\|\ell(f_{\mathbf{W}}(x), y)\| \leq C_2$, for all $\mathbf{W} \in \mathbb{R}^d$ and $(x, y) \in \mathcal{X} \times \mathcal{Y}$.*

(*i*) **Perturbing around $\mathcal{G}(\mathbf{W}_0)$.** First, we adapt Theorem 4.1 in (28) into our notation and quote it below.

**Theorem 3.** *(PAC-Bayes generalization bound for fine-tuning)[(28), Theorem 4.1] Let Assumption 1 hold with the requirement that $C \geq 1$. Let the loss function, $\mathcal{L}$, follow Assumptions 4 and 5. Let $\|W_0^{(i)}\|_2 \leq \mathcal{A}_i$ with fixed $\mathcal{A}_i > 1$, $\|\Delta W^{(i)}\| \leq Q_i$, for all $i \in [L]$ and $V = \max_{i \in [L]}\{m_i, n_i\}$. Let $\epsilon$ and $\delta$ be arbitrary small values. Then with probability $1 - 2\delta$, the following inequality holds:*

$$\mathcal{G}(\mathbf{W}_0 + \Delta W) \leq \epsilon + C_2\sqrt{\frac{\frac{36}{\epsilon^2}C^2 V \log(4LVC_2)(\sum_{i=1}^{L} \frac{\prod_{j=1}^{L}(\mathcal{A}_j + Q_j)}{\mathcal{A}_i + Q_i})^2(\sum_{i=1}^{L} Q_i^2) + 3\ln\frac{|N|}{\delta} + 8}{|N|}}.$$

We now use Theorem 3, to obtain a bound for $\mathcal{G}(\mathbf{W}_0)$. The following Theorem gives that.

**Theorem 4.** *Using the Assumptions made for Theorem 1 and Theorem 3, the following inequality holds with probability at least $1 - 2\delta$ :*

$$\mathcal{G}(\boldsymbol{W}_0 + \Delta W) \leq \epsilon + C_2\sqrt{\frac{3\ln\frac{|N|}{\delta} + 8}{|N|}} + 2L_{\mathcal{L}}\big(C\prod_{i=1}^{L} L_{\sigma_i} \sum_{i=1}^{2^L - 1} \prod_{j=1}^{L} P(i,j) + \sum_{i \neq 2^a - 1 : a \in [L]}^{2^L - 2} F(i)\big).$$

*Proof.* We wish to find $\mathcal{G}(\mathbf{W}_0)$, and note that if we never train the model, we obtain the expression $\|W_0^{(i)} - W_0^{(i)}\|_2 = 0$. Thus, we can use $Q_i = 0$ for all $i \in [L]$ and obtain:

$$\mathcal{G}(\mathbf{W}_0) \overset{\text{Theorem 3}}{\leq} \epsilon + C_2 \sqrt{\frac{\frac{36}{\epsilon^2}C^2 V \log(4LVC_2)(\sum_{i=1}^{L} \frac{\prod_{j=1}^{L}(\mathcal{A}_j + Q_j)}{\mathcal{A}_i + Q_i})^2(\sum_{i=1}^{L} Q_i^2) + 3\ln\frac{|N|}{\delta} + 8}{|N|}}$$

$$\overset{Q_i = 0; i \in [L]}{=} \epsilon + C_2 \sqrt{\frac{\frac{36}{\epsilon^2}C^2 V \log(4LVC_2)(\sum_{i=1}^{L} \frac{\prod_{j=1}^{L}(\mathcal{A}_j + 0)}{\mathcal{A}_i + 0})^2(\sum_{i=1}^{L} 0^2) + 3\ln\frac{|N|}{\delta} + 8}{|N|}}$$

$$= \epsilon + C_2 \sqrt{\frac{3\ln\frac{|N|}{\delta} + 8}{|N|}}.$$

Now that we have an upper bound for $\mathcal{G}(\mathbf{W}_0)$, we can apply Theorem 1 and obtain the following:

$$\mathcal{G}(\mathbf{W}_0 + \Delta\mathbf{W}) \overset{\text{Theorem 1}}{\leq} \mathcal{G}(\mathbf{W}_0) + \Phi_{\Delta\mathbf{w}}$$

$$\leq \epsilon + C_2 \sqrt{\frac{3\ln\frac{|N|}{\delta} + 8}{|N|}} + \Phi_{\Delta\mathbf{w}}.$$

By substituting the expression for $\Phi_{\Delta\mathbf{W}}$, in the above expression we have:

$$\mathcal{G}(\mathbf{W}_0 + \Delta\mathbf{W}) \leq \epsilon + C_2 \sqrt{\frac{3\ln\frac{|N|}{\delta} + 8}{|N|}} + 2L_{\mathcal{L}}\Big(C\prod_{i=1}^{L} L_{\sigma_i} \sum_{i=1}^{2^L - 1} \prod_{j=1}^{L} P(i,j) + \sum_{i \neq 2^a - 1 : a \in [L]}^{2^L - 2} F(i)\Big).$$

This concludes the proof. $\qquad\square$

(*i*) **Perturbing around** $\mathcal{G}(\mathcal{A})$**.** First, we make another assumption on the loss function and then adapt Theorem 1 in (58) to our notation.

**Assumption 6.** *The loss function,* $\ell(\cdot) : \mathbb{R}^d \to \mathbb{R}$, *is* $\sigma$-*sub-gaussian, i.e.,* $\mathbb{E}(e^{\lambda[\ell(f_\mathbf{w}(X),Y) - \mathbb{E}(\ell(f_\mathbf{w}(X),Y))]}) \leq e^{\frac{\lambda^2 \sigma^2}{2}}$ *for all* $\lambda \in \mathbb{R}$, $\mathbf{W} \in \mathbb{R}^d$.

**Theorem 5.** *(Upper bound on generalization error using mutual information)[Theorem 1 (58)] Let* $\mathcal{A}$ *denote a LoRA-based algorithm that outputs* $\{\Delta\mathbf{W}_i\}_{i \in [L]}$ *on a fine-tuning dataset,* $N$. *By* $\nu$ *we denote the underlying distribution of the input space,* $\mathcal{X}$, *of which the elements of the fine-tuning dataset* $N$ *are chosen following i.i.d. Let Assumption 6 hold. Then we have the following:*

$$\mathcal{G}(\mathcal{A})_\nu \leq \sqrt{\frac{2\sigma^2 \mathbf{I}(\{\Delta\mathbf{W}_i\}_{i \in [L]}; N | \mathcal{A}; \mathbf{W})}{|N|}}.$$

Let the loss function $\mathcal{L}$ follow Assumption 6. We present the generalization error upper bounds of the LoRA variants in Table 1. For this, we use the inequality $\mathcal{G}(\mathbf{W}_0 + \mathcal{A}) \leq \mathcal{G}(\mathcal{A}) + \Phi_{\mathbf{W}_0}$, where $\mathcal{G}(\mathcal{A})$ is upper bounded by the use of Lemma 1 quoted below.

**Lemma 1.** *(Upperbound on mutual-information)[(58)] Let* $\{\Delta\mathbf{W}_i\}_{i \in [L]}$ *be an update to a learning algorithm. Then the mutual information is upper bounded by the uniform distribution over an updated support set, i.e.,* $\mathbf{I}(\Delta\{\mathbf{W}_i\}_{i \in [L]}; N | \mathcal{A}; \mathbf{W}) \leq \ln 2^{qp} = qp\ln 2$, *where* $q$ *represents the number of bits the learning algorithm is quantized on, and* $p$ *is the number of trainable parameters. Thus, with the use of Theorem 5, if Assumption 6 holds, then* $\mathcal{G}(\mathcal{A}) \leq \sqrt{\frac{2\sigma^2 qp\ln 2}{|N|}}$.

**How do we arrive at the bounds of different LoRA variants?**

(*a*) **LoRA+** has $\mathcal{G}(\mathcal{A})$ upper bounded by $\sqrt{\frac{2rq\sigma^2\ln 2 \sum_{i=1}^{L}(m_i + n_i)}{|N|}}$. The learning rate does not alter the number of trainable parameters, which leads LoRA+ to possess the same upper bounds as LoRA. We note a unique observation regarding this claim, as $\gamma_A \to 0$, LoRA+ takes the lowered generalization error bound of Asymmetric LoRA since the adapter matrix, $A$, is no longer trainable.

(*b*) **cLA** has the fine-tuned update $B[I_r | \mathbf{0}_{m_i - r}]$, where $[I_r | \mathbf{0}_{m_i - r}]$ is a fixed constant orthogonal matrix. Thus, by using data processing inequality (14), the mutual information between the two is preserved, i.e,

$$\mathbf{I}(\{B_i[I_r | \mathbf{0}_{m_i - r}]_i\}_{i \in [L]}; N | \mathcal{A}; \mathbf{W}) = \mathbf{I}(\{B_i\}_{i \in [L]}; N | \mathcal{A}; \mathbf{W}).$$

Table 4: **Summary of the benchmarks, quality metrics, and trainable parameters.** For LoRA and Asymmetric LoRA methods, we report their ratio of trainable parameters relative to FFT.

| Task | Model | Pretrained On | Fine-Tuned On | Trainable Parameters (FFT) | Quality Metric |
|---|---|---|---|---|---|
| Natural Language Processing | RoBERTa-Base | English language corpora | MRPC | 124.6M | Accuracy |
| | | | CoLA | 124.6M | MCC |
| | RoBERTa-Large | English language corpora | MRPC | 355.4M | Accuracy |
| | | | CoLA | 355.4M | MCC |
| | DeBERTa v2 XXL | English language corpora | MRPC | 1.56B | Accuracy |
| | | | TREC-50 | 1.56B | Accuracy |
| | | | PAWS | 1.56B | Accuracy |
| | DeBERTa v3 Base | English language corpora | MRPC | 184.4M | Accuracy |
| | | | RTE | 184.4M | Accuracy |
| | | | STS-B | 184.4M | Accuracy |
| | | | TREC-50 | 184.4M | Accuracy |
| | | | PAWS | 184.4M | Accuracy |
| | GPT2-Small | WebText | E2E | 124.4M | Accuracy |
| Image Classification | ViT-Tiny | ImageNet-1K | OfficeHome | 5.54M | Accuracy |
| | | | Cifar10 | 5.53M | Accuracy |
| | ViT-Base | ImageNet-21K then ImageNet-1K | OfficeHome | 85.8M | Accuracy |
| | | | Cifar10 | 85.8M | Accuracy |
| Coding Generation | DeepSeek-Coder-Base | Repo-Level Code Corpus | DJANGO | 1.35B | Exact Match |
| Logical Reasoning | TinyLlama | SlimPajama | OpenBookQA | 1.03B | Accuracy |
| | | | FOLIO | 1.03B | Accuracy |
| | | | LogiQA | 1.03B | Accuracy |
| | | | CLUTRR | 1.03B | Accuracy |

Similar to (65), we upper bound mutual information by the uniform distribution of a model's support; particularly $\mathbf{I}(\{B_i\}_{i\in[L]}; N|\Delta\mathbf{W}; \mathbf{W}) \leq qr\ln 2\sum_{i=1}^{L} n_i$, by Lemma 1. Finally, by Theorem 5, we obtain the result $\mathcal{G}(\mathcal{A}) \leq \sqrt{\frac{2rq\sigma^2\ln 2\sum_{i=1}^{L} n_i}{|N|}}$.

(c) $\mathbf{c^3LA}$ has the fine-tuned update $B_1[I_r|\mathbf{0}_{m_i-r}] + B_2[\mathbf{0}_r|I_r|\mathbf{0}_{m_i-2r}] + \cdots + B_k[\mathbf{0}_{r(k-1)}|I_r|\mathbf{0}_{m_i-kr}]$. This expansion can be simplified by $\sum_{j=1}^{k} B_j[0_{r\times r(j-1)} \mid I_r \mid 0_{r\times(m_i-rj)}] = [B_1 \mid B_2 \mid \cdots \mid B_k|\mathbf{0}_{n_i(m_i-kr)}]$. Using (13), we can upper bound the rank of $\sum_{j=1}^{k} B^j[0_{r\times r(j-1)} \mid I_r \mid 0_{r\times(m_i-rj)}]$ by $kr$. Thus, the mapping $[B_1|\cdots|B_k] \to \Delta\mathbf{W}$ is injective and can be inverted by slicing the last $n_i - kr$ columns. Using DPI, this leads to the expression

$$\mathbf{I}(\{\sum_{j=1}^{k} B_i^j[0_{r\times r(j-1)} \mid I_r \mid 0_{r\times(m_i-rj)}]\}_{i\in[L]}; N|\mathcal{A}; \mathbf{W}) = \mathbf{I}(\{[B_1|\cdots|B_k]_i\}_{i\in[L]}; N|\mathcal{A}; \mathbf{W}).$$

We upper bound $\mathbf{I}(\{[B_1|\cdots|B_k]_i\}_{i\in[L]}; N|\mathcal{A}; \mathbf{W})$ by $qrk\ln 2\sum_{i=1}^{L} n_i$, using Lemma 1. Hence, by Theorem 5, we obtain: $\mathcal{G}(\mathcal{A}) \leq \sqrt{\frac{2rq\sigma^2 k\ln 2\sum_{i=1}^{L} n_i}{|N|}}$.

(d) **CoLA** has the update structure $\Delta\mathbf{W} = \sum_{j=1}^{k} B^j A^j$. Using inequality (13), we upper bound the rank of each layer's update by $kr$. By Lemma 1, we upper bound $\mathbf{I}(\{\sum_{j=1}^{L} B_i^j A_i^j\}_{i\in[L]}; N|\mathcal{A}; \mathbf{W})$ by $qrk\ln 2\sum_{i=1}^{L}(m_i + n_i)$. Hence, we obtain $\mathcal{G}(\mathcal{A}) \leq \sqrt{\frac{2rq\sigma^2 k\ln 2\sum_{i=1}^{L}(m_i+n_i)}{|N|}}$, by Theorem 5.

(e) **RAC-LoRA** has the fine-tuned update $\sum_{j=1}^{k} B^j Q^j$, where we consider each $Q^j$ to be a frozen orthogonal matrix. This update can be represented by $\sum_{j=1}^{k} B^j Q^j = [B^1|B^2|\cdots|B^k][Q^1|Q^2|\cdots|Q^k]^T$, where we can invert $[B^1|B^2|\cdots|B^k][Q^1|Q^2|\cdots|Q^L]^T]$ to $[B^1|B^2|\cdots|B^L]$. Thus by using inequality (13), DPI, and Lemma 1 we have
$\mathbf{I}(\{[B^1|B^2|\cdots|B^L]_i[Q^1|Q^2|\cdots|Q^L]_i^T]\}_{i\in[L]}; N|\mathcal{A}; \mathbf{W}) = \mathbf{I}(\{[B^1|B^2|\cdots|B^L]_i\}_{i\in[L]}; N|\mathcal{A}; \mathbf{W})$,
which is

$$\mathbf{I}(\{[B^1|B^2|\cdots|B^L]_i\}_{i\in[L]}; N|\mathcal{A}; \mathbf{W}) \leq qrk\ln 2\sum_{i=1}^{L} n_i.$$

Hence, by Theorem 5, we have the result: $\mathcal{G}(\mathcal{A}) \leq \sqrt{\frac{2rq\sigma^2 k\ln 2\sum_{i=1}^{L} n_i}{|N|}}$.

## C.2 NONCONVEX CONVERGENCE

We make the following general assumptions to prove our nonconvex convergence result.

**Assumption 7.** *(Lipschitz gradient) The gradients are Lipschitz continuous. That is, there exists a constant $L_G$ such that for any $\mathbf{W} = (W^1, ..., W^L)$ and $\Delta \mathbf{W} = (\Delta W^1, ..., \Delta W^L)$, $\|\nabla \mathcal{L}(\mathbf{W} + \Delta \mathbf{W}) - \nabla \mathcal{L}(\mathbf{W})\| \leq L_G \|\Delta \mathbf{W}\|$.*

**Assumption 8.** *(Global minimum) There exists $\boldsymbol{W}^\star$ such that $\mathcal{L}^\star := \mathcal{L}(\boldsymbol{W}^\star) \leq \mathcal{L}(\boldsymbol{W})$, for all $\boldsymbol{W} \in \mathbb{R}^d$.*

We start by introducing a block structure for the parameter space of the network that accurately represents the layer-wise updates of LoRA done in practice with the vector input convention for parameter space done in theory, followed by the assumptions and properties used to prove the results in 3.2.

**Layerwise structure.** LoRA modifies each layer in a network. We adopt block structure notation where each block is a layer in the neural network in order to formally present the proofs. We use the convention in (47).

**Definition 2.** We define $\beta(\cdot) : \mathbb{R}^{n \times m} \to \mathbb{R}^{nm}$ to be a function that applies on a matrix, $A \in \mathbb{R}^{n \times m}$, and produces a vector $\beta(A) \in \mathbb{R}^{nm}$ by stacking the columns of the matrix $A$.

Let $d = \sum_{i=1}^{L} n_i m_i$ be the parameter count of the network. The block structure of $\mathbf{W}$ is given by a decomposition of $\mathbb{R}^d$ as follows: Let $\mathbf{U} = [U^1, ..., U^L]$ be a decomposition of $\mathbf{U} \in \mathbb{R}^d$ into $L$ submatrices, $U^i \in \mathbb{R}^{d \times n_i m_i}$. Then $U^i$ projects $i^{\text{th}}$ layer vectors to $\mathbb{R}^d$ in the following way:

$$
W^i = \begin{pmatrix} w_1 \\ \vdots \\ w_{n_i m_i} \end{pmatrix} \text{ is mapped by } U^i \text{ to } U^i W^i = \underbrace{\begin{pmatrix} 0 \\ \vdots \\ w_1 \\ \vdots \\ w_{n_i m_i} \\ \vdots \\ 0 \end{pmatrix}}_{\text{Only } i^{\text{th}} \text{ layer non-zero}} .
$$

Conversely, $U^{i^\top}$ projects vectors in $\mathbb{R}^d$ to the $i^{\text{th}}$ layer in the following way:

$$
\mathbf{W} = \begin{pmatrix} w_1 \\ \vdots \\ w_{n1m1} \\ w_{n_1 m_1 + 1} \\ \vdots \\ w_{n_1 m_1 + n_2 m_2} \\ \vdots \\ w_d \end{pmatrix} \text{ is mapped by } U^{i^\top} \text{ to } \quad U^{i^\top} \mathbf{W} = \underbrace{\begin{pmatrix} w_{\sum_{j=1}^{i-1} n_j m_j + 1} \\ \vdots \\ w_{\sum_{j=1}^{i} n_j m_j + n_i m_i} \end{pmatrix}}_{}
$$

Note that, by construction, $U^{i^\top} U^j = \begin{cases} I_{n_i m_i} & i = j \\ 0 & \text{otherwise} \end{cases}$.

**Proposition 2.** *(47) For any weight vector $\mathbf{W} = \begin{pmatrix} w_1 & \cdots & w_d \end{pmatrix}^\top$, the set $\{U^{i^\top} \mathbf{W}\}_{i=1}^{L}$ uniquely represents the $L$ layers of $\mathbf{W}$ mapped to their respective spaces $\mathbb{R}^{n_i m_i}$, and we define $W^i := U^{i^\top} \mathbf{W}$ as the weight vector of the $i^{\text{th}}$ layer.*

**L-Layer update.** We denote the full-fine tuning update $\Delta \mathbf{W} \in \mathbb{R}^d$ as follows:

$$
\mathbf{W} + \Delta \mathbf{W} = \mathbf{W} + \sum_{i=1}^{L} U^i \Delta W^i,
$$

where $\Delta W^i = U^{i^\top} \Delta \mathbf{W}$.

For LoRA, the layer-wise update is identical

$$\mathbf{W} + \mathbf{BA} = \mathbf{W} + \sum_{i=1}^{L} U^i \beta(B^i A^i), \tag{15}$$

where $\mathbf{BA} = (\frac{\alpha}{r} B^1 A^1, ..., \frac{\alpha}{r} B^L A^L)$, for each layer $i \in [L]$. Under this update structure, we have that $\nabla_i \mathcal{L}(\mathbf{W}) = {U^i}^\top \nabla \mathcal{L}(\mathbf{W})$. In the proof, we use the standard Euclidean inner product.

**Adapting the update rule.** The update rule $\mathbf{W}^{t+1} = \mathbf{W}^t - \sum_{i=1}^{L} \gamma \nabla_i \mathcal{L}(\mathbf{W}^t) H^{t,i}$, assumes $\nabla_i \mathcal{L}(\mathbf{W}^t) \in \mathbb{R}^{m_i \times n_i}$ and $H^{t,i} \in \mathbb{R}^{n_i \times n_i}$. In the layerwise structure, we have $\nabla_i \mathcal{L}(\mathbf{W}^t) \in \mathbb{R}^{n_i m_i}$. To represent the matrix-product $\nabla_i \mathcal{L}(\mathbf{W}^t) H^{t,i}$ in parameter space, we map the vector $\nabla_i \mathcal{L}(\mathbf{W}^t)$ to $\mathbb{R}^{m_i \times n_i}$, then map $\nabla_i \mathcal{L}(\mathbf{W}^t) H^{t.i}$ back to $\mathbb{R}^{n_i m_i}$. That is, $\beta(\beta^{-1}(\nabla_i \mathcal{L}(\mathbf{W}^t)) H^{t,i})$. To condense this, we apply Roth's lemma from (36) Theorem 18.5,

$$\beta(ABC) = (C^\top \otimes A)\beta(B) \tag{16}$$

to show that $\beta(\beta^{-1}(\nabla_i \mathcal{L}(\mathbf{W}^t)) H^{t,i}) = ({H^{t,i}}^\top \otimes I_{m_i}) \nabla_i \mathcal{L}(\mathbf{W}^t)$. Therefore, the adapted update rule for the layerwise structure is $\mathbf{W}^{t+1} = \mathbf{W}^t - \sum_{i=1}^{L} \gamma ({H^{t,i}}^\top \otimes I_{m_i}) \nabla_i \mathcal{L}$. To help make the proofs more clear, we define $\mathcal{H}^{t,i} := ({H^{t,i}}^\top \otimes I_{m_i})$.

### C.2.1 AUXILIARY RESULTS

**Proposition 3.** *Let $x, y \in \mathbb{R}^d$. Then, under the standard inner products in $\mathbb{R}^d, \mathbb{R}^{n_i m_i}, i \in [L]$ respectively, we have*

$$\langle x, y \rangle = \sum_{i=1}^{L} \langle U^i x^i, U^i y^i \rangle \tag{17}$$

*where $x^i = {U^i}^\top x, y^i = {U^i}^\top y, i \in [L]$.*

*Proof.* Based on the construction, we have that

$$
\begin{aligned}
\langle x, y \rangle &= \left\langle \sum_{i=1}^{L} U^i x^i, \sum_{j=1}^{L} U^j y^j \right\rangle \\
&= \sum_{j=1}^{L} \sum_{i=1}^{L} \left\langle {U^j}^\top U^i x^i, y^j \right\rangle \\
&= \sum_{i=1}^{L} \langle x^i, y^i \rangle.
\end{aligned}
$$

Hence the result. In particular, this shows that $\|\nabla \mathcal{L}(\mathbf{W})\|^2 = \sum_{i=1}^{L} \|\nabla_i \mathcal{L}(\mathbf{W})\|^2$. $\square$

### C.2.2 NONCONVEX CONVERGENCE RESULT

Now we are all set to prove our nonconvex convergence result.

**Theorem 2.** *Let Assumption 7 and 8 hold. Let $\lambda_{\min}^g > 0$ and the stepsize satisfy $0 < \gamma < \frac{1}{L_G}$. Let $\mathbf{W}^{(t,\cdot)}$ represent update steps with RAC-LoRA (9), or random-cLA (5), or $c^3LA$ (6), trained using gradient descent. Then the updates, $\{\mathbf{W}^{(0,\cdot)}, ..., \mathbf{W}^{(T,\cdot)}\}$ satisfy $\mathbb{E}[\|\nabla \mathcal{L}(\tilde{\mathbf{W}}^{(T)})\|^2] \leq \frac{2(\mathcal{L}(\mathbf{W}_0) - \mathcal{L}^*)}{\lambda_{\min}^g \gamma T}$, where $\tilde{\mathbf{W}}^{(T)}$ is sampled uniformly at random from $\{\mathbf{W}^{(0,\cdot)}, ..., \mathbf{W}^{(T,\cdot)}\}$.*

Table 5: **Summary of the hyperparameters.** We used the same learning rate for LoRA methods that train $B$, $A$, and Asymmetric LoRA methods that only train $B$, we write (FFT, LoRA, Asym) to indicate those three sets. We selected the best model out of all epochs based on the lowest validation loss, except for the CoLA dataset, where we used the lowest Matthews Correlation Coefficient. We used rank $r = 16$ and scaling factor $\alpha = 2r$ for all LoRA PEFT methods. For all models, we used the ADAM optimizer (26) with $(\beta_1, \beta_2, \epsilon) = (0.9, 0.999, 1e^{-8})$. For ViT, RoBERTa, and GPT2, we used gradient clipping on global $L_2$ norm with a max of 1, and did not otherwise. For LoRA+, the learning rate for our $B$ matrix is 16 times that of $A$.

| Model | Dataset | Scheduler (Warmup LR, Ratio) | Learning Rates (FFT,LoRA,Asym) | Chain reset frequency | Weight decay (FFT,LoRA) | Batch size | Epochs | Max length or Image size | Seeds |
|---|---|---|---|---|---|---|---|---|---|
| RoBERTa-Base | MRPC | Linear($1e^{-6}$, $0.1$) | ($1e^{-5}, 3e^{-4}, 3e^{-4}$) | 3 | (0.01, 0) | 32 | 20 | 128 | (12,22,32) |
| | CoLA | Linear($1e^{-6}$, $0.1$) | ($1e^{-5}, 3e^{-4}, 3e^{-4}$) | 3 | (0.01, 0) | 32 | 20 | 128 | (12,22,32) |
| RoBERTa-Large | MRPC | Linear($1e^{-6}$, $0.1$) | ($1e^{-5}, 3e^{-4}, 3e^{-4}$) | 3 | (0.01, 0) | 32 | 20 | 128 | (12,22,32) |
| | CoLA | Linear($1e^{-6}$, $0.1$) | ($1e^{-5}, 3e^{-4}, 3e^{-4}$) | 3 | (0.01, 0) | 32 | 20 | 128 | (12,22,32) |
| DeBERTa v2 XXL | MRPC | Constant | ($1e^{-5.5}, 1e^{-4.5}, 1e^{-4}$) | 5 | 0 | 8 | 25 | 512 | (100,101,102) |
| | TREC-50 | Constant | ($1e^{-5.5}, 1e^{-4.5}, 1e^{-4}$) | 5 | 0 | 8 | 25 | 512 | (100,101,102) |
| | PAWS | Constant | ($1e^{-6.5}, 1e^{-4.5}, 1e^{-4}$) | 5 | 0 | 8 | 10 | 512 | (100,101,102) |
| DeBERTa v3 Base | MRPC | Constant | ($1e^{-5}, 1e^{-3.5}, 1e^{-3}$) | 5 | 0 | 8 | 40 | 512 | (100,101,102) |
| | RTE | Constant | ($1e^{-4.75}, 1e^{-3.5}, 1e^{-3}$) | 5 | 0 | 8 | 40 | 512 | (100,101,102) |
| | STS-B | Constant | ($1e^{-4.75}, 1e^{-3.5}, 1e^{-3}$) | 5 | 0 | 8 | 40 | 512 | (100,101,102) |
| | TREC-50 | Constant | ($1e^{-4.75}, 1e^{-3.25}, 1e^{-3}$) | 5 | 0 | 8 | 40 | 512 | (100,101,102) |
| | PAWS | Constant | ($1e^{-5}, 1e^{-3.5}, 1e^{-3}$) | 5 | 0 | 8 | 20 | 512 | (100,101,102) |
| GPT2-Small | E2E | Linear($1e^{-6}$, $0.1$) | ($5e^{-5}, 3e^{-4}, 3e^{-4}$) | 1 | (0.01,0) | 16 | 30 | 64 | (12,22,32) |
| ViT-Tiny | OfficeHome | Cosine($1e^{-6}$, $0.05$) | ($3e^{-4}, 1e^{-3}, 1e^{-3}$) | 5 | (0.05,0) | 64 | 1 | 224 | (12,22,32) |
| | CIFAR-10 | Cosine($1e^{-6}$, $0.05$) | ($3e^{-4}, 1e^{-3}, 1e^{-3}$) | 5 | (0.05,0) | 64 | 1 | 224 | (12,22,32) |
| ViT-Base | OfficeHome | Cosine($1e^{-6}$, $0.05$) | ($3e^{-4}, 1e^{-3}, 1e^{-3}$) | 5 | (0.05,0) | 64 | 1 | 224 | (12,22,32) |
| | CIFAR-10 | Cosine($1e^{-6}$, $0.05$) | ($3e^{-4}, 1e^{-3}, 1e^{-3}$) | 5 | (0.05,0) | 64 | 1 | 224 | (12,22,32) |
| DeepSeek-Coder Base | DJANGO | Constant | ($1e^{-5.5}, 1e^{-4.5}, 1e^{-4}$) | 1 | 0 | 8 | 5 | 512 | (100,101,102) |
| TinyLlama | OpenBookQA | Constant | ($1e^{-6.25}, 1e^{-3.75}, 1e^{-3.25}$) | 2 | 0 | 8 | 10 | 512 | (100,101,102) |
| | FOLIO | Constant | ($1e^{-5}, 1e^{-3.75}, 1e^{-3.5}$) | 2 | 0 | 8 | 10 | 512 | (100,101,102) |
| | LogiQA | Constant | ($1e^{-5.75}, 1e^{-4}, 1e^{-3.25}$) | 2 | 0 | 8 | 10 | 512 | (100,101,102) |
| | CLUTRR | Constant | ($1e^{-6.25}, 1e^{-5.25}, 1e^{-4.75}$) | 2 | 0 | 8 | 10 | 512 | (100,101,102) |

*Proof.* Using the update rule and Lipschitz smoothness, we have

$$\mathcal{L}(\mathbf{W}^{t+1}) \quad = \quad \mathcal{L}(\mathbf{W}^t - \gamma \sum_{i=1}^{L} U^i \mathcal{H}^{t,i} \nabla_i \mathcal{L})$$

$$\overset{\text{By Assumption 7}}{\leq} \quad \mathcal{L}(\mathbf{W}^t) + \langle \nabla \mathcal{L}(\mathbf{W}^t), -\gamma \sum_{i=1}^{L} \mathcal{H}^{t,i} \nabla_i \mathcal{L} \rangle + \frac{L_G}{2} \| \gamma \sum_{i=1}^{L} \mathcal{H}^{t,i} \nabla_i \mathcal{L} \|^2$$

$$\overset{\|x^2\|=\langle x,x\rangle}{=} \quad \mathcal{L}(\mathbf{W}^t) - \gamma \sum_{i=1}^{L} \langle \nabla_i \mathcal{L}(\mathbf{W}^t), \mathcal{H}^{t,i} \nabla_i \mathcal{L} \rangle + \frac{L_G}{2} \gamma^2 \sum_{i=1}^{L} \langle \mathcal{H}^{t,i} \nabla_i \mathcal{L}, \mathcal{H}^{t,i} \nabla_i \mathcal{L} \rangle$$

$$\overset{\langle Ax,Ax\rangle=\langle x,A^\top Ax\rangle}{=} \quad \mathcal{L}(\mathbf{W}^t) - \gamma \sum_{i=1}^{L} \langle \nabla_i \mathcal{L}(\mathbf{W}^t), \mathcal{H}^{t,i} \nabla_i \mathcal{L} \rangle + \frac{L_G}{2} \gamma^2 \sum_{i=1}^{L} \langle \nabla_i \mathcal{L}(\mathbf{W}^t), \mathcal{H}^{t,i^\top} \mathcal{H}^{t,i} \nabla_i \mathcal{L} \rangle$$

$$\overset{\mathcal{H}^{t,i^\top} \mathcal{H}^{t,i}=\mathcal{H}^{t,i}}{=} \quad \mathcal{L}(\mathbf{W}^t) - \gamma \sum_{i=1}^{L} \langle \nabla_i \mathcal{L}(\mathbf{W}^t), \mathcal{H}^{t,i} \nabla_i \mathcal{L} \rangle + \frac{L_G}{2} \gamma^2 \sum_{i=1}^{L} \langle \nabla_i \mathcal{L}(\mathbf{W}^t), \mathcal{H}^{t,i} \nabla_i \mathcal{L} \rangle$$

$$\overset{\gamma \leq \frac{1}{L_G}}{\leq} \quad \mathcal{L}(\mathbf{W}^t) - \frac{\gamma}{2} \sum_{i=1}^{L} \langle \nabla_i \mathcal{L}(\mathbf{W}^t), \mathcal{H}^{t,i} \nabla_i \mathcal{L} \rangle.$$

Taking the expectation conditional on the randomness of $\mathbf{W}^t$, we have

$$\mathbb{E}[\mathcal{L}(\mathbf{W}^{t+1})|\mathbf{W}^t] \quad \leq \quad \mathbb{E}[\mathcal{L}(\mathbf{W}^t) - \frac{\gamma}{2} \sum_{i=1}^{L} \langle \nabla_i \mathcal{L}(\mathbf{W}^t), \mathcal{H}^{t,i} \nabla_i \mathcal{L} \rangle | \mathbf{W}^t]$$

$$= \quad \mathcal{L}(\mathbf{W}^t) - \frac{\gamma}{2} \sum_{i=1}^{L} \langle \nabla_i \mathcal{L}(\mathbf{W}^t), \mathbb{E}[\mathcal{H}^{t,i}|\mathbf{W}^t] \nabla_i \mathcal{L}(\mathbf{W}^t) \rangle$$

$$\overset{\mathbb{E}[\mathcal{H}^{t,i}|\mathbf{W}^t] \leq \lambda_{\min}^{H,i}}{\leq} \quad \mathcal{L}(\mathbf{W}^t) - \frac{\gamma}{2} \sum_{i=1}^{L} \lambda_{\min}^{H,i} \langle \nabla_i \mathcal{L}(\mathbf{W}^t), \nabla_i \mathcal{L}(\mathbf{W}^t) \rangle$$

$$\overset{\lambda_{\min}^{g} \leq \lambda_{\min}^{H,i} \text{ for all } i \in [L]}{\leq} \quad \mathcal{L}(\mathbf{W}^t) - \frac{\gamma}{2} \lambda_{\min}^{g} \sum_{i=1}^{L} \langle \nabla_i \mathcal{L}(\mathbf{W}^t), \nabla_i \mathcal{L}(\mathbf{W}^t) \rangle$$

$$\overset{\text{By equation (17)}}{=} \quad \mathcal{L}(\mathbf{W}^t) - \frac{\gamma}{2} \lambda_{\min}^{g} \| \nabla \mathcal{L}(\mathbf{W}^t) \|^2.$$

Subtracting $\mathcal{L}^*$ from both sides of the above relation and solving for $\frac{\gamma}{2} \lambda_{\min}^{g} \| \nabla \mathcal{L}(\mathbf{W}^t) \|^2$ we arrive at

$$\frac{\gamma}{2} \lambda_{\min}^{g} \| \nabla \mathcal{L}(\mathbf{W}^t) \|^2 \quad \leq \quad (\mathcal{L}(\mathbf{W}^t) - \mathcal{L}^*) - (\mathbb{E}[\mathcal{L}(\mathbf{W}^{t+1})|\mathbf{W}^t] - \mathcal{L}^*).$$

Denote $e^t := \mathbb{E}[\mathcal{L}(\mathbf{W}^t)|\mathbf{W}^{t-1}] - \mathcal{L}^*$. Taking expectation conditioned on $\mathbf{W}^{t-1}$, we have

$$\frac{\gamma}{2} \lambda_{\min}^{g} \mathbb{E}[\| \nabla \mathcal{L}(\mathbf{W}^t) \|^2 | \mathbf{W}^{t-1}] \leq e^t - e^{t+1}.$$

In the above relation, we take summation from $t = 0$ to $t = T - 1$, apply the telescoping property of $e^t - e^{t+1}$, and divide by $T$, to obtain

$$\sum_{t=0}^{T} \frac{\gamma}{2} \lambda_{\min}^{g} \mathbb{E}[\| \nabla \mathcal{L}(\mathbf{W}^t) \|^2] \leq \sum_{t=0}^{T-1} (e^t - e^{t+1}),$$

which further reduces to

$$\frac{1}{T} \sum_{t=0}^{T-1} \frac{\gamma}{2} \lambda_{\min}^{g} \mathbb{E}[\| \nabla \mathcal{L}(\mathbf{W}^t) \|^2] \leq \frac{e^0 - e^T}{T} \leq \frac{e^0}{T}.$$

Multiplying the above by $\frac{2}{\gamma \lambda_{\min}^{g}}$ we have

$$\frac{1}{T} \sum_{T=0}^{T-1} \mathbb{E}[\| \nabla \mathcal{L}(\mathbf{W}^t) \|^2] \leq \frac{2e^0}{\lambda_{\min}^{g} \gamma T} = \frac{2(\mathcal{L}(\mathbf{W}_0) - \mathcal{L}^*)}{\lambda_{\min}^{g} \gamma T}.$$

The left-hand side is equivalent to selecting one value uniformly from $\{\mathbf{W}^0, ..., \mathbf{W}^{T-1}\}$ for our argument of $\mathcal{L}$

$$\mathbb{E}[\|\nabla\mathcal{L}(\tilde{\mathbf{W}}^T)\|^2] \leq \frac{2(\mathcal{L}(\mathbf{W}_0) - \mathcal{L}^*)}{\lambda_{\min}^g \gamma T}.$$

This concludes the proof. $\qquad\square$

### C.2.3 Additional Results

**Proposition 4.** *Let* $A^i \in \mathbb{R}^{r \times n_i}$, *and* $\mathcal{D}^i = \{A^{1,i}, A^{2,i}, ..., A^{k,i}\}$, *where* $A^{j,i} = \left[\mathbf{0}_{r \times jr} \middle| I_r \middle| \mathbf{0}_{r \times (n_i-(j+1)r)}\right]$ *and* $n_i = rk$ *for some* $k \in \mathbb{N}$. *Then, for* $c^3LA$ *and random-CLA,* $\lambda_{\min}^{H,i} = \frac{r}{n_i}$, *for* $i \in [L]$.

*Proof.* By definition, $\lambda_{\min}^{H,i} = \lambda_{\min}[\mathbb{E}_{\mathcal{D}^i}[H^i]]$, and $H^{j,i} = (A^{j,i})^\top (A^{j,i}(A^{j,i})^\top)^\dagger A^{j,i}$. We start by writting $A^{j,i}$ in terms of row vectors

$$A^{j,i} = \begin{pmatrix} e_{j,1} & \cdots & e_{j,r}\end{pmatrix}^\top,$$

where $(e_{j,m})_p = \begin{cases} 1 & p = j + m \\ 0 & \text{otherwise}\end{cases}$ .

We calculate

$$A^{j,i}(A^{j,i})^\top = \begin{pmatrix} e_{j,1}e_{j,1}^\top & \cdots & e_{j,r}e_{j,1}^\top \\ \vdots & & \vdots \\ e_{j,1}e_{j,r}^\top & \cdots & e_{j,r}e_{j,r}^\top \end{pmatrix} = I_r.$$

Next, writing $A^{j,i}$ in terms of column vectors we have

$$A^{j,i} = \begin{pmatrix} c_{j,1} & \cdots & c_{j,n_i}\end{pmatrix},$$

where $(c_{j,m})_p = \begin{cases} 0 & p \neq m \pmod r \text{ or } m \leq jr \text{ or } m > (j+1)r \\ 1 & \text{otherwise}\end{cases}$ .

Next, we calculate

$$(A^{j,i})^\top A^{j,i} = \begin{pmatrix} c_{j,1}^\top c_{j,1} & \cdots & c_{j,n_i}^\top c_{j,1} \\ \vdots & & \vdots \\ c_{j,1}^\top c_{j,n_i}^\top & \cdots & c_{j,n_i}^\top c_{j,n_i}^\top \end{pmatrix} = \text{Diag}(0,...,0,\overbrace{1,...,1}^{r(j-1)\text{ to }r},0,...,0).$$

Using these, we find

$$\mathbb{E}_{\mathcal{D}^i}[H^i]] = \frac{1}{k}\sum_{j=1}^k \text{Diag}(0,...,0,\overbrace{1,...,1}^{r(j-1)\text{ to }r},0,...,0) = \text{Diag}(\frac{1}{k},...,\frac{1}{k}).$$

Finally, $\lambda_{\min}^{H,i} = \lambda_{\min}[\text{Diag}(\frac{1}{k},...,\frac{1}{k})] = \frac{1}{k} = \frac{r}{n_i}$. This concludes the proof. $\qquad\square$

**Theorem 6.** *(Smoothness conditions)[(51) Appendix A.1 Theorem 2] For a low-rank decomposition on model parameter* $\mathbf{W}$ *to* $\mathbf{W}_0 + \mathbf{B}\mathbf{A}_0$, *we have the following properties:*(i) *If* $\mathbf{B}$ *is trainable,* $\mathbf{A}_0$ *is fixed with* $\|A_0^i\| \leq C, i \in [L]$ *and* $\mathcal{L}(\mathbf{W})$ *is Lipschitz smooth with factor* $L_G$ *then the loss function* $\mathcal{L}(\mathbf{W}_0 + \mathbf{B}\mathbf{A}_0)$ *is Lipschitz smooth with respect to* $\mathbf{B}$ *with factor* $L_G C^2 \sqrt{L}$. (ii) *If both* $\mathbf{A}$ *and* $\mathbf{B}$ *are trainable and* $\mathcal{L}(\mathbf{W})$ *is Lipschitz smooth with factor* $L$, *the loss function* $\mathcal{L}(\mathbf{W}_0 + \mathbf{B}\mathbf{A}_0)$ *has no Lipschitz smoothness guarantees.*

*Proof.* First, denote the layerwise gradient with respect to $\mathbf{W}, \mathbf{B}$ as $\nabla_{\mathbf{W},i}\mathcal{L}(\mathbf{W}+\mathbf{B}\mathbf{A}_0), \nabla_{\mathbf{B},i}\mathcal{L}(\mathbf{W}+\mathbf{B}\mathbf{A}_0)$ respectively. Then $\nabla_{\mathbf{W},i}\mathcal{L}(\mathbf{W} + \mathbf{B}\mathbf{A}_0) = \nabla_{\mathbf{B},i}\mathcal{L}(\mathbf{W} + \mathbf{B}\mathbf{A}_0)A_0^{i\top}$ since, for each $i \in [L]$, we have

$$\begin{aligned} \langle B_1^i - B_2^i, \nabla_{\mathbf{B},i}\mathcal{L}(\mathbf{W} + \mathbf{B}\mathbf{A}_0)\rangle &= \langle B_1^i A_0^i - B_2^i A_0^i, \nabla_{\mathbf{W},i}\mathcal{L}(\mathbf{W} + \mathbf{B}\mathbf{A}_0)\rangle \\ &= \langle B_1^i - B_2^i, \nabla_{\mathbf{W},i}\mathcal{L}(\mathbf{W} + \mathbf{B}\mathbf{A}_0)A_0^{i\top}\rangle.\end{aligned}$$

Similarly, we have $\nabla_{\mathbf{A},i}\mathcal{L}(\mathbf{W} + \mathbf{B}\mathbf{A}) = B^{i\top}\nabla_{\mathbf{W},i}\mathcal{L}(\mathbf{W} + \mathbf{B}\mathbf{A})$ for $i \in [L]$. We provide the proof for each property below.

For property 1, we know that for any $\mathbf{B}_1, \mathbf{B}_2$ Asymmetric LoRA (8) updates with shared fixed $\mathbf{A}_0$, we have that

$$\|\nabla_{\mathbf{B}}\mathcal{L}(\mathbf{W} + \mathbf{B}_1\mathbf{A}_0) - \nabla_{\mathbf{B}}\mathcal{L}(\mathbf{W} + \mathbf{B}_2\mathbf{A}_0)\|$$

$$\overset{\nabla\mathcal{L}=\sum_{i=1}^{L}\nabla_i\mathcal{L}}{=} \|\sum_{i=1}^{L} U^i(\nabla_{\mathbf{B},i}\mathcal{L}(\mathbf{W} + \mathbf{B}_1\mathbf{A}_0) - \nabla_{\mathbf{B},i}\mathcal{L}(\mathbf{W} + \mathbf{B}_2\mathbf{A}_0))\|$$

$$\overset{\substack{\nabla_{\mathbf{B},i}\mathcal{L}(\cdot)=\nabla_{\mathbf{W},i}\mathcal{L}(\cdot)A_0^{i\top} \\ \text{and equation (16)}}}{=} \|\sum_{i=1}^{L} U^i((A_0^i \otimes I_{m_i})\nabla_{\mathbf{W},i}\mathcal{L}(\mathbf{W} + \mathbf{B}_1\mathbf{A}_0) - (A_0^i \otimes I_{m_i})\nabla_{\mathbf{W},i}\mathcal{L}(\mathbf{W} + \mathbf{B}_2\mathbf{A}_0))\|$$

$$\overset{\|PQ\|\leq\|P\|\|Q\|}{\leq} \sum_{i=1}^{L} \|U^i(\nabla_{\mathbf{W},i}\mathcal{L}(\mathbf{W} + \mathbf{B}_1\mathbf{A}_0) - \nabla_{\mathbf{W},i}\mathcal{L}(\mathbf{W} + \mathbf{B}_2\mathbf{A}_0))\|\|A_0^i \otimes I_{m_i}\|$$

$$\overset{\substack{\|A_0^i\otimes I_{m_i}\|_2=\|A_0^i\|_2 \\ \|A_0^i\|\leq C}}{\leq} C\sum_{i=1}^{L} \|U^i(\nabla_{\mathbf{W},i}\mathcal{L}(\mathbf{W} + \mathbf{B}_1\mathbf{A}_0) - \nabla_{\mathbf{W},i}\mathcal{L}(\mathbf{W} + \mathbf{B}_2\mathbf{A}_0))\|$$

$$\overset{\sum_{i=1}^{L}\|x^i\|\leq\sqrt{L\sum_{i=1}^{L}\|x^i\|^2}}{\leq} C\sqrt{L\sum_{i=1}^{L} \|U^i(\nabla_{\mathbf{W},i}\mathcal{L}(\mathbf{W} + \mathbf{B}_1\mathbf{A}_0) - \nabla_{\mathbf{W},i}\mathcal{L}(\mathbf{W} + \mathbf{B}_2\mathbf{A}_0))\|^2}$$

$$\overset{\text{By equation (17)}}{=} C\sqrt{L}\sqrt{\|\nabla_{\mathbf{W}}\mathcal{L}(\mathbf{W} + \mathbf{B}_1\mathbf{A}_0) - \nabla_{\mathbf{W}}\mathcal{L}(\mathbf{W} + \mathbf{B}_2\mathbf{A}_0)\|^2}$$

$$\overset{\mathcal{L}\text{ lipschitz smooth}}{\leq} L_G C\sqrt{L}\|\mathbf{W} + \mathbf{B}_1\mathbf{A}_0 - \mathbf{W} - \mathbf{B}_2\mathbf{A}_0\|$$

$$\overset{\text{By equation (15) and (16)}}{=} L_G C\sqrt{L}\|\sum_{i=1}^{L} U^i\beta((B_1^i - B_2^i)A_0^i)\|$$

$$\overset{\|x+y\|\leq\|x\|+\|y\|}{\leq} L_G C\sqrt{L}\sum_{i=1}^{L} \|U^i(A_0^{i\top} \otimes I_{m_i})\beta(B_1^i - B_2^i))\|$$

$$\overset{\substack{\|A_0^{i\top}\otimes I_{m_i}\|_2=\|A_0^i\|_2 \\ \|AB\|\leq\|A\|\|B\|}}{\leq} L_G C\sqrt{L}\sum_{i=1}^{L} \|A_0^i\|\|U^i(B_1^i - B_2^i)\|$$

$$\overset{\|A_0^i\|\leq C}{\leq} L_G C^2\sqrt{L}\sum_{i=1}^{L} \|U^i\beta(B_1^i - B_2^i)\|$$

$$\overset{\mathbf{B_k}=\sum_{i=1}^{L} U^i\beta(B_k^i)}{=} L_G C^2\sqrt{L}\|\mathbf{B_1} - \mathbf{B_2}\|.$$

This completes the proof of property 1.

For the second property, we construct a counter-example such that the 1-Lipschitz smooth function $\mathcal{L}(\mathbf{W}) = \frac{1}{2}\|\mathbf{W}\|^2$ is not Lipschitz smooth with respect to both B,A updating simultaneously $\nabla_{\mathbf{B},\mathbf{A}}$. Consider an MLP where $n_i = m_i, i \in [L]$, let $r = n_i$. Define the sequence $\{\mathbf{A}_k, \mathbf{B}_k\}_{k=1}^{\infty}$ such that $\mathbf{A}_k, \mathbf{B}_k = [kI_r^i], i \in [L]$.

Then

$$\lim_{k\to\infty} \frac{\|\nabla_{\mathbf{B},\mathbf{A}}\mathcal{L}(\mathbf{0}+\mathbf{B}_k\mathbf{A}_k)-\nabla_{\mathbf{B},\mathbf{A}}\mathcal{L}(\mathbf{0}+\mathbf{B}_0\mathbf{A}_0)\|}{\|\mathbf{B}_k\mathbf{A}_k-\mathbf{B}_0\mathbf{A}_0\|}$$

$$\overset{\nabla_{\mathbf{B},\mathbf{A}}\mathcal{L}=\begin{pmatrix}\nabla_{\mathbf{A}}\mathcal{L}\\\nabla_{\mathbf{B}}\mathcal{L}\end{pmatrix}}{=}\quad \lim_{k\to\infty}\frac{\|\nabla_{\mathbf{A}}\mathcal{L}(\mathbf{B}_k\mathbf{A}_k)\|+\|\nabla_{\mathbf{B}}\mathcal{L}(\mathbf{B}_k\mathbf{A}_k)\|}{\|\mathbf{B}_k\mathbf{A}_k\|}$$

$$\overset{\nabla\mathcal{L}=\sum_{i=1}^{L}\nabla_i\mathcal{L}}{=}\quad \lim_{k\to\infty}\frac{\|\sum_{i=1}^{L}\nabla_{\mathbf{A},i}\mathcal{L}(\mathbf{B}_k\mathbf{A}_k)\|+\|\sum_{i=1}^{L}\nabla_{\mathbf{B},i}\mathcal{L}(\mathbf{B}_k\mathbf{A}_k)\|}{\|\mathbf{B}_k\mathbf{A}_k\|}$$

$$\overset{\substack{\text{Definition}\\\text{By equation (16)}}}{=}\quad \lim_{k\to\infty}\frac{\|\sum_{i=1}^{L}k^3\beta(I_r)\|+\|\sum_{i=1}^{L}k^3\beta(I_r)\|}{\|\sum_{i=1}^{L}U^i(A_k^{i\top}\otimes I_{m_i})\beta(B_k^i)\|}$$

$$=\quad \lim_{k\to\infty}\frac{2Lk^3\|\beta(I_r)\|}{k^2\|\sum_{i=1}^{L}U^i(A_k^{i\top}\otimes I_{m_i})\beta(B_k^i)\|}$$

$$=\quad \infty.$$

This concludes the proof. $\qquad\square$

## D ADDENDUM TO BENCHMARKING AND EVALUATION

In §D.1, we summarize the quality metrics and trainable parameters used for training the models in Table 2 and provide the specific hyperparameters for fine-tuning each model for each dataset in Table 5. In §D.2, we present ablation studies on the effects of learning rate ($\gamma$), scaling factor ($\alpha$), and chain reset indices on the resulting test accuracy and test loss for varying ranks. In §D.3, we comment on the potential of our methods by naively leveraging the sparsity of our $A$ matrices. In §D.4.2 and §D.4.1, we extend 4.2 with the implementation details of the loss landscapes and provide additional loss landscapes and intruder dimension results. In §D.5, we extend section §3.1 with empirical results on generalization.

### D.1 IMPLEMENTATION DETAILS

We implement the framework in Python using PyTorch (41). We train all models with the ADAM optimizer (26). The training of most models was done with one 80 GB NVIDIA H100 GPU. The ablation studies on ViT-Tiny in Tables 7, 9, and 11 were trained using one NVIDIA V100 GPU. We provide the hyperparameter settings, i.e., the learning rates, learning rate scheduler, chain reset frequency, weight decay, batch size, training epochs, maximum token length or image resolution, and random seeds for all of the runs used in Table 5.

### D.2 THE EFFECTS OF LEARNING RATE, SCALING FACTOR, AND CHAIN RESET FREQUENCY ON QUALITY METRIC OVER VARIOUS RANKS

The ideal learning rate of an LLM tends to scale inversely with its size (30). Many papers suggest a default scaling factor of $2r$ (4; 49). (38) suggests that, for sufficiently low learning rates, performing a chain reset every epoch is optimal. We validate the first claim under LoRA fine-tuning methods via ablation studies over learning rates presented in Tables 6-7. Similarly, we assess the scaling factor baseline choice in Tables 8 and 9 and the optimal chain reset frequency in Table 10. For the ablation studies, we fine-tuned DeBERTaV3-Base on the MRPC, TREC-50, and PAWS for learning rate, MRPC and TREC-50 for scaling factor, and MRPC, CoLA, RTE, and TREC-50 for chain reset all over various ranks. We then re-ran the same experiments on ViT-Tiny fine-tuned on the OfficeHome and CIFAR-10 datasets. We ran for 30 epochs.

As shown in Tables 6 and 7, Asymmetric LoRA methods are more sensitive to varying learning rates than methods that train both matrices $B$, $A$. We notice that the cLA has a wide variety of acceptable learning rates. Furthermore, across varying ranks, cLA and $c^3$LA often underperform compared to other LoRA variants. As rank increases, this gap tends to narrow. This is a byproduct of their structure, limiting how much of the pretrained weights they can update at any one time.

For our ablation study on scaling factor shown in Tables , the use of $\alpha = 2r$ works as a baseline given how often it was the best choice 8 and 9. With Asymmetric methods, the ideal scaling factor

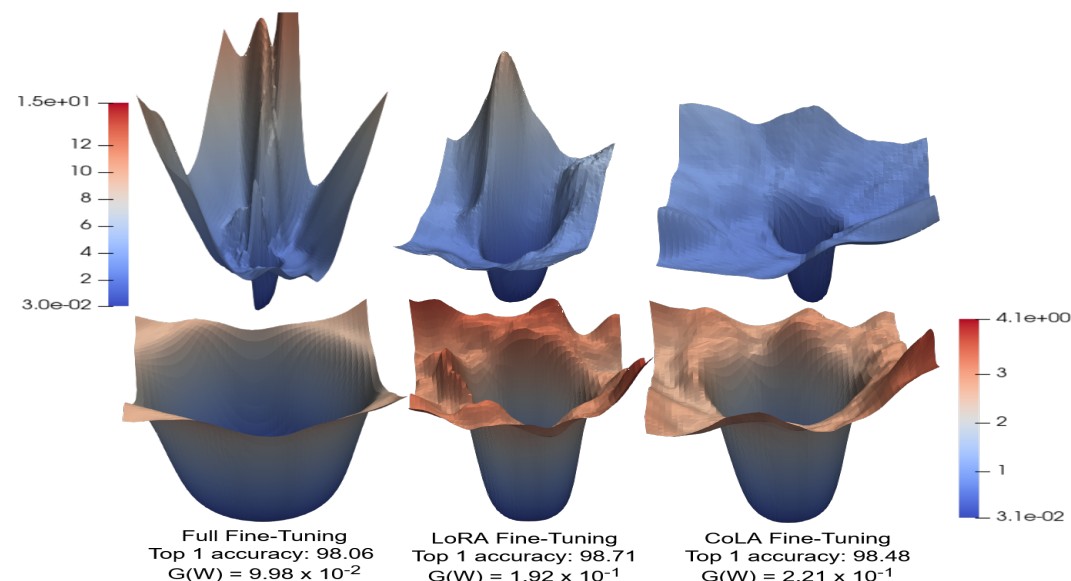

Figure 5: 3D loss landscapes of ViT-Base (11) pretrained on ImageNet-1K (7) and fine-tuned on CIFAR-10 (27) using the PCA directions of the model's weights updates (top) and random directions (bottom).

tends to be larger; this follows from the number of trainable parameters decreasing, requiring a larger effective learning rate, as the scaling factor can be interpreted as a scale on the learning rate.

Our ablation study on chain reset frequency, shown in Tables 10, revealed no clear correlation between the frequency of chain resets.

### D.3 COMPUTATIONAL COST, FLOPS, AND EFFICIENCY

We report empirical results regarding the computational efficiency of PEFT methods developed in this paper. We report the percentage of trainable parameters for each PEFT method in Table 4.

Our focus in this study is on examining the behaviors of the PEFT methods, including our proposed variant. Although the total fine-tuning time is an important factor, we do not present wall-clock results because our unoptimized implementation does not present a valid point for the training speedup that well-engineered PEFT methods can offer.

We naively leverage the sparsity inherent in the structure of cLA, $c^3$LA, and its random variants by replacing the $Ax$ product in $B(A(x))$ for each layer's LoRA adapter with a `gather` operation, removing any FLOPs accrued from the multiplications by zero. In Table 12, we show that this leads to a minor reduction in FLOPs. This could be improved by more advanced implementations of leveraging the sparsity. However, running inference on the frozen base model accrues most of the training FLOPs; thus, this use may be limited for very high rank adapters.

### D.4 PERFORMANCE ANALYSIS—CONTINUED

We extend §4.2 by reporting additional empirical results regarding PEFT models, including prediction capacity and model behaviors.

#### D.4.1 LOSS LANDSCAPE—CONTINUED

**3D landscapes.** We obtained the top two principle directions of the model's update path via PCA of the update matrix $[\mathbf{W}^0 - \mathbf{W}^T; ...; \mathbf{W}^{T-1} - \mathbf{W}^T]$, where $\{\mathbf{W}^t\}_{t=0}^{T}$. are the model weight's update steps. Let $\delta, \eta$ be those two directions. For random directions, we generate them via a Gaussian distribution. For LoRA methods, we merged the adapters into the base weights before calculating. We normalize the directions similar to the methods of (29). We plot the function $f(\alpha, \beta) := \mathcal{L}(\mathbf{W} + \alpha\delta + \beta\eta)$ over a $51^2$ grid of $\alpha, \beta$ values uniformly distributed over $[-2, 2] \times [-2, 2]$, we use mini-batches of size 12 when finding the values for $\mathcal{L}$.

Table 6: Test accuracies obtained by fine-tuning DeBERTa v3 on MRPC, TREC-50, and PAWS varying learning rates (columns), ranks (rows), and LoRA PEFT methods. We center our search at $1e^{-4}$. The learning rate for all methods decreases with increasing rank; the relationship between learning rate and model size observed in LLMs (30) persists when fine-tuning via LoRA methods. Chain methods and their non-chain counterparts produce the best results in similar learning rate ranges, therefore, chain resets do not influence the optimal learning rate. We repeated the experiment with ViT-Tiny on Table 7.

### DeBERTa v3 LoRA MRPC

| Rank/LR | 1e-6 | 1e-5.5 | 1e-5 | 1e-4.5 | 1e-4 | 1e-3.5 | 1e-3 | 1e-2.5 | 1e-2 |
|---|---|---|---|---|---|---|---|---|---|
| 2 | 66.4 | 66.4 | 79.9 | 84.2 | 85.5 | 87.3 | 88.1 | 66.4 | 66.4 |
| 4 | 66.4 | 74.4 | 81.8 | 84.6 | 85.6 | 87.9 | 87.7 | 66.4 | 66.4 |
| 8 | 66.4 | 76.1 | 83.0 | 85.1 | 86.9 | 87.3 | 87.1 | 66.4 | 66.4 |
| 16 | 66.4 | 78.1 | 83.1 | 85.0 | 86.8 | 87.9 | 72.9 | 66.4 | 66.4 |
| 32 | 66.4 | 80.1 | 83.9 | 84.9 | 87.1 | 87.9 | 66.4 | 66.4 | 66.4 |
| 64 | 76.1 | 81.7 | 84.6 | 85.1 | 88.0 | 81.5 | 66.4 | 66.4 | 66.4 |
| 128 | 77.5 | 81.9 | 84.9 | 86.2 | 88.5 | 87.6 | 66.4 | 66.4 | 66.4 |

### DeBERTa v3 CoLA MRPC

| Rank/LR | 1e-6 | 1e-5.5 | 1e-5 | 1e-4.5 | 1e-4 | 1e-3.5 | 1e-3 | 1e-2.5 | 1e-2 |
|---|---|---|---|---|---|---|---|---|---|
| 2 | 66.5 | 66.5 | 75.9 | 82.2 | 85.7 | 87.7 | 88.3 | 66.5 | 66.5 |
| 4 | 66.5 | 66.5 | 76.7 | 82.7 | 85.1 | 87.8 | 87.2 | 66.5 | 66.5 |
| 8 | 66.5 | 66.5 | 79.1 | 84.0 | 86.8 | 88.3 | 86.9 | 66.5 | 66.5 |
| 16 | 66.5 | 75.3 | 80.8 | 84.0 | 86.4 | 88.9 | 79.6 | 66.5 | 66.5 |
| 32 | 66.5 | 76.8 | 82.3 | 83.9 | 86.5 | 88.0 | 66.5 | 66.5 | 66.5 |
| 64 | 69.3 | 79.6 | 83.3 | 85.7 | 87.1 | 88.5 | 66.5 | 66.5 | 66.5 |
| 128 | 75.8 | 80.7 | 83.1 | 85.9 | 88.2 | 88.0 | 66.5 | 66.5 | 66.5 |

### DeBERTa v3 Asym MRPC

| Rank/LR | 1e-6 | 1e-5.5 | 1e-5 | 1e-4.5 | 1e-4 | 1e-3.5 | 1e-3 | 1e-2.5 | 1e-2 |
|---|---|---|---|---|---|---|---|---|---|
| 2 | 66.4 | 66.4 | 66.4 | 68.1 | 81.4 | 85.3 | 86.1 | 86.9 | 86.0 |
| 4 | 66.4 | 66.4 | 66.4 | 76.0 | 82.9 | 84.2 | 85.7 | 86.8 | 85.6 |
| 8 | 66.4 | 66.4 | 66.5 | 80.6 | 84.2 | 85.2 | 86.5 | 86.3 | 72.3 |
| 16 | 66.4 | 66.4 | 76.7 | 82.6 | 84.3 | 86.2 | 86.8 | 87.3 | 66.4 |
| 32 | 66.4 | 66.6 | 79.6 | 83.1 | 84.6 | 86.1 | 87.3 | 75.0 | 66.4 |
| 64 | 66.4 | 77.7 | 81.7 | 82.5 | 84.7 | 86.1 | 87.4 | 66.4 | 66.4 |
| 128 | 69.0 | 79.6 | 82.2 | 84.3 | 86.1 | 80.1 | 66.4 | 66.4 | 66.4 |

### DeBERTa v3 RAC MRPC

| Rank/LR | 1e-6 | 1e-5.5 | 1e-5 | 1e-4.5 | 1e-4 | 1e-3.5 | 1e-3 | 1e-2.5 | 1e-2 |
|---|---|---|---|---|---|---|---|---|---|
| 2 | 66.5 | 66.5 | 66.5 | 66.5 | 66.5 | 76.5 | 82.0 | 87.0 | 86.2 |
| 4 | 66.5 | 66.5 | 66.5 | 66.5 | 74.0 | 79.4 | 84.6 | 85.4 | 85.5 |
| 8 | 66.5 | 66.5 | 66.5 | 66.5 | 76.9 | 82.8 | 86.3 | 87.5 | 76.7 |
| 16 | 66.5 | 66.5 | 66.5 | 74.1 | 78.8 | 83.0 | 86.9 | 87.2 | 66.5 |
| 32 | 66.5 | 66.5 | 66.5 | 76.7 | 83.9 | 86.3 | 87.7 | 72.4 | 66.5 |
| 64 | 66.5 | 66.5 | 74.2 | 80.4 | 84.1 | 87.9 | 87.3 | 66.5 | 66.5 |
| 128 | 66.5 | 66.5 | 76.1 | 82.3 | 86.4 | 87.6 | 72.0 | 66.5 | 66.5 |

### DeBERTa v3 cLA MRPC

| Rank/LR | 1e-6 | 1e-5.5 | 1e-5 | 1e-4.5 | 1e-4 | 1e-3.5 | 1e-3 | 1e-2.5 | 1e-2 |
|---|---|---|---|---|---|---|---|---|---|
| 2 | 66.4 | 66.4 | 66.4 | 71.0 | 82.4 | 84.0 | 85.0 | 83.1 | 80.4 |
| 4 | 66.4 | 66.4 | 70.2 | 79.0 | 84.9 | 85.5 | 85.9 | 85.5 | 66.4 |
| 8 | 66.4 | 66.4 | 73.7 | 81.5 | 84.6 | 84.5 | 85.2 | 85.5 | 66.4 |
| 16 | 66.4 | 67.0 | 76.1 | 81.9 | 83.3 | 84.4 | 85.3 | 78.8 | 66.4 |
| 32 | 66.4 | 74.4 | 79.6 | 80.8 | 83.2 | 85.0 | 85.2 | 66.4 | 66.4 |
| 64 | 66.4 | 77.0 | 80.8 | 81.8 | 84.5 | 85.1 | 86.7 | 66.4 | 66.4 |
| 128 | 71.3 | 78.6 | 79.5 | 82.0 | 85.5 | 87.1 | 84.8 | 66.4 | 66.4 |

### DeBERTa v3 $c^3$LA MRPC

| Rank/LR | 1e-6 | 1e-5.5 | 1e-5 | 1e-4.5 | 1e-4 | 1e-3.5 | 1e-3 | 1e-2.5 | 1e-2 |
|---|---|---|---|---|---|---|---|---|---|
| 2 | 66.5 | 66.5 | 66.5 | 66.5 | 73.0 | 80.1 | 85.6 | 86.4 | 72.3 |
| 4 | 66.5 | 66.5 | 66.5 | 66.8 | 74.6 | 83.1 | 85.7 | 87.2 | 66.5 |
| 8 | 66.5 | 66.5 | 66.5 | 72.6 | 79.5 | 86.2 | 86.8 | 85.5 | 66.5 |
| 16 | 66.5 | 66.5 | 66.5 | 75.6 | 82.8 | 86.0 | 86.9 | 78.6 | 66.5 |
| 32 | 66.5 | 66.5 | 71.9 | 79.1 | 84.5 | 86.8 | 86.8 | 66.5 | 66.5 |
| 64 | 66.5 | 66.5 | 73.9 | 82.4 | 86.3 | 87.2 | 86.4 | 66.5 | 66.5 |
| 128 | 66.5 | 66.5 | 81.9 | 86.3 | 87.9 | 86.7 | 72.3 | 66.5 | 66.5 |

### DeBERTa v3 LoRA TREC-50

| Rank/LR | 1e-6 | 1e-5.5 | 1e-5 | 1e-4.5 | 1e-4 | 1e-3.5 | 1e-3 | 1e-2.5 | 1e-2 |
|---|---|---|---|---|---|---|---|---|---|
| 2 | 3.2 | 10.9 | 10.9 | 39.1 | 59.5 | 76.6 | 86.9 | 10.9 | 10.9 |
| 4 | 10.1 | 10.9 | 10.9 | 42.3 | 70.6 | 82.3 | 87.7 | 10.9 | 10.9 |
| 8 | 10.9 | 10.9 | 10.9 | 50.0 | 70.6 | 84.7 | 90.1 | 10.9 | 10.9 |
| 16 | 1.4 | 10.9 | 10.9 | 50.0 | 73.0 | 89.3 | 88.3 | 10.9 | 10.9 |
| 32 | 1.4 | 10.9 | 42.9 | 59.5 | 76.4 | 89.1 | 10.9 | 10.9 | 10.9 |
| 64 | 10.9 | 10.9 | 48.2 | 66.1 | 82.9 | 87.1 | 10.9 | 10.9 | 10.9 |
| 128 | 10.9 | 10.9 | 58.1 | 71.6 | 86.1 | 10.9 | 10.9 | 10.9 | 10.9 |

### DeBERTa v3 CoLA TREC-50

| Rank/LR | 1e-6 | 1e-5.5 | 1e-5 | 1e-4.5 | 1e-4 | 1e-3.5 | 1e-3 | 1e-2.5 | 1e-2 |
|---|---|---|---|---|---|---|---|---|---|
| 2 | 10.9 | 10.9 | 42.1 | 54.4 | 71.8 | 88.1 | 89.1 | 10.9 | 10.9 |
| 4 | 10.9 | 10.9 | 42.7 | 58.3 | 81.7 | 84.5 | 88.3 | 10.9 | 10.9 |
| 8 | 10.9 | 10.9 | 42.9 | 65.5 | 82.3 | 87.1 | 90.9 | 10.9 | 10.9 |
| 16 | 10.9 | 10.9 | 39.9 | 66.7 | 84.9 | 87.1 | 68.1 | 10.9 | 10.9 |
| 32 | 10.9 | 26.8 | 53.2 | 71.4 | 85.3 | 86.7 | 10.9 | 10.9 | 10.9 |
| 64 | 10.9 | 37.5 | 58.5 | 75.6 | 86.9 | 10.9 | 10.9 | 10.9 | 10.9 |
| 128 | 10.9 | 43.5 | 66.1 | 82.7 | 86.7 | 10.9 | 10.9 | 10.9 | 10.9 |

### DeBERTa v3 Asym TREC-50

| Rank/LR | 1e-6 | 1e-5.5 | 1e-5 | 1e-4.5 | 1e-4 | 1e-3.5 | 1e-3 | 1e-2.5 | 1e-2 |
|---|---|---|---|---|---|---|---|---|---|
| 2 | 10.9 | 10.9 | 10.9 | 32.5 | 46.2 | 80.6 | 86.5 | 82.5 | 26.6 |
| 4 | 10.9 | 10.9 | 10.9 | 33.3 | 58.9 | 85.1 | 88.1 | 85.7 | 10.9 |
| 8 | 10.9 | 10.9 | 10.9 | 40.1 | 73.6 | 87.3 | 86.7 | 84.5 | 10.9 |
| 16 | 10.9 | 10.9 | 10.9 | 42.9 | 78.8 | 89.1 | 86.9 | 82.7 | 10.9 |
| 32 | 10.9 | 10.9 | 10.9 | 57.5 | 83.1 | 90.9 | 91.7 | 56.5 | 10.9 |
| 64 | 10.9 | 10.9 | 41.9 | 73.0 | 88.5 | 90.7 | 87.5 | 10.9 | 10.9 |
| 128 | 10.9 | 10.9 | 52.2 | 78.8 | 89.9 | 90.9 | 10.9 | 10.9 | 10.9 |

### DeBERTa v3 RAC TREC-50

| Rank/LR | 1e-6 | 1e-5.5 | 1e-5 | 1e-4.5 | 1e-4 | 1e-3.5 | 1e-3 | 1e-2.5 | 1e-2 |
|---|---|---|---|---|---|---|---|---|---|
| 2 | 10.9 | 10.9 | 10.9 | 10.9 | 34.9 | 46.6 | 72.8 | 85.9 | 87.5 |
| 4 | 1.2 | 10.9 | 10.9 | 10.9 | 38.5 | 59.7 | 82.9 | 88.3 | 89.5 |
| 8 | 10.9 | 10.9 | 10.9 | 10.9 | 43.8 | 68.1 | 82.9 | 88.1 | 72.4 |
| 16 | 10.1 | 10.9 | 10.9 | 13.3 | 59.5 | 78.0 | 87.1 | 90.3 | 10.9 |
| 32 | 10.9 | 10.9 | 10.9 | 45.6 | 70.6 | 84.9 | 88.5 | 88.1 | 10.9 |
| 64 | 10.9 | 10.9 | 34.9 | 50.6 | 74.2 | 86.7 | 87.7 | 10.9 | 10.9 |
| 128 | 2.0 | 10.9 | 42.9 | 62.3 | 84.5 | 88.9 | 87.3 | 10.9 | 10.9 |

### DeBERTa v3 cLA TREC-50

| Rank/LR | 1e-6 | 1e-5.5 | 1e-5 | 1e-4.5 | 1e-4 | 1e-3.5 | 1e-3 | 1e-2.5 | 1e-2 |
|---|---|---|---|---|---|---|---|---|---|
| 2 | 10.9 | 10.9 | 10.9 | 10.9 | 10.9 | 40.9 | 71.2 | 81.3 | 34.5 |
| 4 | 9.5 | 10.9 | 10.9 | 10.9 | 35.3 | 61.9 | 79.6 | 82.7 | 10.9 |
| 8 | 0.4 | 10.9 | 10.9 | 10.9 | 53.0 | 71.8 | 83.1 | 86.3 | 10.9 |
| 16 | 10.9 | 10.9 | 10.9 | 40.7 | 60.7 | 84.3 | 85.5 | 87.5 | 10.9 |
| 32 | 10.1 | 10.1 | 10.9 | 47.0 | 70.0 | 86.1 | 88.9 | 10.9 | 10.9 |
| 64 | 10.9 | 10.9 | 42.7 | 62.3 | 76.2 | 86.9 | 89.1 | 52.4 | 10.9 |
| 128 | 3.6 | 10.9 | 50.2 | 67.1 | 85.1 | 86.7 | 66.7 | 10.9 | 10.9 |

### DeBERTa v3 $c^3$LA TREC-50

| Rank/LR | 1e-6 | 1e-5.5 | 1e-5 | 1e-4.5 | 1e-4 | 1e-3.5 | 1e-3 | 1e-2.5 | 1e-2 |
|---|---|---|---|---|---|---|---|---|---|
| 2 | 10.9 | 10.9 | 10.9 | 34.7 | 42.3 | 79.2 | 66.7 | 62.1 | 10.9 |
| 4 | 10.9 | 10.9 | 19.4 | 34.7 | 56.9 | 86.5 | 87.7 | 73.8 | 10.9 |
| 8 | 10.9 | 10.9 | 20.6 | 36.9 | 68.7 | 87.3 | 78.8 | 66.7 | 10.9 |
| 16 | 10.9 | 10.9 | 10.9 | 43.8 | 76.8 | 88.5 | 83.9 | 71.0 | 10.9 |
| 32 | 10.9 | 10.9 | 38.7 | 58.1 | 84.3 | 88.1 | 80.4 | 34.3 | 10.9 |
| 64 | 10.9 | 10.9 | 45.8 | 74.2 | 85.9 | 89.1 | 80.4 | 10.9 | 10.9 |
| 128 | 10.9 | 35.1 | 56.5 | 79.2 | 85.9 | 90.9 | 10.9 | 10.9 | 10.9 |

### DeBERTa v3 LoRA PAWS

| Rank/LR | 1e-6 | 1e-5.5 | 1e-5 | 1e-4.5 | 1e-4 | 1e-3.5 | 1e-3 | 1e-2.5 | 1e-2 |
|---|---|---|---|---|---|---|---|---|---|
| 2 | 92.1 | 93.7 | 94.0 | 94.2 | 94.7 | 94.5 | 94.0 | 55.8 | 55.8 |
| 4 | 92.3 | 94.1 | 94.0 | 94.4 | 94.2 | 94.4 | 94.0 | 55.8 | 55.8 |
| 8 | 92.4 | 93.5 | 94.3 | 94.3 | 94.7 | 94.5 | 93.5 | 55.8 | 55.8 |
| 16 | 93.1 | 94.0 | 94.4 | 94.6 | 94.6 | 93.6 | 55.8 | 55.8 | 55.8 |
| 32 | 93.8 | 93.9 | 94.7 | 94.7 | 55.8 | 55.8 | 55.8 | 55.8 | 55.8 |
| 64 | 93.6 | 94.1 | 94.6 | 94.6 | 94.6 | 93.5 | 55.8 | 55.8 | 50.0 |
| 128 | 94.0 | 94.2 | 94.4 | 94.7 | 94.7 | 55.8 | 55.8 | 55.8 | 50.0 |

### DeBERTa v3 CoLA PAWS

| Rank/LR | 1e-6 | 1e-5.5 | 1e-5 | 1e-4.5 | 1e-4 | 1e-3.5 | 1e-3 | 1e-2.5 | 1e-2 |
|---|---|---|---|---|---|---|---|---|---|
| 2 | 55.8 | 89.8 | 92.5 | 93.9 | 94.6 | 94.7 | 94.2 | 55.8 | 55.8 |
| 4 | 55.8 | 90.6 | 92.9 | 94.0 | 94.8 | 94.8 | 94.0 | 55.8 | 55.8 |
| 8 | 55.8 | 89.3 | 93.2 | 94.1 | 94.3 | 94.1 | 93.0 | 55.8 | 55.8 |
| 16 | 55.8 | 91.8 | 93.7 | 94.3 | 94.5 | 94.7 | 55.8 | 55.8 | 55.8 |
| 32 | 89.9 | 92.9 | 94.3 | 94.5 | 94.3 | 94.8 | 55.8 | 55.8 | 55.8 |
| 64 | 90.5 | 93.3 | 94.5 | 94.5 | 94.8 | 93.2 | 55.8 | 55.8 | 55.8 |
| 128 | 91.8 | 93.8 | 94.7 | 95.1 | 95.0 | 92.7 | 55.8 | 55.8 | 44.2 |

### DeBERTa v3 Asym PAWS

| Rank/LR | 1e-6 | 1e-5.5 | 1e-5 | 1e-4.5 | 1e-4 | 1e-3.5 | 1e-3 | 1e-2.5 | 1e-2 |
|---|---|---|---|---|---|---|---|---|---|
| 2 | 55.8 | 55.8 | 86.9 | 92.3 | 93.7 | 93.9 | 93.9 | 94.4 | 93.4 |
| 4 | 55.8 | 55.8 | 90.3 | 92.5 | 93.2 | 94.0 | 94.4 | 94.2 | 92.7 |
| 8 | 55.8 | 55.8 | 92.3 | 93.1 | 94.1 | 94.7 | 94.7 | 94.2 | 55.8 |
| 16 | 55.8 | 55.8 | 92.4 | 94.1 | 94.0 | 94.5 | 94.6 | 94.0 | 55.8 |
| 32 | 55.8 | 92.9 | 93.6 | 94.4 | 94.4 | 94.6 | 94.1 | 92.9 | 55.8 |
| 64 | 55.8 | 92.5 | 94.1 | 94.1 | 94.8 | 94.8 | 93.4 | 55.8 | 55.8 |
| 128 | 91.7 | 92.9 | 94.1 | 94.4 | 94.6 | 94.7 | 91.8 | 55.0 | 44.2 |

### DeBERTa v3 RAC PAWS

| Rank/LR | 1e-6 | 1e-5.5 | 1e-5 | 1e-4.5 | 1e-4 | 1e-3.5 | 1e-3 | 1e-2.5 | 1e-2 |
|---|---|---|---|---|---|---|---|---|---|
| 2 | 55.8 | 55.8 | 55.8 | 89.0 | 93.3 | 94.1 | 93.8 | 94.4 | 93.4 |
| 4 | 55.8 | 55.8 | 91.0 | 93.0 | 93.5 | 93.9 | 94.4 | 93.8 | 90.6 |
| 8 | 55.8 | 55.8 | 89.4 | 93.4 | 93.8 | 94.5 | 94.2 | 94.3 | 88.9 |
| 16 | 55.8 | 55.8 | 92.6 | 92.8 | 94.2 | 95.1 | 94.7 | 93.5 | 55.8 |
| 32 | 55.8 | 91.0 | 92.6 | 93.8 | 94.2 | 94.0 | 94.7 | 55.8 | 55.8 |
| 64 | 55.8 | 92.7 | 93.5 | 94.3 | 94.8 | 94.5 | 93.8 | 55.8 | 55.8 |
| 128 | 91.8 | 93.1 | 94.2 | 94.3 | 94.6 | 94.6 | 55.8 | 55.8 | 55.8 |

### DeBERTa v3 cLA PAWS

| Rank/LR | 1e-6 | 1e-5.5 | 1e-5 | 1e-4.5 | 1e-4 | 1e-3.5 | 1e-3 | 1e-2.5 | 1e-2 |
|---|---|---|---|---|---|---|---|---|---|
| 2 | 55.8 | 55.8 | 55.8 | 90.5 | 92.5 | 94.0 | 93.8 | 93.7 | 55.8 |
| 4 | 55.8 | 55.8 | 89.3 | 92.7 | 92.9 | 94.3 | 93.9 | 93.7 | 55.8 |
| 8 | 55.8 | 55.8 | 89.7 | 92.6 | 92.9 | 94.3 | 93.9 | 93.7 | 55.8 |
| 16 | 55.8 | 55.8 | 91.5 | 93.0 | 93.7 | 94.5 | 94.1 | 55.8 | 55.8 |
| 32 | 55.8 | 89.5 | 91.6 | 93.4 | 93.8 | 94.2 | 94.7 | 55.8 | 55.8 |
| 64 | 55.8 | 90.0 | 93.2 | 93.8 | 93.9 | 94.3 | 93.5 | 55.8 | 55.8 |
| 128 | 87.2 | 92.5 | 93.6 | 93.9 | 94.7 | 94.2 | 55.8 | 55.8 | 55.8 |

### DeBERTa v3 $c^3$LA PAWS

| Rank/LR | 1e-6 | 1e-5.5 | 1e-5 | 1e-4.5 | 1e-4 | 1e-3.5 | 1e-3 | 1e-2.5 | 1e-2 |
|---|---|---|---|---|---|---|---|---|---|
| 2 | 55.8 | 55.8 | 55.8 | 91.4 | 93.3 | 93.8 | 93.9 | 93.9 | 55.8 |
| 4 | 55.8 | 55.8 | 89.8 | 91.2 | 93.7 | 94.0 | 94.0 | 94.2 | 55.8 |
| 8 | 55.8 | 55.8 | 91.8 | 93.7 | 93.9 | 94.7 | 93.5 | 93.3 | 55.8 |
| 16 | 55.8 | 55.8 | 92.6 | 93.5 | 94.1 | 93.7 | 55.8 | 55.8 | 55.8 |
| 32 | 55.8 | 92.0 | 93.2 | 94.0 | 94.1 | 94.4 | 93.9 | 55.8 | 55.8 |
| 64 | 91.3 | 92.2 | 93.8 | 94.1 | 93.9 | 94.2 | 93.7 | 55.8 | 55.8 |
| 128 | 92.5 | 93.3 | 94.0 | 94.3 | 94.8 | 55.8 | 55.8 | 55.8 | 55.8 |

Table 7: Test accuracies obtained by fine-tuning ViT-Tiny on CIFAR-10 and OfficeHome over varying learning rates (columns), ranks (rows), and LoRA PEFT methods. We center our search at $1e^{-3}$. Consistent with the results of 6, the learning rate for all methods decreases with increasing rank. Chain methods and their non-chain counterparts produce the best results in similar learning rate ranges.

### ViT-Tiny LoRA CIFAR-10

| | 1e-5 | 1e-4.5 | 1e-4 | 1e-3.5 | 1e-3 | 1e-2.5 | 1e-2 | 1e-1.5 | 1e-1 |
|---|---|---|---|---|---|---|---|---|---|
| 2 | 89.08 | 90.91 | 92.85 | 92.92 | 93.56 | 91.76 | 87.13 | 49.75 | 10.70 |
| 4 | 89.38 | 91.55 | 93.49 | 94.50 | 94.18 | 95.11 | 81.16 | 17.82 | 11.15 |
| 8 | 89.56 | 92.04 | 94.00 | 95.20 | 95.68 | 95.65 | 77.10 | 11.85 | 10.00 |
| 16 | 90.03 | 92.69 | 94.36 | 95.69 | 95.91 | 91.27 | 59.09 | 13.52 | 10.08 |
| 32 | 90.85 | 93.05 | 95.14 | 96.14 | 96.26 | 87.87 | 17.79 | 18.07 | 10.67 |
| 64 | 91.79 | 94.00 | 95.30 | 96.44 | 96.43 | 81.73 | 14.33 | 11.30 | 13.21 |
| 128 | 92.47 | 94.71 | 96.03 | 96.50 | 96.17 | 64.03 | 11.16 | 12.00 | 11.41 |

### ViT-Tiny CoLA CIFAR-10

| | 1e-5 | 1e-4.5 | 1e-4 | 1e-3.5 | 1e-3 | 1e-2.5 | 1e-2 | 1e-1.5 | 1e-1 |
|---|---|---|---|---|---|---|---|---|---|
| 2 | 87.90 | 90.43 | 92.13 | 93.79 | 93.77 | 93.57 | 87.13 | 20.52 | 12.61 |
| 4 | 88.43 | 90.77 | 92.73 | 94.30 | 95.21 | 94.49 | 83.74 | 17.82 | 11.15 |
| 8 | 88.96 | 91.26 | 93.37 | 94.95 | 95.39 | 94.74 | 77.09 | 11.85 | 10.00 |
| 16 | 89.50 | 92.00 | 93.86 | 95.26 | 95.72 | 91.27 | 62.12 | 15.42 | 11.25 |
| 32 | 90.02 | 92.63 | 94.54 | 95.72 | 95.96 | 87.87 | 19.10 | 18.07 | 12.41 |
| 64 | 91.18 | 93.51 | 95.13 | 96.07 | 96.01 | 81.73 | 14.33 | 17.77 | 13.23 |
| 128 | 92.06 | 94.20 | 95.56 | 96.17 | 95.54 | 65.78 | 17.45 | 10.30 | 11.03 |

### ViT-Tiny Asym CIFAR-10

| | 1e-5 | 1e-4.5 | 1e-4 | 1e-3.5 | 1e-3 | 1e-2.5 | 1e-2 | 1e-1.5 | 1e-1 |
|---|---|---|---|---|---|---|---|---|---|
| 2 | 85.34 | 89.63 | 90.86 | 91.64 | 92.03 | 91.88 | 90.85 | 90.18 | 80.81 |
| 4 | 86.82 | 90.29 | 91.66 | 92.58 | 93.32 | 92.73 | 91.63 | 87.88 | 78.91 |
| 8 | 88.34 | 90.87 | 92.35 | 93.34 | 93.71 | 93.81 | 93.74 | 86.88 | 64.43 |
| 16 | 89.23 | 91.61 | 93.18 | 94.23 | 94.65 | 95.08 | 90.39 | 82.52 | 50.44 |
| 32 | 90.12 | 92.17 | 93.81 | 95.20 | 95.15 | 95.36 | 93.94 | 73.55 | 34.13 |
| 64 | 91.20 | 93.11 | 94.66 | 95.77 | 96.08 | 92.99 | 92.18 | 53.07 | 24.52 |
| 128 | 92.27 | 94.03 | 95.36 | 96.09 | 94.56 | 94.97 | 69.81 | 27.77 | 16.58 |

### ViT-Tiny RAC CIFAR-10

| | 1e-5 | 1e-4.5 | 1e-4 | 1e-3.5 | 1e-3 | 1e-2.5 | 1e-2 | 1e-1.5 | 1e-1 |
|---|---|---|---|---|---|---|---|---|---|
| 2 | 85.61 | 89.68 | 90.98 | 91.87 | 92.72 | 91.94 | 91.47 | 89.51 | 80.23 |
| 4 | 86.64 | 90.35 | 91.89 | 92.96 | 93.71 | 93.46 | 93.49 | 87.88 | 77.56 |
| 8 | 88.26 | 90.87 | 92.40 | 93.65 | 94.09 | 94.33 | 90.69 | 86.88 | 64.43 |
| 16 | 89.26 | 91.75 | 93.26 | 94.40 | 95.31 | 94.24 | 89.56 | 82.52 | 50.44 |
| 32 | 90.27 | 92.28 | 94.05 | 95.43 | 95.54 | 95.68 | 93.14 | 73.55 | 24.92 |
| 64 | 91.12 | 93.20 | 94.86 | 95.91 | 94.78 | 95.35 | 82.86 | 53.07 | 24.52 |
| 128 | 92.19 | 94.03 | 95.56 | 96.11 | 96.10 | 94.21 | 69.81 | 27.77 | 16.58 |

### ViT-Tiny cLA CIFAR-10

| | 1e-5 | 1e-4.5 | 1e-4 | 1e-3.5 | 1e-3 | 1e-2.5 | 1e-2 | 1e-1.5 | 1e-1 |
|---|---|---|---|---|---|---|---|---|---|
| 2 | 87.30 | 89.65 | 91.13 | 92.12 | 92.49 | 91.70 | 88.92 | 82.43 | 51.06 |
| 4 | 88.06 | 90.47 | 91.91 | 92.00 | 93.45 | 92.00 | 88.78 | 73.27 | 11.47 |
| 8 | 89.14 | 91.38 | 93.13 | 93.36 | 93.22 | 90.72 | 85.78 | 68.40 | 13.39 |
| 16 | 90.43 | 92.27 | 93.91 | 94.46 | 94.53 | 94.11 | 78.83 | 44.03 | 17.11 |
| 32 | 91.32 | 93.37 | 94.91 | 95.63 | 95.33 | 89.55 | 71.95 | 30.90 | 19.23 |
| 64 | 92.50 | 94.17 | 95.82 | 96.26 | 95.54 | 83.63 | 50.91 | 21.61 | 15.50 |
| 128 | 93.86 | 95.39 | 96.50 | 96.45 | 94.85 | 75.52 | 28.65 | 19.95 | 31.06 |

### ViT-Tiny $c^3$LA CIFAR-10

| | 1e-5 | 1e-4.5 | 1e-4 | 1e-3.5 | 1e-3 | 1e-2.5 | 1e-2 | 1e-1.5 | 1e-1 |
|---|---|---|---|---|---|---|---|---|---|
| 2 | 87.29 | 89.83 | 91.49 | 92.48 | 93.04 | 91.35 | 88.65 | 80.39 | 53.28 |
| 4 | 88.57 | 90.76 | 92.50 | 93.36 | 93.98 | 91.55 | 87.25 | 73.27 | 11.47 |
| 8 | 89.69 | 91.84 | 93.52 | 94.68 | 94.80 | 90.72 | 85.78 | 68.40 | 27.84 |
| 16 | 90.63 | 92.69 | 94.33 | 95.35 | 95.30 | 90.04 | 81.13 | 44.03 | 17.11 |
| 32 | 91.39 | 93.68 | 95.12 | 95.57 | 95.30 | 89.55 | 71.95 | 21.94 | 17.60 |
| 64 | 92.86 | 94.71 | 95.92 | 96.41 | 95.07 | 83.63 | 41.60 | 21.61 | 14.03 |
| 128 | 93.83 | 95.38 | 96.43 | 96.22 | 94.89 | 75.52 | 27.84 | 19.95 | 14.48 |

### ViT-Tiny LoRA OfficeHome

| | 1e-5 | 1e-4.5 | 1e-4 | 1e-3.5 | 1e-3 | 1e-2.5 | 1e-2 | 1e-1.5 | 1e-1 |
|---|---|---|---|---|---|---|---|---|---|
| 2 | 47.33 | 65.03 | 70.80 | 75.55 | 77.81 | 75.93 | 77.08 | 40.57 | 1.80 |
| 4 | 48.57 | 64.92 | 71.36 | 75.33 | 77.85 | 78.50 | 77.55 | 26.51 | 2.01 |
| 8 | 49.42 | 64.81 | 72.08 | 75.93 | 79.39 | 78.88 | 59.09 | 4.28 | 2.61 |
| 16 | 50.41 | 65.33 | 72.30 | 77.21 | 79.69 | 49.17 | 1.97 | 4.53 | |
| 32 | 50.53 | 65.50 | 73.54 | 79.09 | 79.56 | 66.01 | 36.43 | 2.44 | 2.05 |
| 64 | 51.86 | 66.35 | 74.99 | 78.97 | 79.86 | 61.48 | 10.77 | 1.75 | 2.09 |
| 128 | 54.68 | 66.82 | 76.70 | 79.91 | 80.33 | 53.53 | 2.78 | 1.75 | 2.91 |

### ViT-Tiny CoLA OfficeHome

| | 1e-5 | 1e-4.5 | 1e-4 | 1e-3.5 | 1e-3 | 1e-2.5 | 1e-2 | 1e-1.5 | 1e-1 |
|---|---|---|---|---|---|---|---|---|---|
| 2 | 45.28 | 64.64 | 70.50 | 74.48 | 76.96 | 76.70 | 74.05 | 40.57 | 2.01 |
| 4 | 46.43 | 64.94 | 70.63 | 74.39 | 76.87 | 76.61 | 72.60 | 26.51 | 2.01 |
| 8 | 47.71 | 65.11 | 70.97 | 75.12 | 77.73 | 76.96 | 59.09 | 3.04 | 5.77 |
| 16 | 49.47 | 64.51 | 71.40 | 75.63 | 78.71 | 77.68 | 49.17 | 1.97 | 2.22 |
| 32 | 50.32 | 65.20 | 72.12 | 77.38 | 80.38 | 66.01 | 36.43 | 2.44 | 2.05 |
| 64 | 51.05 | 65.63 | 73.79 | 78.32 | 79.35 | 61.48 | 10.77 | 1.75 | 2.69 |
| 128 | 52.29 | 66.35 | 75.29 | 79.48 | 79.52 | 52.29 | 2.78 | 1.75 | 4.10 |

### ViT-Tiny Asym OfficeHome

| | 1e-5 | 1e-4.5 | 1e-4 | 1e-3.5 | 1e-3 | 1e-2.5 | 1e-2 | 1e-1.5 | 1e-1 |
|---|---|---|---|---|---|---|---|---|---|
| 2 | 43.91 | 63.06 | 70.20 | 73.45 | 74.65 | 74.48 | 74.22 | 75.29 | 74.09 |
| 4 | 44.72 | 63.83 | 70.59 | 73.71 | 75.76 | 73.92 | 76.14 | 75.37 | 51.69 |
| 8 | 45.79 | 64.47 | 71.31 | 74.65 | 75.93 | 75.50 | 75.84 | 75.72 | 40.49 |
| 16 | 47.20 | 65.80 | 72.21 | 75.12 | 77.13 | 75.67 | 76.87 | 54.77 | 21.42 |
| 32 | 49.68 | 66.27 | 72.47 | 76.66 | 78.50 | 77.94 | 77.04 | 45.28 | 13.85 |
| 64 | 51.52 | 67.04 | 74.13 | 77.81 | 79.31 | 78.37 | 57.46 | 23.56 | 8.85 |
| 128 | 53.53 | 68.06 | 75.25 | 79.35 | 80.72 | 77.85 | 46.81 | 10.65 | 2.05 |

### ViT-Tiny RAC OfficeHome

| | 1e-5 | 1e-4.5 | 1e-4 | 1e-3.5 | 1e-3 | 1e-2.5 | 1e-2 | 1e-1.5 | 1e-1 |
|---|---|---|---|---|---|---|---|---|---|
| 2 | 44.38 | 63.75 | 70.54 | 73.54 | 75.29 | 76.14 | 75.16 | 70.07 | 54.85 |
| 4 | 44.76 | 64.04 | 70.71 | 74.01 | 76.36 | 74.22 | 72.85 | 66.44 | 51.69 |
| 8 | 45.83 | 65.07 | 71.65 | 74.78 | 76.31 | 77.04 | 75.97 | 64.73 | 40.49 |
| 16 | 47.29 | 65.71 | 72.55 | 75.29 | 77.94 | 77.47 | 75.46 | 67.38 | 21.42 |
| 32 | 49.64 | 66.10 | 72.60 | 77.47 | 79.26 | 78.79 | 75.07 | 45.28 | 13.85 |
| 64 | 51.60 | 67.12 | 74.39 | 78.32 | 79.91 | 75.33 | 57.46 | 23.56 | 9.49 |
| 128 | 53.53 | 67.76 | 75.72 | 79.26 | 80.63 | 76.74 | 46.81 | 10.65 | 2.05 |

### ViT-Tiny cLA OfficeHome

| | 1e-5 | 1e-4.5 | 1e-4 | 1e-3.5 | 1e-3 | 1e-2.5 | 1e-2 | 1e-1.5 | 1e-1 |
|---|---|---|---|---|---|---|---|---|---|
| 2 | 44.04 | 64.04 | 70.12 | 73.45 | 75.89 | 75.72 | 74.35 | 73.19 | 30.14 |
| 4 | 46.09 | 64.86 | 70.84 | 74.90 | 76.53 | 75.37 | 74.99 | 54.81 | 2.01 |
| 8 | 47.50 | 65.16 | 72.12 | 75.16 | 76.57 | 76.87 | 75.25 | 45.10 | 2.86 |
| 16 | 50.75 | 65.63 | 72.98 | 77.00 | 78.37 | 77.17 | 79.30 | 36.55 | 3.42 |
| 32 | 52.93 | 66.99 | 74.39 | 77.85 | 76.83 | 76.83 | 51.13 | 12.61 | 2.35 |
| 64 | 55.96 | 68.53 | 75.67 | 79.14 | 79.26 | 63.06 | 35.49 | 4.02 | 2.69 |
| 128 | 61.22 | 71.95 | 77.34 | 79.78 | 77.77 | 53.91 | 17.44 | 3.42 | 3.42 |

### ViT-Tiny $c^3$LA OfficeHome

| | 1e-5 | 1e-4.5 | 1e-4 | 1e-3.5 | 1e-3 | 1e-2.5 | 1e-2 | 1e-1.5 | 1e-1 |
|---|---|---|---|---|---|---|---|---|---|
| 2 | 45.02 | 64.51 | 70.29 | 73.88 | 76.36 | 75.93 | 73.07 | 57.46 | 30.14 |
| 4 | 46.30 | 64.94 | 70.80 | 74.31 | 77.17 | 76.19 | 66.65 | 54.81 | 2.01 |
| 8 | 49.21 | 65.37 | 72.68 | 75.93 | 77.51 | 72.60 | 64.73 | 45.10 | 3.04 |
| 16 | 51.13 | 66.44 | 73.15 | 77.55 | 78.71 | 71.23 | 59.30 | 36.55 | 2.48 |
| 32 | 53.31 | 67.55 | 74.22 | 79.26 | 79.22 | 67.38 | 51.13 | 12.01 | 2.35 |
| 64 | 56.18 | 69.22 | 76.06 | 79.52 | 78.24 | 63.06 | 35.49 | 6.50 | 3.59 |
| 128 | 61.18 | 72.00 | 77.38 | 79.91 | 78.45 | 53.91 | 17.44 | 8.38 | 3.42 |

Table 8: Test accuracies obtained by fine-tuning DeBERTa v3 on MRPC and TREC-50 over varying scaling factors (columns), ranks (rows), and LoRA PEFT methods. The standard baseline $2r$ often was the best, and asymmetric methods preferred higher scaling factors.

### DeBERTa v3 LoRA MRPC

| Rank/$\alpha$ | $\frac{r}{4}$ | $\frac{r}{2}$ | $r$ | $2r$ | $4r$ |
|---|---|---|---|---|---|
| 4 | 87.2 | 88.3 | 88.5 | 88.1 | 87.4 |
| 8 | 86.9 | 86.1 | 89.2 | 87.0 | 66.5 |
| 16 | 87.8 | 88.9 | 89.1 | 66.5 | 66.5 |

### DeBERTa v3 CoLA MRPC

| Rank/$\alpha$ | $\frac{r}{4}$ | $\frac{r}{2}$ | $r$ | $2r$ | $4r$ |
|---|---|---|---|---|---|
| 4 | 87.8 | 88.9 | 88.5 | 89.2 | 87.1 |
| 8 | 89.6 | 87.4 | 88.7 | 87.2 | 86.3 |
| 16 | 89.2 | 87.6 | 86.9 | 87.2 | 66.5 |

### DeBERTa v3 Asym MRPC

| Rank/$\alpha$ | $\frac{r}{4}$ | $\frac{r}{2}$ | $r$ | $2r$ | $4r$ |
|---|---|---|---|---|---|
| 4 | 75.5 | 79.9 | 80.4 | 85.0 | 84.2 |
| 8 | 76.7 | 82.1 | 83.6 | 86.1 | 86.9 |
| 16 | 79.2 | 81.4 | 84.8 | 84.8 | 86.1 |

### DeBERTa v3 RAC MRPC

| Rank/$\alpha$ | $\frac{r}{4}$ | $\frac{r}{2}$ | $r$ | $2r$ | $4r$ |
|---|---|---|---|---|---|
| 4 | 75.6 | 79.4 | 82.2 | 85.0 | 85.7 |
| 8 | 77.8 | 81.6 | 84.6 | 85.7 | 87.2 |
| 16 | 80.4 | 84.5 | 85.0 | 85.6 | 87.0 |

### DeBERTa v3 cLA MRPC

| Rank/$\alpha$ | $\frac{r}{4}$ | $\frac{r}{2}$ | $r$ | $2r$ | $4r$ |
|---|---|---|---|---|---|
| 4 | 86.2 | 86.0 | 86.3 | 86.4 | 86.4 |
| 8 | 86.6 | 84.8 | 85.4 | 85.5 | 85.9 |
| 16 | 86.8 | 86.9 | 86.2 | 86.2 | 86.4 |

### DeBERTa v3 $c^3$LA MRPC

| Rank/$\alpha$ | $\frac{r}{4}$ | $\frac{r}{2}$ | $r$ | $2r$ | $4r$ |
|---|---|---|---|---|---|
| 4 | 79.3 | 83.3 | 86.5 | 88.5 | 86.1 |
| 8 | 78.3 | 84.9 | 86.9 | 87.6 | 86.9 |
| 16 | 85.0 | 85.7 | 87.3 | 85.8 | 66.5 |

### DeBERTa v3 LoRA TREC-50

| Rank/$\alpha$ | $\frac{r}{4}$ | $\frac{r}{2}$ | $r$ | $2r$ | $4r$ |
|---|---|---|---|---|---|
| 4 | 88.9 | 89.7 | 90.7 | 83.1 | 90.3 |
| 8 | 88.7 | 90.7 | 91.3 | 85.3 | 75.4 |
| 16 | 91.1 | 91.5 | 90.7 | 88.5 | 10.9 |

### DeBERTa v3 CoLA TREC-50

| Rank/$\alpha$ | $\frac{r}{4}$ | $\frac{r}{2}$ | $r$ | $2r$ | $4r$ |
|---|---|---|---|---|---|
| 4 | 92.1 | 91.9 | 92.5 | 90.7 | 91.1 |
| 8 | 91.7 | 89.7 | 90.9 | 90.3 | 85.5 |
| 16 | 91.9 | 92.3 | 86.1 | 87.3 | 10.9 |

### DeBERTa v3 Asym TREC-50

| Rank/$\alpha$ | $\frac{r}{4}$ | $\frac{r}{2}$ | $r$ | $2r$ | $4r$ |
|---|---|---|---|---|---|
| 4 | 79.8 | 84.7 | 87.7 | 90.5 | 89.9 |
| 8 | 82.9 | 87.7 | 84.5 | 89.3 | 90.7 |
| 16 | 89.3 | 86.7 | 90.7 | 91.3 | 89.7 |

### DeBERTa v3 RAC TREC-50

| Rank/$\alpha$ | $\frac{r}{4}$ | $\frac{r}{2}$ | $r$ | $2r$ | $4r$ |
|---|---|---|---|---|---|
| 4 | 60.1 | 72.6 | 81.0 | 85.5 | 88.7 |
| 8 | 75.2 | 81.5 | 85.7 | 88.1 | 89.7 |
| 16 | 83.1 | 86.3 | 87.7 | 90.3 | 78.0 |

### DeBERTa v3 cLA TREC-50

| Rank/$\alpha$ | $\frac{r}{4}$ | $\frac{r}{2}$ | $r$ | $2r$ | $4r$ |
|---|---|---|---|---|---|
| 4 | 57.9 | 74.8 | 80.0 | 82.9 | 86.3 |
| 8 | 73.6 | 76.6 | 83.7 | 82.5 | 87.3 |
| 16 | 79.6 | 80.2 | 87.9 | 88.1 | 86.1 |

### DeBERTa v3 $c^3$LA TREC-50

| Rank/$\alpha$ | $\frac{r}{4}$ | $\frac{r}{2}$ | $r$ | $2r$ | $4r$ |
|---|---|---|---|---|---|
| 4 | 73.8 | 81.5 | 83.1 | 89.3 | 88.7 |
| 8 | 78.2 | 82.3 | 83.9 | 84.7 | 81.7 |
| 16 | 83.1 | 85.5 | 85.3 | 87.5 | 86.3 |

Table 9: Test accuracies obtained by fine-tuning ViT-Tiny on OfficeHome and CIFAR-10 over varying scaling factors (columns), ranks (rows), and LoRA PEFT methods. The standard baseline $2r$ often was the best, and asymmetric methods preferred higher scaling factors.

**ViT-Tiny LoRA CIFAR-10**

| Rank/$\alpha$ | $\frac{r}{4}$ | $\frac{r}{2}$ | r | 2r |
|---|---|---|---|---|
| 4 | 93.9 | 94.1 | 94.0 | 94.0 |
| 8 | 94.8 | 95.7 | 95.7 | 95.8 |
| 16 | 95.8 | 96.1 | 96.1 | 95.2 |

**ViT-Tiny CoLA CIFAR-10**

| Rank/$\alpha$ | $\frac{r}{4}$ | $\frac{r}{2}$ | r | 2r |
|---|---|---|---|---|
| 4 | 94.3 | 94.5 | 94.5 | 95.3 |
| 8 | 94.7 | 94.9 | 95.3 | 95.1 |
| 16 | 95.0 | 95.5 | 95.7 | 96.2 |

**ViT-Tiny Asym CIFAR-10**

| Rank/$\alpha$ | $\frac{r}{4}$ | $\frac{r}{2}$ | r | 2r |
|---|---|---|---|---|
| 4 | 91.7 | 92.2 | 92.8 | 92.4 |
| 8 | 92.6 | 93.1 | 93.7 | 94. |
| 16 | 93.1 | 94.0 | 94.4 | 94.4 |

**ViT-Tiny RAC CIFAR-10**

| Rank/$\alpha$ | $\frac{r}{4}$ | $\frac{r}{2}$ | r | 2r |
|---|---|---|---|---|
| 4 | 91.8 | 92.6 | 93.2 | 94.2 |
| 8 | 92.8 | 93.4 | 94.0 | 94.8 |
| 16 | 93.6 | 94.3 | 94.8 | 95.6 |

**ViT-Tiny cLA CIFAR-10**

| Rank/$\alpha$ | $\frac{r}{4}$ | $\frac{r}{2}$ | r | 2r |
|---|---|---|---|---|
| 4 | 92.0 | 92.6 | 93.1 | 93.5 |
| 8 | 93.4 | 93.4 | 93.5 | 93.4 |
| 16 | 94.3 | 94.5 | 94.5 | 94.5 |

**ViT-Tiny $c^3$LA CIFAR-10**

| Rank/$\alpha$ | $\frac{r}{4}$ | $\frac{r}{2}$ | r | 2r |
|---|---|---|---|---|
| 4 | 92.7 | 93.4 | 93.4 | 94.4 |
| 8 | 93.8 | 94.4 | 94.6 | 94.8 |
| 16 | 94.8 | 95.3 | 95.2 | 95.3 |

**ViT-Tiny LoRA OfficeHome**

| Rank/$\alpha$ | $\frac{r}{4}$ | $\frac{r}{2}$ | r | 2r |
|---|---|---|---|---|
| 4 | 76.8 | 77.1 | 77.8 | 77.9 |
| 8 | 76.9 | 77.9 | 78.5 | 79.4 |
| 16 | 77.9 | 78.4 | 79.2 | 79.4 |

**ViT-Tiny CoLA OfficeHome**

| Rank/$\alpha$ | $\frac{r}{4}$ | $\frac{r}{2}$ | r | 2r |
|---|---|---|---|---|
| 4 | 75.5 | 75.9 | 76.6 | 77.8 |
| 8 | 76.0 | 76.4 | 77.4 | 79.6 |
| 16 | 76.3 | 77.2 | 78.2 | 79.4 |

**ViT-Tiny Asym OfficeHome**

| Rank/$\alpha$ | $\frac{r}{4}$ | $\frac{r}{2}$ | r | 2r |
|---|---|---|---|---|
| 4 | 74.0 | 74.5 | 75.2 | 75.6 |
| 8 | 74.5 | 75.1 | 75.6 | 76.2 |
| 16 | 75.2 | 75.9 | 76.4 | 76.9 |

**ViT-Tiny RAC OfficeHome**

| Rank/$\alpha$ | $\frac{r}{4}$ | $\frac{r}{2}$ | r | 2r |
|---|---|---|---|---|
| 4 | 74.3 | 74.6 | 75.6 | 76.0 |
| 8 | 74.8 | 75.2 | 75.9 | 77.1 |
| 16 | 75.2 | 75.5 | 76.4 | 77.7 |

**ViT-Tiny cLA OfficeHome**

| Rank/$\alpha$ | $\frac{r}{4}$ | $\frac{r}{2}$ | r | 2r |
|---|---|---|---|---|
| 4 | 75.3 | 75.9 | 76.5 | 76.5 |
| 8 | 76.3 | 76.5 | 76.6 | 76.5 |
| 16 | 76.4 | 76.9 | 77.3 | 78.4 |

**ViT-Tiny $c^3$LA OfficeHome**

| Rank/$\alpha$ | $\frac{r}{4}$ | $\frac{r}{2}$ | r | 2r |
|---|---|---|---|---|
| 4 | 75.6 | 75.9 | 76.3 | 77.3 |
| 8 | 76.4 | 77.0 | 76.9 | 77.5 |
| 16 | 76.6 | 77.7 | 78.4 | 78.5 |

Table 10: Test accuracies obtained by fine tuning DeBERTa v3 on MRPC, CoLA, TREC-50 and RTE using chain LoRA methods CoLA, RAC, and $c^3$LA over varying ranks and chain reset frequencies. No clear correlation between optimal chain reset frequency and rank is observed.

**DeBERTa v3 MRPC**

| Variant | Rank | Chain Reset Frequency | | | | | |
|---|---|---|---|---|---|---|---|
| | | 1 | 2 | 5 | 10 | 15 | 20 |
| CoLA | 4 | 88.0 | 86.8 | 89.2 | 88.1 | 86.7 | 86.7 |
| | 8 | 87.8 | 88.0 | 87.2 | 87.2 | 86.7 | 87.2 |
| | 16 | 66.5 | 87.2 | 87.2 | 87.2 | 87.2 | 87.2 |
| RAC | 4 | 68.3 | 77.4 | 85.0 | 85.7 | 85.7 | 86.6 |
| | 8 | 68.1 | 82.0 | 85.7 | 86.4 | 85.7 | 85.6 |
| | 16 | 69.1 | 84.2 | 85.6 | 86.1 | 86.5 | 86.3 |
| $c^3$LA | 4 | 84.8 | 86.7 | 87.2 | 85.2 | 85.8 | 85.2 |
| | 8 | 85.2 | 87.7 | 86.6 | 86.7 | 85.3 | 86.9 |
| | 16 | 87.6 | 87.0 | 86.7 | 86.6 | 86.6 | 87.6 |

**DeBERTa v3 TREC-50**

| Variant | Rank | Chain Reset Frequency | | | | | |
|---|---|---|---|---|---|---|---|
| | | 1 | 2 | 5 | 10 | 15 | 20 |
| CoLA | 4 | 91.3 | 91.1 | 89.9 | 88.5 | 90.5 | 91.3 |
| | 8 | 92.7 | 91.1 | 85.3 | 10.9 | 92.7 | 90.5 |
| | 16 | 10.9 | 10.9 | 93.1 | 91.7 | 92.1 | 65.1 |
| RAC | 4 | 84.3 | 84.1 | 85.5 | 84.1 | 86.3 | 86.5 |
| | 8 | 88.3 | 88.5 | 88.1 | 88.7 | 87.7 | 88.9 |
| | 16 | 87.7 | 91.5 | 90.3 | 89.9 | 89.9 | 88.9 |
| $c^3$LA | 4 | 86.3 | 88.1 | 89.3 | 85.9 | 88.9 | 88.9 |
| | 8 | 86.1 | 89.3 | 84.7 | 83.7 | 86.1 | 90.7 |
| | 16 | 89.7 | 90.5 | 87.5 | 91.1 | 87.3 | 88.1 |

**DeBERTa v3 CoLA**

| Variant | Rank | Chain Reset Frequency | | | | | |
|---|---|---|---|---|---|---|---|
| | | 1 | 2 | 5 | 10 | 15 | 20 |
| CoLA | 4 | 86.9 | 86.5 | 86.2 | 86.6 | 87.1 | 86.7 |
| | 8 | 85.5 | 85.1 | 85.1 | 85.1 | 85.1 | 85.1 |
| | 16 | 84.2 | 69.1 | 69.1 | 69.1 | 69.1 | 69.1 |
| RAC | 4 | 87.0 | 86.7 | 87.7 | 87.4 | 88.0 | 87.7 |
| | 8 | 87.5 | 87.8 | 87.8 | 87.5 | 86.6 | 86.6 |
| | 16 | 86.7 | 86.9 | 87.3 | 87.0 | 87.0 | 87.6 |
| $c^3$LA | 4 | 86.4 | 86.6 | 86.1 | 85.8 | 86.0 | 86.3 |
| | 8 | 86.0 | 86.1 | 86.1 | 86.2 | 86.3 | 86.3 |
| | 16 | 86.2 | 85.7 | 86.3 | 85.6 | 85.4 | 86.7 |

**DeBERTa v3 RTE**

| Variant | Rank | Chain Reset Frequency | | | | | |
|---|---|---|---|---|---|---|---|
| | | 1 | 2 | 5 | 10 | 15 | 20 |
| CoLA | 4 | 82.9 | 84.4 | 85.1 | 83.7 | 85.4 | 86.2 |
| | 8 | 88.2 | 84.6 | 84.8 | 87.1 | 87.1 | 86.7 |
| | 16 | 85.1 | 52.6 | 81.4 | 84.8 | 84.3 | 73.5 |
| RAC | 4 | 82.3 | 83.0 | 81.6 | 82.1 | 82.4 | 82.4 |
| | 8 | 85.5 | 86.8 | 86.4 | 87.5 | 87.5 | 87.5 |
| | 16 | 84.2 | 84.4 | 84.1 | 83.5 | 83.7 | 84.3 |
| $c^3$LA | 4 | 79.0 | 77.9 | 72.6 | 71.9 | 74.0 | 72.4 |
| | 8 | 80.0 | 80.3 | 76.6 | 73.9 | 76.2 | 75.4 |
| | 16 | 85.0 | 82.5 | 83.6 | 83.0 | 82.9 | 82.4 |

Table 11: Test accuracies obtained by fine tuning ViT-Tiny on OfficeHome and CIFAR-10 using chain LoRA methods CoLA, RAC, and $c^3$LA over varying ranks and chain reset frequencies.

**ViT-Tiny OfficeHome**

| Variant | Rank | Chain Reset Frequency | | | | | |
|---|---|---|---|---|---|---|---|
| | | 1 | 2 | 5 | 10 | 15 | 20 |
| CoLA | 4 | 76.4 | 76.5 | 77.2 | 77.2 | 77.6 | 77.8 |
| | 8 | 77.6 | 77.1 | 78.3 | 77.3 | 78.7 | 79.6 |
| | 16 | 77.9 | 77.9 | 78.8 | 78.6 | 79.4 | 79.1 |
| RAC | 4 | 75.5 | 75.8 | 76.0 | 75.7 | 75.7 | 75.7 |
| | 8 | 77.1 | 76.1 | 76.3 | 76.6 | 76.1 | 76.4 |
| | 16 | 77.4 | 77.7 | 77.7 | 77.0 | 77.3 | 77.0 |
| $c^3$LA | 4 | 76.7 | 77.3 | 77.3 | 76.3 | 76.5 | 76.6 |
| | 8 | 77.5 | 76.9 | 77.2 | 76.7 | 76.8 | 76.8 |
| | 16 | 77.5 | 78.1 | 78.4 | 78.5 | 78.1 | 78.4 |

**ViT-Tiny CIFAR-10**

| Variant | Rank | Chain Reset Frequency | | | | | |
|---|---|---|---|---|---|---|---|
| | | 1 | 2 | 5 | 10 | 15 | 20 |
| CoLA | 4 | 94.5 | 94.8 | 94.7 | 95.3 | 94.0 | 94.0 |
| | 8 | 95.1 | 95.1 | 95.3 | 94.9 | 94.7 | 94.5 |
| | 16 | 95.5 | 95.5 | 95.7 | 96.0 | 96.0 | 96.2 |
| RAC | 4 | 94.2 | 94.0 | 94.0 | 93.4 | 92.4 | 92.5 |
| | 8 | 94.5 | 94.8 | 94.5 | 94.2 | 94.1 | 94.0 |
| | 16 | 95.6 | 95.2 | 95.3 | 95.1 | 95.0 | 94.3 |
| $c^3$LA | 4 | 94.4 | 94.2 | 94.0 | 93.9 | 92.7 | 92.7 |
| | 8 | 93.6 | 94.2 | 94.8 | 94.5 | 93.4 | 93.4 |
| | 16 | 94.0 | 93.6 | 95.3 | 95.1 | 94.8 | 95.0 |

Table 12: Total FLOPs per epoch to fine-tune RoBERTa-Base/Large (CoLA/MRPC), GPT2-Small (E2E), and ViT-Tiny/Base (OfficeHome/CIFAR-10) using FFT, LoRA, Asymmetric LoRA, and a naive sparse implementation. All models use rank $r = 16$. In each row, the best value is **bold**.

| Model | Dataset | Flops per epoch | | | |
|---|---|---|---|---|---|
| | | FFT | LoRA | Asym | Naive Sparse |
| RoBERTa-Base | MRPC | $1.1 \times 10^{14}$ | $7.5 \times 10^{13}$ | $7.5 \times 10^{13}$ | $\mathbf{7.4 \times 10^{13}}$ |
| | CoLA | $6.8 \times 10^{13}$ | $4.6 \times 10^{13}$ | $4.6 \times 10^{13}$ | $\mathbf{4.5 \times 10^{13}}$ |
| RoBERTa-Large | MRPC | $3.9 \times 10^{14}$ | $\mathbf{2.6 \times 10^{14}}$ | $\mathbf{2.6 \times 10^{14}}$ | $\mathbf{2.6 \times 10^{14}}$ |
| | CoLA | $2.4 \times 10^{14}$ | $\mathbf{1.6 \times 10^{14}}$ | $\mathbf{1.6 \times 10^{14}}$ | $\mathbf{1.6 \times 10^{14}}$ |
| ViT-Tiny | OfficeHome | $6.8 \times 10^{13}$ | $4.8 \times 10^{13}$ | $4.7 \times 10^{13}$ | $\mathbf{4.6 \times 10^{13}}$ |
| | CIFAR-10 | $3.0 \times 10^{14}$ | $2.1 \times 10^{14}$ | $2.1 \times 10^{14}$ | $\mathbf{2.0 \times 10^{14}}$ |
| ViT-Base | OfficeHome | $1.1 \times 10^{15}$ | $7.4 \times 10^{14}$ | $7.4 \times 10^{14}$ | $\mathbf{7.3 \times 10^{14}}$ |
| | CIFAR-10 | $4.8 \times 10^{15}$ | $\mathbf{3.2 \times 10^{15}}$ | $\mathbf{3.2 \times 10^{15}}$ | $\mathbf{3.2 \times 10^{15}}$ |

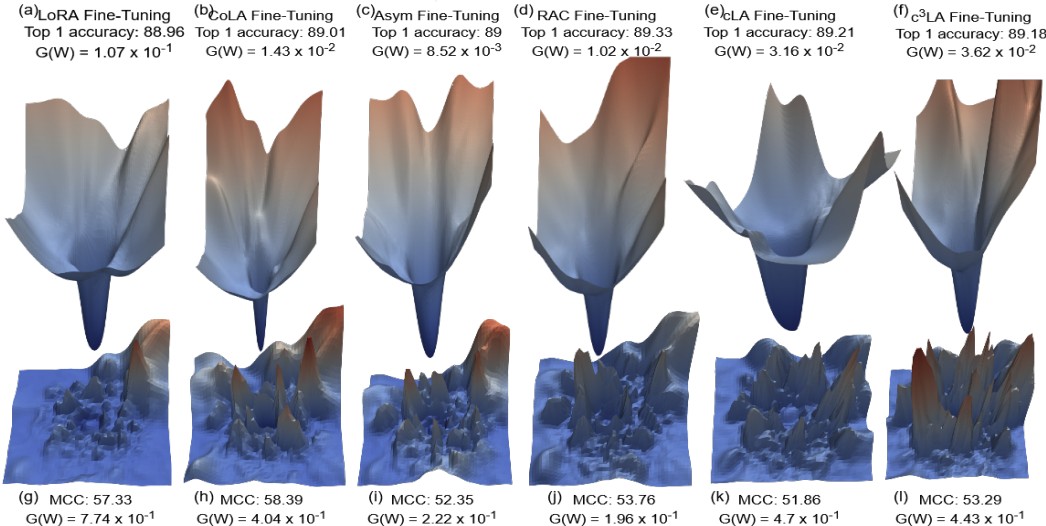

(a) LoRA Fine-Tuning — Top 1 accuracy: 88.96 — $G(W) = 1.07 \times 10^{-1}$
(b) CoLA Fine-Tuning — Top 1 accuracy: 89.01 — $G(W) = 1.43 \times 10^{-2}$
(c) Asym Fine-Tuning — Top 1 accuracy: 89 — $G(W) = 8.52 \times 10^{-3}$
(d) RAC Fine-Tuning — Top 1 accuracy: 89.33 — $G(W) = 1.02 \times 10^{-2}$
(e) CLA Fine-Tuning — Top 1 accuracy: 89.21 — $G(W) = 3.16 \times 10^{-2}$
(f) c³LA Fine-Tuning — Top 1 accuracy: 89.18 — $G(W) = 3.62 \times 10^{-2}$

(g) MCC: 57.33 — $G(W) = 7.74 \times 10^{-1}$
(h) MCC: 58.39 — $G(W) = 4.04 \times 10^{-1}$
(i) MCC: 52.35 — $G(W) = 2.22 \times 10^{-1}$
(j) MCC: 53.76 — $G(W) = 1.96 \times 10^{-1}$
(k) MCC: 51.86 — $G(W) = 4.7 \times 10^{-1}$
(l) MCC: 53.29 — $G(W) = 4.43 \times 10^{-1}$

Figure 6: 3D loss landscapes of ViT-Base (11) pretrained on ImageNet-1K (7) and fine-tuned on Office-Home (55) (top) and RoBERTa-Base (35) pretrained on a corpus of English text fine-tuned on CoLA (56) (bottom) using the non-chain then chain variants of each LoRA method. The chain variants consistently produce sharper landscapes than the non-chain variants. In asymmetric LoRA methods, this often correlates to worse generalizability, but not in symmetric methods where $B, A$ are both trained as shown in 15.

**Comparison between using random or PCA directions.** To understand the differences between the loss landscapes of the models in the PCA directions compared to random directions, we plotted the loss landscape of ViT-Base fine-tuned on CIFAR-10 in both PCA directions (top) and random directions (bottom) in Figure 5. For random directions, the FFT landscape is substantially smoother; this is consistent with (29), but this is inconsistent with the loss landscapes of RoBERTa-Base with random direction in Figure 6, where chain methods produce spikier landscapes with no substantial change in generalizability.

**2D landscapes.** The initial setup is identical to the 3D landscape. We obtain the same principal directions and plot the same function. For 2D landscapes, when generating our $\alpha, \beta$ grid of values, we uniformly distribute over $[-m, m] \times [-m, m]$ where $m$ is chosen to ensure the optimizer trajectory (blue arrows) is entirely contained in the image. As shown in Figure 7, chain methods have more diverse loss landscapes than their non-chain counterparts due to their overall update to the pre-trained weights having a higher effective rank (57).

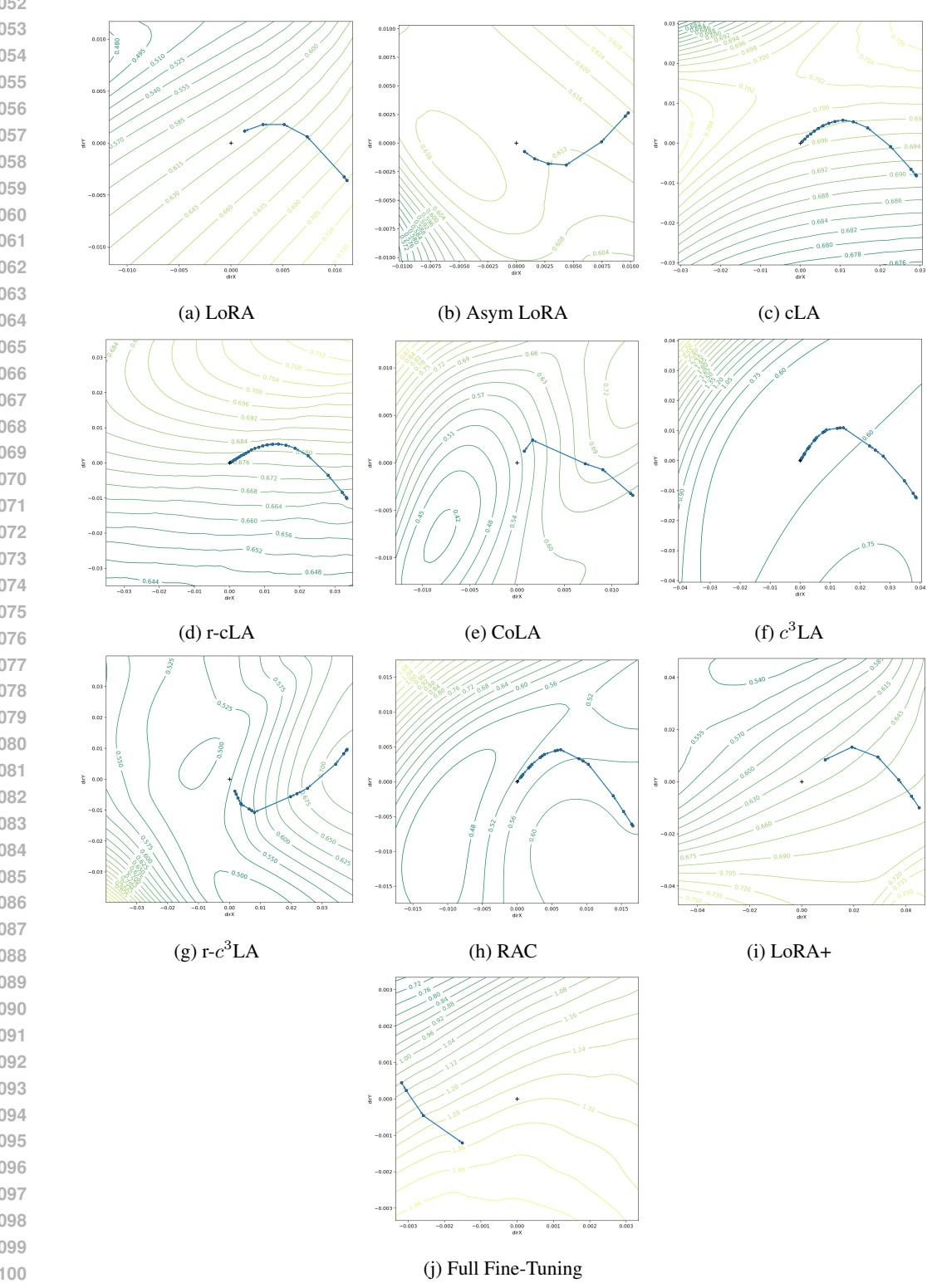

(a) LoRA

(b) Asym LoRA

(c) cLA

(d) r-cLA

(e) CoLA

(f) $c^3$LA

(g) r-$c^3$LA

(h) RAC

(i) LoRA+

(j) Full Fine-Tuning

Figure 7: 2D loss landscapes of RoBERTa-Base fine-tuned on CoLA for FFT and many PEFT LoRA methods. The axes dirX and dirY are the constants we scale the top two PCA components of the weight displacement matrix with. The range was chosen to contain the entire gradient path. The top row is the non-chain variant of the bottom row, save for the last column. The center is marked with a cross for visibility; the arrows indicate the direction of the model's updates.

Table 13: Average per-layer count of intruder dimensions for given $\epsilon$-thresholds for RoBERTa-Base fine-tuned on CoLA at 25%, 50%, 75%, and 100% points of training.

| | $\varepsilon = 0.4$ | | | | $\varepsilon = 0.8$ | | | |
|---|---|---|---|---|---|---|---|---|
| Method | 25% | 50% | 75% | 100% | 25% | 50% | 75% | 100% |
| FFT | 0.2 | 0.4 | 0.6 | 1.0 | 47.8 | 93.3 | 110.5 | 128.4 |
| LoRA | 0.3 | 1.4 | 1.9 | 3.0 | 73.0 | 112.9 | 141.0 | 162.8 |
| CoLA | 1.4 | 4.2 | 5.1 | 5.9 | 129.0 | 156.5 | 164.4 | 168.5 |
| Asym | 1.1 | 3.2 | 4.2 | 4.4 | 81.2 | 106.8 | 122.7 | 126.7 |
| RAC | 1.2 | 3.4 | 5.5 | 5.3 | 142.5 | 167.3 | 177.6 | 183.6 |
| cLA | 3.0 | 6.7 | 8.2 | 8.5 | 261.1 | 318.0 | 345.0 | 356.7 |
| r-cLA | 2.5 | 6.6 | 9.0 | 9.1 | 257.2 | 329.1 | 364.2 | 376.9 |
| $c^3$LA | 9.3 | 18.2 | 21.2 | 23.2 | 392.6 | 433.4 | 444.3 | 446.6 |
| r-$c^3$LA | 5.8 | 8.8 | 14.1 | 14.6 | 383.2 | 409.7 | 433.9 | 437.9 |

Table 14: Average per-layer count of intruder dimensions for given $\epsilon$-thresholds for ViT-Base fine-tuned on CIFAR-10 at 25%, 50%, 75%, and 100% points of training completion.

| | $\varepsilon = 0.4$ | | | | $\varepsilon = 0.8$ | | | |
|---|---|---|---|---|---|---|---|---|
| Method | 25% | 50% | 75% | 100% | 25% | 50% | 75% | 100% |
| FFT | 517.1 | 630.2 | 646.2 | 648.1 | 745.2 | 750.9 | 751.7 | 751.7 |
| LoRA | 133.2 | 230.3 | 254.0 | 258.4 | 718.3 | 726.5 | 728.0 | 728.1 |
| CoLA | 147.2 | 393.5 | 550.2 | 582.5 | 716.6 | 731.3 | 734.0 | 734.2 |
| Asym | 41.0 | 105.0 | 152.0 | 168.8 | 653.2 | 686.6 | 699.8 | 704.1 |
| RAC | 176.4 | 447.3 | 502.1 | 534.8 | 696.9 | 715.5 | 717.8 | 718.4 |
| cLA | 199.3 | 276.3 | 290.6 | 294.4 | 709.5 | 720.7 | 722.2 | 722.7 |
| r-cLA | 215.7 | 289.0 | 306.2 | 315.2 | 707.4 | 720.2 | 723.4 | 724.6 |
| $c^3$LA | 340.8 | 552.0 | 640.6 | 651.1 | 716.0 | 726.4 | 729.8 | 730.4 |
| r-$c^3$LA | 416.0 | 621.0 | 645.3 | 660.6 | 720.2 | 729.8 | 730.5 | 731.3 |

### D.4.2 INTRUDER DIMENSION IMPLEMENTATION

Given the pretrained and fine-tuned models, $\mathbf{W}_0$ and $\mathbf{W}_0 + \Delta\mathbf{W}$ we find intruder dimensions as follows: first, we decompose each layer of $\mathbf{W}_0$ and $\mathbf{W}_0 + \Delta\mathbf{W}$ into their corresponding SVDs, $U^i \Sigma^i V^{i^T}_{(\mathbf{W}_0)^i}$ and $U^i \Sigma^i V^{i^T}_{(\mathbf{W}_0 + \Delta\mathbf{W})^i}, i \in [L]$, respectively. Then, given a threshold $\varepsilon \in (0,1)$, a singular vector $u^{j,i}_{(\mathbf{W}_0 + \Delta\mathbf{W})}$ in $U^i_{(\mathbf{W}_0 + \Delta\mathbf{W})}$ is an intruder dimension if for all $u^{k,i}_{(\mathbf{W}_0)}$ in $U^i_{(\mathbf{W}_0)}$, the expression, $\frac{|\langle u^{j,i}_{(\mathbf{W}_0 + \Delta\mathbf{W})}, u^{k,i}_{(\mathbf{W}_0)}\rangle|}{\|u^{j,i}_{(\mathbf{W}_0 + \Delta\mathbf{W})}\|\|u^{k,i}_{(\mathbf{W}_0)}\|}| < \varepsilon$. For $\varepsilon$ small enough, this indicates the vector $u^{j,i}_{(\mathbf{W}_0 + \Delta\mathbf{W})}$ is almost orthogonal to all vectors in $U^i_{(\mathbf{W}_0)}$. We denote these vectors as *intruder dimensions*.

### D.5 GENERALIZATION ERROR—CONTINUED

Let $\mathcal{X} \times \mathcal{Y}$ be our input space and label space with $\nu$ distribution of pairs $(x, y) \in \mathcal{X} \times \mathcal{Y}$, our dataset $N = \{(x_1, y_1), ..., (x_n, y_n)\}$ where each $(x_i, y_i)$ is i.i.d. from $\nu$ distribution of $\mathcal{X} \times \mathcal{Y}$, thus the distribution over our dataset does not represent the true distribution of input-output pairs from our instance space. Let $\mathcal{H}$ be our hypothesis space, where $w \in \mathcal{H}; w(x_i) = \hat{y}_i$ thus, we are concerned with how accurately $w$ can adapt to the true distribution $\nu$ of $\mathcal{X} \times \mathcal{Y}$. This can be addressed by the generalization error of our hypothesis $w \in \mathcal{H}$ given our loss function $\ell$. The true risk of $w$ over $\mathcal{X} \times \mathcal{Y}$ given $\ell$ is $\mathcal{L}_{\text{global}}(w) := \mathbb{E}_{\mathcal{X},\mathcal{Y}}[\ell(w(x), y)] = \int_{\mathcal{X} \times \mathcal{Y}} \ell(w(x), y) d\nu$, while empirical risk is $\mathcal{L} := \frac{1}{n} \sum^n \ell(w(x_i), y_i); (x_i, y_i) \in N$. Let $M$ denote the full dataset, where $M = N \cup T$, $N$ being the train dataset, and $T$ being the test dataset. In practice, the empirical risk can be computed based on $N$, and the test dataset, $T$, can be used to show how well the model has generalized. $N$ and $T$ are independent samples from $\nu$; their distributions approximate $\nu$ but differ due to random and finite sampling. Although $\mathcal{L}_{\text{test}} - \mathcal{L}_{\text{train}}$ is not a true testament for calculating the generalization error of a

Table 15: Generalization error approximations (test-loss minus train-loss) on the past (FFT, LoRA), the present (CoLA, Asymm, RAC, LoRA+), and the future (cLA, $c^3$LA, r-cLA, r-$c^3$LA) fine tuning methods over various models and datasets. For the dataset CoLA we report the Matthews Correlation Coefficient and test accuracy otherwise. The color green indicates the best result for each particular model and dataset combination, red is the second best result and blue the third.

| Model | Dataset | The Past | | The Present | | | | The Future | | | |
|---|---|---|---|---|---|---|---|---|---|---|---|
| | | FFT | LoRA | CoLA | Asymm | RAC | LoRA+ | cLA | $c^3$LA | r-cLA | r-$c^3$LA |
| ViT-Tiny (11) | OfficeHome | $4.85e^{-1}$ | $6.96e^{-2}$ | $9.55e^{-3}$ | $7.22e^{-2}$ | $6.17e^{-2}$ | $7.39e^{-2}$ | $1.98e^{-2}$ | $3.40e^{-2}$ | $2.16e^{-2}$ | $3.51e^{-2}$ |
| | CIFAR-10 | $1.42e^{-1}$ | $2.64e^{-1}$ | $2.87e^{-1}$ | $3.36e^{-1}$ | $3.18e^{-1}$ | $2.80e^{-1}$ | $3.13e^{-1}$ | $3.03e^{-1}$ | $3.12e^{-1}$ | $2.92e^{-1}$ |
| ViT-Base (11) | OfficeHome | $3.66e^{-1}$ | $1.07e^{-1}$ | $1.43e^{-2}$ | $8.52e^{-3}$ | $1.02e^{-2}$ | $1.41e^{-1}$ | $3.16e^{-2}$ | $3.62e^{-2}$ | $5.53e^{-2}$ | $3.00e^{-2}$ |
| | CIFAR-10 | $9.98e^{-2}$ | $1.92e^{-1}$ | $2.21e^{-1}$ | $2.38e^{-1}$ | $2.30e^{-1}$ | $1.84e^{-1}$ | $2.33e^{-1}$ | $2.34e^{-1}$ | $2.26e^{-1}$ | $2.15e^{-1}$ |
| DeBERTa v2 XXL (21) | MRPC | $8.15e^{-2}$ | $6.89e^{-2}$ | $6.53e^{-2}$ | $8.09e^{-2}$ | $8.02e^{-2}$ | $9.08e^{-2}$ | $9.31e^{-2}$ | $1.10e^{-1}$ | $9.47e^{-2}$ | $1.22e^{-1}$ |
| | TREC50 | $3.38e^{-1}$ | $2.36e^{-1}$ | $7.04e^{-2}$ | $1.53e^{-1}$ | $2.24e^{-1}$ | $1.36e^{-1}$ | $1.85e^{-1}$ | $2.22e^{-1}$ | $1.93e^{-1}$ | $1.92e^{-1}$ |
| | PAWS | $6.07e^{-2}$ | $1.99e^{-2}$ | $3.63e^{-2}$ | $3.26e^{-2}$ | $3.95e^{-2}$ | $5.41e^{-2}$ | $6.68e^{-2}$ | $5.11e^{-2}$ | $1.98e^{-2}$ | $6.99 e^{-2}$ |
| DeBERTa v3 Base (20) | MRPC | $1.06e^{-1}$ | $8.90e^{-2}$ | $2.59e^{-2}$ | $7.28e^{-2}$ | $9.86e^{-2}$ | $1.52e^{-2}$ | $2.58e^{-2}$ | $8.52e^{-3}$ | $1.16e^{-1}$ | $2.57e^{-2}$ |
| | TREC50 | $4.56e^{-1}$ | $2.73e^{-1}$ | $3.99e^{-1}$ | $2.16e^{-1}$ | $2.67e^{-1}$ | $2.61e^{-2}$ | $2.25e^{-1}$ | $3.70e^{-1}$ | $3.36e^{-1}$ | $2.63e^{-1}$ |
| | PAWS | $2.62e^{-2}$ | $6.43e^{-2}$ | $2.40e^{-2}$ | $6.27e^{-2}$ | $8.17e^{-2}$ | $5.55e^{-2}$ | $7.39e^{-2}$ | $5.77e^{-2}$ | $1.01e^{-1}$ | $5.82e^{-2}$ |
| RoBERTa-Base (35) | MRPC | $9.48e^{-1}$ | $6.01e^{-1}$ | $2.05e^{-1}$ | $1.64e^{-1}$ | $2.20e^{-1}$ | $5.33e^{-1}$ | $4.37e^{-1}$ | $3.78e^{-1}$ | $3.35e^{-1}$ | $3.21e^{-1}$ |
| | CoLA | $1.39$ | $7.74e^{-1}$ | $4.04e^{-1}$ | $2.22e^{-1}$ | $1.96e^{-1}$ | $8.10e^{-1}$ | $4.70e^{-1}$ | $4.43e^{-1}$ | $4.38e^{-1}$ | $4.01e^{-1}$ |
| RoBERTa-Large (35) | MRPC | $7.29e^{-1}$ | $4.64e^{-1}$ | $4.71e^{-1}$ | $2.77e^{-1}$ | $2.68e^{-1}$ | $2.64e^{-1}$ | $6.54e^{-1}$ | $5.57e^{-1}$ | $5.27e^{-1}$ | $3.84e^{-1}$ |
| | CoLA | $8.06e^{-1}$ | $4.25e^{-1}$ | $4.18e^{-1}$ | $2.36e^{-1}$ | $1.75e^{-1}$ | $2.28e^{-1}$ | $4.96e^{-1}$ | $4.56e^{-1}$ | $6.14e^{-1}$ | $4.05e^{-1}$ |
| TinyLlama (61) | OpenBookQA | $1.78e^{-1}$ | $2.82e^{-1}$ | $3.41e^{-1}$ | $2.15e^{-1}$ | $1.86e^{-1}$ | $2.07e^{-1}$ | $1.51e^{-1}$ | $2.20e^{-1}$ | $3.16e^{-1}$ | $7.59e^{-2}$ |
| | FOLIO | $1.82e^{-1}$ | $2.37e^{-1}$ | $2.17e^{-1}$ | $1.75e^{-1}$ | $1.93e^{-1}$ | $5.11e^{-2}$ | $2.35e^{-1}$ | $1.91e^{-1}$ | $1.05e^{-1}$ | $2.49e^{-1}$ |
| | LogiQA | $3.61e^{-1}$ | $6.12e^{-3}$ | $1.45e^{-1}$ | $1.16e^{-2}$ | $1.75e^{-1}$ | $2.37e^{-1}$ | $8.60e^{-2}$ | $1.1e^{-1}$ | $6.64e^{-2}$ | $6.25e^{-2}$ |
| | CLUTRR | $4.29$ | $2.25$ | $1.55$ | $2.34$ | $2.27$ | $5.48$ | $2.16$ | $2.19$ | $2.59$ | $4.23$ |
| DeepseekCoder (16) | DJANGO | $3.48e^{-2}$ | $4.65e^{-2}$ | $3.4e^{-2}$ | $5.16e^{-2}$ | $4.64e^{-2}$ | $3.87e^{-2}$ | $4.19e^{-2}$ | $3.89e^{-2}$ | $3.64e^{-2}$ | $3.62e^{-2}$ |
| GPT2-Small | E2E | $1.65e^{-1}$ | $1.93e^{-1}$ | $1.85e^{-1}$ | $1.83e^{-1}$ | $1.85e^{-1}$ | $1.87e^{-1}$ | $1.77e^{-1}$ | $1.82e^{-1}$ | $1.88e^{-1}$ | $1.82e^{-1}$ |

model, it can be used as a heuristic for determining generalization. An important aspect of evaluating fine-tuning methods is not only their peak performance but also their consistency across training runs. Understanding how stable these models are provides insight into their reliability and reputability for practical use.

Tying our theoretical developments to our empirical tests, we see a connection where the frozen variants tend to have a lower difference between test loss and train loss over varying epochs.

An important aspect of evaluating fine-tuning methods is not only their peak performance but also their consistency across training runs. Understanding how stable these models are provides insight into their reliability and reputability for practical use.

# E  LIMITATIONS AND DISCUSSION

cLA and $c^3$LA particularly train only a small subsection of our pretrained model at a time, leading to underperformance on lower ranks in comparison to alternate LoRA variants.

We observed that cLA and $c^3$LA performed nearly as well as their non-sparse counterparts, Asymmetric LoRA and RAC, while being less expensive. The nature of the methods they were inspired by already had a frozen matrix component; we leave it up to researchers to study more potential identity-based LoRA variants to save computational resources.

We emphasize that some of the analytical tools in §4.2 are not necessarily strong indicators of a model's performance. However, they tell us about some of the fine-tuned model's subspace properties, such as changes in direction and magnitude. Particularly, they depict how a fine-tuned model deviates from a pretrained model. This is relevant if the preservation of structure is important for alternative purposes such as cross-training, hybrid fine-tuning, or preservation of historical datasets.

# F  TABLE OF NOTATIONS

Table 16: Table of notations.

| Notation | Definition |
|---|---|
| $\|x\|$ | The $\ell_2$ norm of a vector, $x$ |
| $\|A\|$ | The Frobenius norm of a matrix, $A$ |
| $\|A\|_2$ | The spectral norm of a matrix, $A$ |
| $A^\dagger$ | The Moore-Penrose pseudoinverse of a matrix $A$. |
| $L$ | Number of layers in a deep neural network |
| $W^i$ | $i^{\text{th}}$ layer of network |
| $\mathbf{W}$ | $(W^1, ..., W^L)$ |
| $x$ | Input to the network |
| $f_{\mathbf{W}}(x)$ | $\sigma^L(W^L \cdots \sigma^3(W^3 \sigma^2(W^2 \sigma^1(W^1(x))...)))$ |
| $\sigma_i(\cdot)$ | $i^{\text{th}}$ layer non-linear activation function |
| $N_{\text{pre}}$ | pre-training dataset $(x_i, y_i)_{i=1}^{|N_{\text{pre}}|}$ |
| $\ell_{\text{pre}}(\cdot)$ | pre-training loss function |
| $\mathbf{W}_0$ | pre-training weights |
| $\Delta\mathbf{W}$ | FFT weight-update |
| $\Delta\hat{\mathbf{W}}$ | FFT argmin update |
| $\ell(\cdot)$ | fine-tuning loss function |
| $\mathbf{BA}$ | LoRA weight-update |
| $\hat{\mathbf{B}}\hat{\mathbf{A}}$ | LoRA argmin weight update |
| $k$ | Chain-length of chain methods (CoLA, RAC, C3LA) |
| $\mathbf{B}^j \mathbf{A}^j$ | CoLA $j^{\text{th}}$ chain weight update |
| $\hat{\mathbf{B}}^j \hat{\mathbf{A}}^j$ | CoLA $j^{\text{th}}$ chain argmin weight update |
| $\mathbf{W}_0^{(k,BA)}$ | $k$ chains of CoLA updates, where $\mathbf{W}_0^{(k,BA)} := \mathbf{W}_0 + \sum_{j=1}^k \hat{\mathbf{B}}^j \hat{\mathbf{A}}^j$ |
| $\mathbf{A}_0$ | Frozen $A$ layers. |
| $\mathbf{BA}_0$ | Assymetric LoRA weight update |
| $\hat{\mathbf{B}}\mathbf{A}_0$ | Assymetric LoRA argmin weight update |
| $\mathbf{B}^j \mathbf{A}_0^j$ | RAC-LoRA $j^{\text{th}}$ chain weight update |
| $\hat{\mathbf{B}}^j \mathbf{A}_0^j$ | RAC-LoRA $j^{\text{th}}$ chain argmin weight update |
| $\mathbf{W}_0^{(k,B)}$ | $k$ chains of RAC-LoRA updates, where $\mathbf{W}_0^{(k,B)} := \mathbf{W}_0 + \sum_{j=1}^k \hat{\mathbf{B}}^j \mathbf{A}_0^j$ |
| $\mathbf{B}^c$ | Cheap LoRA (cLA) weight update |
| $\hat{\mathbf{B}}^c$ | cLA argmin weight update |
| $\mathbf{B}^{c^3,j}$ | Circulant chain of cheap LoRA's ($c^3$LA) $j^{\text{th}}$ chain weight update |
| $\hat{\mathbf{B}}^{c^3}$ | $c^3$LA $j^{\text{th}}$ chain argmin weight update |
| $\mathbf{W}_0^{(k,B^{c^3})}$ | $k$ chains of $c^3$LA updates, where $\mathbf{W}_0^{(k,B^{c^3})} := \mathbf{W}_0 + \sum_{j=1}^k \hat{\mathbf{B}}^{c^3,j}$ |
| $L_G$ | Lipschitz constant for the gradient of the loss function. |
| $\mathcal{X}$ | feature space of the network |
| $\mathcal{Y}$ | label space of the network |
| $\hat{\mathcal{L}}_{\text{global}}(\cdot)$ | true risk of an input network |

