# OpenReview forum: "LoRA: The Past, Present, and Future"
_ICLR.cc/2026/Conference — ICLR 2026 Conference Withdrawn Submission_

### Official Review · Reviewer_CsQR · 2025-10-26

**Soundness:** 2
**Presentation:** 1
**Contribution:** 2
**Rating:** 2
**Confidence:** 4

**Summary:**

The paper investigates Low-Rank Adaptation (LoRA) and its variants. It introduces new, computationally more efficient extensions, such as Cheap LoRA (cLA) and its variants, along with a chained circulant variant LA. The study conducts theoretical analyses, including novel information-theoretic generalization error bounds, and an extensive empirical study showing that cheaper LoRA variants can reduce costs and improve generalization.

**Strengths:**

The paper provides in depth theoretical analysis of the proposed methods.
The proposed methods were examined in comparison with different LoRA variations in several benchmarks.

**Weaknesses:**

Citation format should be revised utilizing the traditional format. Using reference numbers in parenthesis is very confusing since equations are referenced with the same format. Therefore, the paper is not easily readable.

Notation should be revised and redundancy in some terms should be fixed.

In the paper, it is stated as: We extend this principle to cLA with a structured chaining, c^3LA. Let B^c^3 denote c^3LA’s update. The c^3LA and B^c^3 should be defined precisely.

The paper states that: We report empirical results regarding the computational efficiency of PEFT methods developed in this paper. We report the percentage of trainable parameters for each PEFT method in Table 4.

However, a comparison of the number of trainable parameters for the proposed and baseline LoRA methods was not given.

The paper states that: Our focus in this study is on examining the behaviors of the PEFT methods, including our proposed variant.

Although the total fine-tuning time is an important factor, we do not present wall-clock results because our unoptimized implementation does not present a valid point for the training speedup that well-engineered PEFT methods can offer.

However, the paper also claims that: We present a historical framing (the past: full fine-tuning and original LoRA; the present: different variants of LoRA) and introduce simpler, cheaper, parameter-efficient extensions: Cheap LoRA (cLA).

That is, one of the main claims is the computational efficiency of the proposed methods. However, this was not analyzed and verified in the paper.

**Questions:**

How do you define B^c^3 and c^3LA, more precisely?

Why do you initialize A^i by sampling from a Gaussian distribution with variance 0.02 in the theoretical setup of LoRA?

Could you please provide a detailed comparative complexity analysis of the proposed methods and the other LoRA variants?

Could you please provide a detailed comparative analysis of convergence rates of the proposed methods and the other LoRA variants?

---

### Official Review · Reviewer_5TiF · 2025-10-30

**Soundness:** 2
**Presentation:** 2
**Contribution:** 2
**Rating:** 4
**Confidence:** 3

**Summary:**

This paper presents a broad conceptual and empirical study of low-rank adaptation (LoRA) and its variants. It proposes two new parameter-efficient fine-tuning (PEFT) methods — Cheap LoRA (cLA) and Circulant Chain Cheap LoRA (c3LA) — which simplify LoRA by freezing one of the low-rank matrices and introducing structured chaining. The authors claim theoretical contributions via generalization error bounds and nonconvex convergence analysis, and conduct extensive experiments across nine pre-trained models (NLP, vision, coding, and reasoning tasks) comparing 9 LoRA-based methods and full fine-tuning.

However, the paper suffers from a critical flaw that undermines its core narrative: a significant misalignment between its stated conclusions and its empirical evidence. The claim that cheaper LoRA variants are "advantageous" is not consistently or convincingly supported by the results. Furthermore, the practical utility of the proposed methods is questionable, and several analytical choices lack justification.

**Strengths:**

Breadth of empirical study: The authors evaluate across a diverse set of models (language, vision, code), which is rare in LoRA-related literature.

Theoretical ambition: Attempts to unify PEFT variants under a generalization-error and convergence framework are commendable.

Novel variants: cLA and c3LA are intuitive and computationally motivated; the idea of fixing one low-rank factor and introducing circulant chaining is elegant and easy to implement.

Relevance: Parameter-efficient fine-tuning remains highly relevant for large models and edge deployment, aligning well with ICLR’s audience.

Comprehensive references and contextual framing: The paper situates LoRA in its historical and practical context effectively.

**Weaknesses:**

1. The central conclusion is that "it is advantageous to use [the] cheaper variants for effective cost reduction and a better generalizability". However, the data in Table 2 consistently shows that the proposed methods (cLA, c³LA) rarely achieve top performance. They are often middling or, in some cases (e.g., DeepseekCoder on DJANGO, TinyLlama on OpenBookQA), perform drastically worse than existing methods. The argument then pivots to suggesting that one should choose methods based on "characteristics and user-specific needs rather than on generated accuracy." This is a defensible position, but the paper fails to clearly demonstrate a compelling use-case for its new variants. The computational savings shown in Table 12 are minimal ("naïve sparse implementation"), and the proposed methods do not consistently show superior generalizability (Table 3) to justify their accuracy trade-offs. The conclusion feels like a post-hoc rationalization of mediocre results rather than a finding supported by evidence.

2. The performance of cLA and c³LA is often significantly weaker, especially at lower ranks and epochs. The claim that they "generalize well" (Page 7) even when accuracy is low is not a persuasive argument for adoption in most practical scenarios where predictive performance is paramount. The computational benefit is overstated. The FLOP reduction from the inherent sparsity (Table 12) is marginal. Without a highly optimized kernel that demonstrates meaningful wall-clock speedup, the practical advantage of these "cheaper" variants is not proven.

3. The core idea (fixing one factor in LoRA) is incremental and has been explored under asymmetric LoRA and sparse low-rank adaptations. The claimed “future” direction (c3LA) does not show clear empirical superiority or distinctive behavior justifying its introduction.

**Questions:**

a. How sensitive are cLA and c3LA to the choice of rank r and chain length k?

b. Why does deterministic vs. randomized initialization make little difference—can this be theoretically explained?

c. How were the FLOP reductions computed (are sparsity and practical speedups validated)?

d. Are the theoretical results empirically verifiable beyond toy examples (e.g., bounding G(W) in practice)?

**Details Of Ethics Concerns:**

This is a "kitchen-sink" paper that does a lot but lacks a coherent and supported main message. The extensive benchmarking and theoretical contributions are valuable to the field and could warrant acceptance at a weaker venue. However, for ICLR, the fundamental disconnect between the promotional language ("advantageous," "cheaper," "better generalizability") and the empirical results, coupled with the inconclusive auxiliary analysis, is a critical flaw. The work would be significantly strengthened by tempering its claims and reframing itself as a extensive benchmark and theoretical study that reveals the complex, task-dependent nature of PEFT performance, rather than as a proposal for new state-of-the-art methods.

---

### Official Review · Reviewer_adqS · 2025-10-31

**Soundness:** 3
**Presentation:** 2
**Contribution:** 1
**Rating:** 4
**Confidence:** 2

**Summary:**

This paper reviews existing LoRA-based parameter-efficient fine-tuning methods and proposes two simplified sparse variants—cLA and c³LA—to reduce computational costs. It presents theoretical analyses covering generalization and convergence behaviors across multiple LoRA variants. Empirically, the authors compare these methods across a diverse set of models and tasks, concluding no single method universally dominates, although the proposed sparse variants achieve reasonable performance with lower adapter complexity.

**Strengths:**

- Theoretical Analysis: The paper's main contribution is a clear theoretical framework that systematically explains and compares the generalization and convergence properties of various LoRA-style methods.
- Practical Simplicity: The introduced cLA and c³LA methods are straightforward and intuitive, potentially useful for practitioners seeking simplified parameter-efficient solutions.

**Weaknesses:**

- Limited Novelty: The proposed methods are incremental, largely reusing known sparse and chained adapter ideas with only modest structural changes. The conceptual innovation is limited relative to prior PEFT literature.
- Lack of Empirical Validation for Efficiency: The efficiency claims rely solely on theoretical FLOP counts without any real measurements of training speed or memory footprint. Modern GPU efficiency is heavily architecture-dependent—kernels optimized for dense matrix multiplications may not favor sparse or fixed-structure adapters. Without actual profiling (e.g., wall-clock time, throughput, memory usage), the practical efficiency advantage remains speculative.
- Small-Scale Experiments: Most experiments use small models such as TinyLlama, which limits the generality of the conclusions. Validation on larger, modern architectures like Llama3 8B or recent code/math benchmarks would better demonstrate scalability and relevance.

**Questions:**

See weakness

---

### Official Review · Reviewer_Sb6e · 2025-11-01

**Soundness:** 3
**Presentation:** 3
**Contribution:** 2
**Rating:** 4
**Confidence:** 4

**Summary:**

The key contributions of this paper are two-fold. Firstly, the paper provides a theoretical analysis of the convergence properties of current PEFT methods, including LoRA, LoRA+, Asym-LoRA, CoLA, and RAC. Secondly, the paper proposes four extensions of LoRA to improve the parameter efficiency. These methods are validated theoretically in terms of convergence behavior, and empirically on various pretrained models.

**Strengths:**

The paper demonstrates several notable strengths:

+ It presents a novel and rigorous theoretical analysis of the convergence behaviors of various LoRA extensions from an information-theoretic perspective, offering new insights into their underlying mechanisms.

+ The authors propose four methods that achieve greater computational efficiency while preserving the strong performance characteristic of standard LoRA approaches.

+ The paper is clearly and coherently written, with a particularly effective presentation of the theoretical results and their implications. I have examined the theoretical proofs in detail and find them to be correct and logically sound.

**Weaknesses:**

The paper has the following weaknesses:

+ **About the contributions:** The contributions of the proposed methods are not clearly established. From a theoretical view, the generalization error upper bound of random-cLA is identical to that of Asym-LoRA, and the bound for cLA matches that of RAC. This suggests that the proposed methods do not introduce any substantial theoretical novelty. On the empirical side, the authors claim that their algorithms improve computational efficiency over prior LoRA extensions. However, this claim is not adequately supported by the presented experiments: the paper does not report computational overhead, and according to Table 2, the performance of the proposed methods is slightly inferior to LoRA and CoLA. To strengthen this section, it would be helpful for the authors to include additional details such as the number of trainable parameters and the average performance metrics of each method.

+ **About the theoretical results:** The theoretical analysis is conducted under the assumption that the network is a simple DNN. However, LoRA and its extensions are most commonly applied to Transformer-based architectures. Therefore, the theoretical results would be more relevant and impactful if the analysis were extended to even a simplified Transformer-based model. Although the paper claims that the proposed framework can be adapted to more complex architectures such as CNNs, RNNs, and Transformers with minor modifications, this extension appears nontrivial and requires further clarification and justification.

+ **Other minor issues:** It appears that some figures and headings have been compressed using \vspace{} commands. Moreover, certain plots—such as Figure 1—are slightly blurry and could be improved for better clarity in the revised version.

**Questions:**

Apart from the issues noted in the weaknesses section, I have no additional concerns. I would be willing to raise my evaluation if these concerns are effectively addressed.

---

### Note · Authors · 2025-11-12

I have read and agree with the venue's withdrawal policy on behalf of myself and my co-authors.